# Androgens show sex-dependent differences in myelination in immune and non-immune murine models of CNS demyelination

Amina Zahaf[1], Abdelmoumen Kassoussi[1], Tom Hutteau-Hamel[2], Amine Mellouk[2], Corentine Marie[3], Lida Zoupi [4], Foteini Tsouki [4], Claudia Mattern[5], Pierre Bobé[2], Michael Schumacher [1], Anna Williams [4], Carlos Parras [3] & Elisabeth Traiffort [1] ✉

Neuroprotective, anti-inflammatory, and remyelinating properties of androgens are well-characterized in demyelinated male mice and men suffering from multiple sclerosis. However, androgen effects mediated by the androgen receptor (AR), have been only poorly studied in females who make low androgen levels. Here, we show a predominant microglial AR expression in demyelinated lesions from female mice and women with multiple sclerosis, but virtually undetectable AR expression in lesions from male animals and men with multiple sclerosis. In female mice, androgens and estrogens act in a synergistic way while androgens drive microglia response towards regeneration. Transcriptomic comparisons of demyelinated mouse spinal cords indicate that, regardless of the sex, androgens up-regulate genes related to neuronal function integrity and myelin production. Depending on the sex, androgens down-regulate genes related to the immune system in females and lipid catabolism in males. Thus, androgens are required for proper myelin regeneration in females and therapeutic approaches of demyelinating diseases need to consider male-female differences.

Multiple sclerosis (MS), known as the most common cause of non-traumatic disability in young adults, is characterized as an auto-immune, demyelinating and neurodegenerative pathology of the central nervous system (CNS). If the early relapsing-remitting form leads to spontaneous regeneration of lost myelin (or remyelination), the latter fails in progressive MS resulting in irreversible neurological disabilities[1]. The disease is sexually dimorphic, namely with a three-fold higher prevalence in women and more severe forms occurring at a later age in men[2,3]. Any dysregulation of the female or male sexual hormones were correlated with worsening of the disease[4-6].

The experimental autoimmune encephalomyelitis (EAE), used as an immune model for MS, led to show the neuroprotective / anti-inflammatory and remyelinating effects of estrogens in females through the astroglial estrogen receptors (ER) α and the oligoden-droglial ERβ[7,8]. Similarly, neuroprotective and anti-inflammatory properties of androgens - mediated through the androgen receptor (AR) – were characterized in males upon their preventive[9-12] or curative[13] administration in EAE animals and namely attributed to the negative selection of T cells depending on AR-mediated up-regulation of the autoimmune regulator Aire in the thymus[13]. Androgen remyeli-nating properties demonstrated in the cuprizone model of chronic CNS demyelination[14] and the model of focal demyelination induced by stereotactic injection of lysolecithin (LPC) demonstrated AR-mediated recruitment of oligodendrocyte progenitor cells (OPCs) and their dif-ferentiation into oligodendrocytes, the CNS myelinating cells[14,15]. The apparent critical role of sexual hormones in MS also led to clinical trials

[1]U1195 Inserm, Paris-Saclay University, Kremlin-Bicêtre, France. [2]UMR996 Inserm, Paris-Saclay University, Clamart, France. [3]Paris Brain Institute, Sorbonne University, Paris, France. [4]Centre for Regenerative Medicine, Institute for Regeneration and Repair, The University of Edinburgh, Edinburgh BioQuarter, Edinburgh, UK. [5]M et P Pharma AG, Emmetten, Switzerland. ✉e-mail: elisabeth.traiffort@inserm.fr

in relapsing-remitting MS using the pregnancy hormone estriol in women and testosterone in men, which led to improved subclinical markers of disease activity[16] and reduced brain atrophy, respectively[17,18].

Although the hormonal environment of males and females is obviously different, it should however not be restricted to the existence of high androgen levels in males, and fluctuating estrogen and progesterone levels in females since males also make estrogens, particularly in the brain that has high levels of aromatase enzyme converting testosterone into estradiol, and females also make low levels of androgens[19]. In male mice, prophylactic administration of estrogens reduces EAE incidence and severity[10] while testosterone-induced remyelination upon LPC injection requires aromatase activity for recruiting OPCs to the lesion[20]. In female mice, preventive administration of androgens beneficially controlled T cell-mediated spleen secretion of the pro- and anti-inflammatory cytokines IFN-γ and IL-10, respectively[9,21] whereas androgens prevented the contact between EAE female T cells and astrocytes responsible for the production of pro-inflammatory molecules in vitro[22].

However, it is not known if androgens are required for spontaneous remyelination in females or how androgens may act in the female demyelinated CNS. Here, we addressed these questions by using pharmacological and genetic approaches in immune and non-immune models of CNS demyelination. We show strong AR up-regulation in the demyelinated lesions from female mice and MS female patients mostly in microglia/macrophages, but not from male animals and patients. In females, we demonstrate the synergism of androgens and estrogens for increasing OPC recruitment and the unique involvement of androgens in the response of microglia/macrophages to demyelination. Finally, we uncover sexual dimorphism characterizing dihydrotestosterone (DHT) effects in the demyelinated CNS at the molecular level.

## Results

### AR is up-regulated in demyelinated female but not male patients

We delineated AR transcript expression in the corpus callosum from female and male mice demyelinated by LPC stereotactic injection (Fig. 1a). At 7 days post-lesion (dpl), we observed a strong AR up-regulation in female lesions (Fig. 1b, d) while AR transcripts could be detected at much lower level in male lesions (Fig. 1c, d). In contrast, the cerebral cortex from both sexes displayed high AR transcription. In female lesions, AR and the microglial Iba1 staining were substantially colocalized suggesting that AR was mostly up-regulated in microglia (and possibly infiltrated macrophages[23,24]) and at a much lower level in GFAP-expressing astrocytes and Olig2+ oligodendrocytes (Fig. 1e). Antibodies directed to AR and the main conversion product of testosterone, DHT, led to confirm AR expression in microglia and a clear DHT staining in both nuclear and perinuclear areas (Fig. 1f–h). In males, the cortex mostly displayed AR+ Iba1- and AR+ GFAP- cells (Fig. 1i) in agreement with the previously reported neuronal expression of AR[25]. The faint AR expression in the lesion incited us to visualize the transcription of the estrogen receptors ERα and ERβ. Both transcripts (Esr1 and Esr2, respectively) were observed in female and male lesions (Fig. 1j, k). However, Esr1 expression was significantly higher in male lesions compared to Esr2 and to both Esr1 and Esr2 in female lesions (Fig. 1l, m). Thus, the female demyelinated areas display a high level of AR protein and transcripts whereas the demyelinated male lesions display quite undetectable AR but substantial Esr1 expression.

To evaluate the relevance of AR expression in MS lesions, we performed AR in situ hybridization (ISH) and immunostaining experiments in brain sections from non-neurological control and MS female and male donors (Supplementary Tables 1 and 2). White matter samples were used to detect AR mRNA expression in immunofluorescently labeled Iba1+ microglia/macrophages. Quantification showed a significantly greater proportion of Iba1+ cells expressing AR mRNA in MS

samples compared to controls with a significant effect of sex, as significantly more Iba1+ cells expressed AR in females (Fig. 2a, b). Within MS donors, female samples showed a significantly greater proportion of AR+Iba1+ cells compared to males without significant differences among different demyelinated lesion types (Fig. 2c). Additional female and male gray matter were immunostained and showed AR protein expression in many cells with the morphology of neurons, while the white matter had some but not much AR expression in both sexes (Supplementary Fig. 1a, b, h, i). However, in females with MS, cells expressing AR protein appeared to be increased in active lesions and perilesional areas of chronic active lesions, where there is potential to remyelinate, and fewer in the center of chronic active lesions and normal appearing white matter (Supplementary Fig. 1c–f). In males, there was little AR expression in active or chronic active lesions (Supplementary Fig. 1j, k). In females, some lesional AR+ cells were CD68+ corresponding to activated microglia/macrophages (Supplementary Fig. 1g). Also in support of these data, AR mRNA expression in microglia from MS and control donors from a publicly available single-nuclei RNA sequencing database[26] appeared higher in MS female samples compared to males (Supplementary Fig. 1l). Altogether these data confirmed that the sexual discrepancy observed in mice may be relevant in human and microglia/macrophages may be a target for AR signaling during remyelination in females.

### Androgens induce remyelination in LPC-demyelinated females

AR up-regulation in the demyelinated female mice and patients led us to delineate the effects of testosterone and DHT, which both are AR ligands, even though testosterone may also act via ERs. We induced LPC-demyelination of ovariectomized female animals that received a daily intranasal administration of testosterone, DHT or vehicle starting 15 hrs after LPC injection (Fig. 3a). At 7 dpl, only testosterone significantly increased the density of PDGFRα+ OPCs (p = 0.001) without significant modification of the number of PDGFRα+ Ki67+ proliferating OPCs (Fig. 3b, c). In contrast, both molecules increased the percentage of CC1+ differentiated oligodendrocytes (p = 0.0004 and p = 0.0084, respectively). As expected from the increased number of OPCs, the total number of Olig2+ oligodendroglial cells was significantly increased under testosterone (Fig. 3d, e, p = 0.0016). Testosterone and DHT promoted the expression of one of the main myelin proteins, MBP (p < 0.0001) with however a significantly stronger effect of testosterone (Fig. 3f, g; p = 0.005) suggesting that the latter may induce additional effects. Then, we evaluated the local inflammatory cells, microglia and astrocytes. Both androgens significantly decreased the amount of microglia in the lesion (Fig. 3h, i; p < 0.0001). Only DHT was able to promote the expression of the anti-inflammatory microglial marker Arg-1 (p = 0.0003). Moreover, unlike previous data from LPC-demyelinated males[15,20], testosterone and DHT (Fig. 3j, k) significantly decreased GFAP (p < 0.0001) and STAT3 (p < 0.0001) labeling suggesting a decreased astrocyte reactivity in testosterone- and DHT-treated females. To quantify remyelination under DHT treatment, we evaluated myelin sheath thickness by determining the g-ratio (axon diameter / total outer diameter of the myelinated fiber) at 14 dpl (Fig. 3l, m). Electron microscopy visualized a higher number of myelinated axons in DHT-treated females consistent with the significant decrease of the g-ratio values plotted according to axon diameters and mean g-ratio value (Fig. 3n, o; p < 0.0001). Thus, androgens promote remyelination in female mice as previously described in male mice with nevertheless differences between testosterone and DHT-induced control of local inflammatory cells and oligodendrocytes.

### Synergistic androgen and estrogen effects in myelin repair in females

The differential effects observed above for testosterone compared to DHT suggested that testosterone could induce its effects via both AR and ER after its aromatase-mediated conversion to estradiol (E2). In a

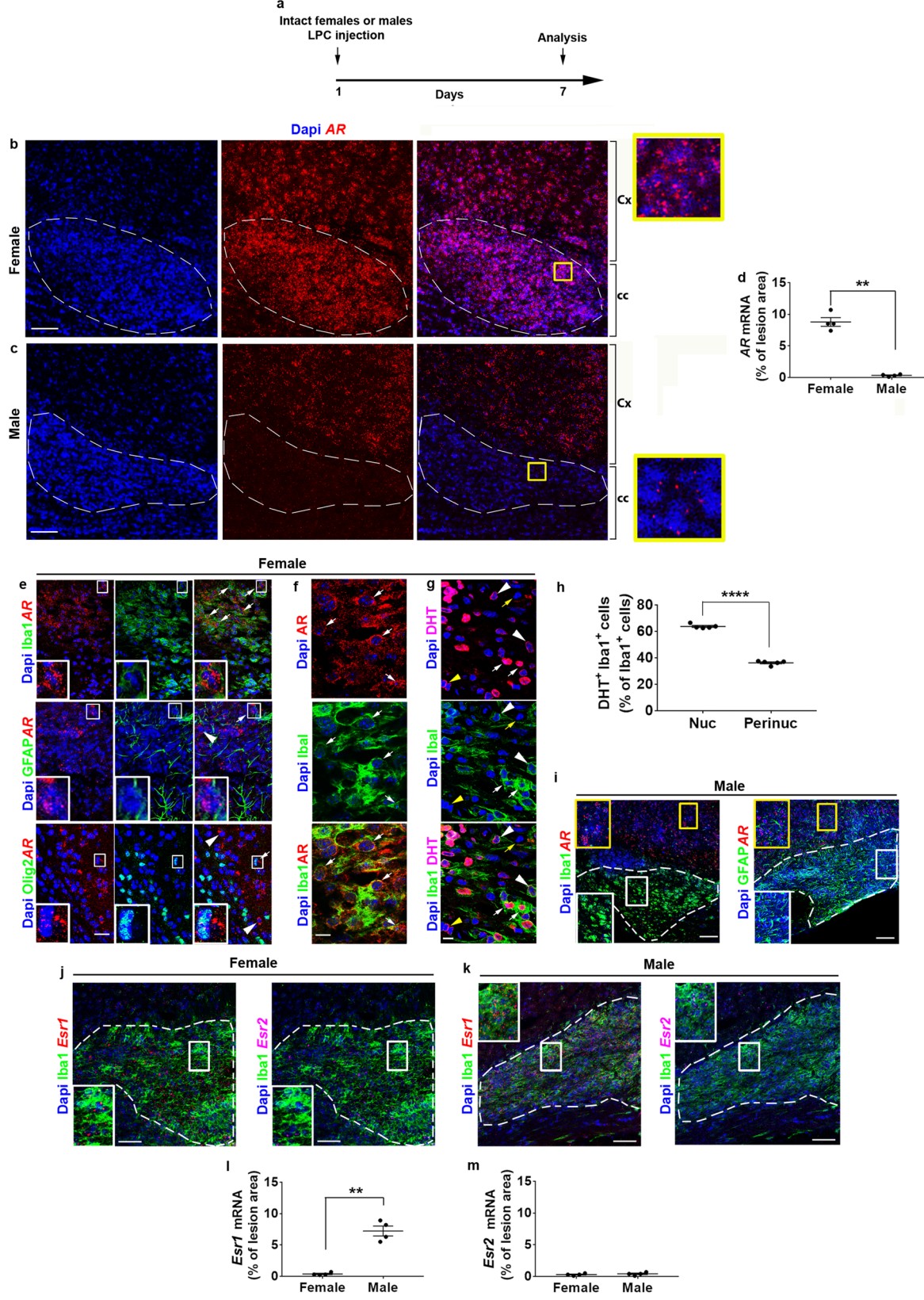

consistent way, aromatase was up-regulated in the female mouse demyelinated area, independently of the mechanical injury induced by the injection (Supplementary Fig. 2). We analyzed the LPC lesions from female mice treated with intranasal administration of vehicle, DHT, E2 or DHT + E2 at 7 dpl (Fig. 4a). Neither DHT nor E2 administered alone modified the number of OPCs unlike DHT + E2, which significantly

increased them (Fig. 4b, c; p = 0.034) without increasing their proliferation. Similarly, only DHT + E2 increased the total number of Olig2⁺ cells (Fig. 4d, e; p < 0.0001). In contrast, the percentage of CC1⁺ oligodendrocytes (p < 0.0001; p = 0.0002; p < 0.0001, for DHT, E2 and DHT + E2, respectively) and MBP (p < 0.0001) immunolabeling were increased regardless of the treatment with nevertheless a significantly

**Fig. 1 | The androgen receptor is strongly up-regulated in the LPC-demyelinated corpus callosum from female but not male mice. a** Scheme of the experimental protocol. **b, c** Differential detection of *AR* transcripts in the corpus callosum (cc), but not cortex (Cx) from LPC-injected females or males. Dashed lines delineate the demyelinated area. **d** *AR* signal quantification in the lesions. **e** Double *AR* ISH (left) and Iba1, GFAP or Olig2 immunostainings (middle) and, merge images (right) in the female demyelinated lesions. The white arrows show a high number of AR-expressing microglial cells compared to a more restricted number of *AR*+GFAP+ astrocytes or *AR*+Olig2+ oligodendroglia. The white arrowheads indicate AR-expressing cells clearly devoid of GFAP or Olig2 markers. The boxed areas are magnified in the insets. **f, g** Visualization of cells co-expressing either the AR protein or the DHT ligand with Iba1 marker. The white arrows show microglia co-expressing nuclear/perinuclear AR **f** and nuclear DHT staining **g**. In **g**, microglial or non-microglial (white and yellow arrowheads, respectively) cells displaying a perinuclear DHT labeling or cells expressing none of the markers (yellow arrow) are shown. **h** Quantification of the percentage of Iba1+ cells displaying a nuclear (Nuc) or perinuclear (Perinuc) labeling. **i** Co-vizualization of *AR* transcripts with Iba1 or GFAP immunostainings in the demyelinated corpus callosum (dotted line) from LPC-injected male mice. The boxes are cortical (yellow) and callosal (white) areas magnified in the corresponding inset. **j, k** Triple labeling of female or male lesions using Iba1 immunostaining with *Esr1* and *Esr2* ISH. The lesions are delineated by the dashed lines. The boxed areas are magnified in the insets. *Esr2* can be observed colocalized or not with *Esr1*. **l, m** Quantification of *Esr1* and *Esr2* signals. Scale bars (µm): 100 **b, c, i–k**, 50 **e**, 10 **f, g**. Data are mean values±SEM from n = 4 **d, l, m** or n = 5 **h** animals/condition examined over two independent experiments. *P* values **d, h, l, m** were calculated using the unpaired two-tailed *t*-test with Welch's correction **d, l**; **p = 0.001 **d**, p = 0.003 **l**; ****p < 0.0001 **h**. Source data are provided as a Source Data file.

stronger effect of DHT + E2 compared to E2 alone (*p* = 0.004; Fig. 4d–g). Iba1+ microglial area was also significantly decreased by each treatment compared to the vehicle (Fig. 4h, i; *p* < 0.0001) whereas only DHT alone or combined with E2 promoted the expression of Arg-1 (Fig. 4h, i; *p* < 0.0001). Thus, the effects of estrogens and androgens appeared synergistic for increasing the density of differentiated oligodendrocytes, while DHT induces a specific additional effect on microglia phenotype. We further validated that endogenous testosterone may induce effects via both AR and ER by using the aromatase inhibitor fadrozole[20]. In the presence of fadrozole, testosterone still increased MBP staining (p < 0.0001) even though the increase was significantly lower (*p* = 0.04) than the one induced by testosterone alone (Fig. 4j–l) corroborating the additive effects of DHT and E2 on MBP expression. Moreover, testosterone+fadrozole decreased Iba1 (*p* = 0.0014) and increased Arg-1 (*p* < 0.0001) (Fig. 4m, n) in agreement with the unique ability of DHT to induce this marker.

## AR blockade alters spontaneous remyelination in females

To determine if androgen effects are required for the full regeneration of myelin upon demyelination, we blocked AR by using the specific AR antagonist flutamide (Fig. 5a). At 7 dpl, flutamide significantly decreased the number of OPCs (*p* = 0.0014) and Olig2+ (*p* = 0.03) cells. Moreover, the percentage of Olig2+ CC1+ differentiated oligodendrocytes was decreased in flutamide-treated animals (*p* = 0.002; Fig. 5b–e). As expected, flutamide also decreased MBP staining in the lesion (Fig. 5f, g; *p* = 0.002), increased the level of Iba1+ microglia (*p* = 0.0022) and decreased the proportion of microglia expressing Arg-1 (Fig. 5h, i; *p* = 0.0016). At 10 dpl, the blocking effect of flutamide could still be detected on MBP (Fig. 5j, k, *p* < 0.0001 and Supplementary Fig. 3) and microglia (Fig. 5l, m; *p* < 0.0001). As a whole, these results support the idea that androgen signaling via AR plays a role during remyelination in female mice both in oligodendroglia and microglia.

## AR is required for microglial response to demyelination in females

The specific response of microglia to DHT and the predominant expression of AR in this cell type incited us to conditionally remove AR from CX3CR1+ microglia/macrophages. LPC-demyelinated females displaying or not the floxed AR alleles were treated with DHT or the vehicle and analyzed at 7 dpl (Fig. 6a). In mutant animals, DHT treatment failed to decrease Iba1 staining and to increase the proportion of these cells co-expressing Arg-1 unlike in the wild-type animals (Fig. 6b, c). Moreover, conditional AR removal prevented OPC differentiation into Olig2+CC1+ oligodendrocytes under DHT treatment indicating that microglial AR controls OPC differentiation upon CNS demyelination in female mice (Fig. 6d, e). Similarly, the removal of AR from microglia prevented the ability of DHT to decrease astrogliosis (Fig. 6f, g).

## DHT efficiently mitigates the course of EAE disease in females

Given the important role of AR-mediated signal in myelin regeneration in females and because remyelination cannot be considered independently of the peripheral immune process characterizing MS, we further investigated the role of testosterone and DHT in the EAE model. Androgens were administered according to a curative protocol at onset of neurological symptoms[20] in ovariectomized EAE female mice assigned to intranasal administration of the vehicle, testosterone or DHT for 30 days (Fig. 7a). Vehicle-treated females displayed the typical profile of disease progression with hindlimb paralysis with a 3.0–3.5 clinical score reached by day 8 and persisting until the end of the experiment. Testosterone or DHT-treated females displayed significantly lower scores throughout the whole experiment. However, DHT tended to be less efficient (Fig. 7a) suggesting that testosterone may act via different mechanisms compared with DHT.

MBP immunostaining visualized higher myelin levels in the spinal cord from androgen-treated females compared to the vehicle (Fig. 7b, c; *p* = 0.0004 and *p* = 0.007 for T and DHT, respectively). The detection of the non-phosphorylated neurofilament, Smi-32, an established marker for axonal damage revealed a much lower expression in androgen-treated animals suggesting androgen-mediated neuroprotection in agreement with electron microscopy images. Indeed, a lower number of axons surrounded by thin layers of myelin or devoid of myelin sheaths was observed in the drug-treated mice (Fig. 7d, e). Assessment of myelin sheath thickness revealed that androgens significantly reduced the mean g-ratio compared with the vehicle (*p* < 0.0001). In addition, testosterone treatment resulted in a mean g-ratio value significantly lower (*p* = 0.029) than the value from the DHT-treated animals.

In order to investigate androgen effects on local inflammatory cells, we labeled spinal cord slices with the pan-microglia/macrophage and the anti-inflammatory markers, Iba1 and Arg-1, respectively. Iba1+ staining was significantly decreased by testosterone (*p* = 0.0018) and DHT (*p* = 0.0023) compared to the control, whereas Arg-1 staining was fully collapsed in the presence of androgens (Fig. 7f, g; *p* = 0.0002). GFAP+ astrogliosis also decreased under androgen treatment (*p* < 0.0001) in correlation with the higher MBP level (*p* < 0.0001) observed in those conditions (Fig. 7h, i). The expression of Claudin-5, one of the tight junctional proteins expressed by endothelial cells comprising the blood–brain barrier (BBB), was also increased in the drug-treated animals (Fig. 7j, k; *p* < 0.0001 and *p* = 0.0126 for T and DHT, respectively) suggesting that androgens are involved in the preservation of BBB integrity. In female mice, androgens thus appeared to mitigate the severity of EAE disease including the neurological scores, demyelination, the inflammatory cell density, as well as the expression of the junctional protein Claudin-5.

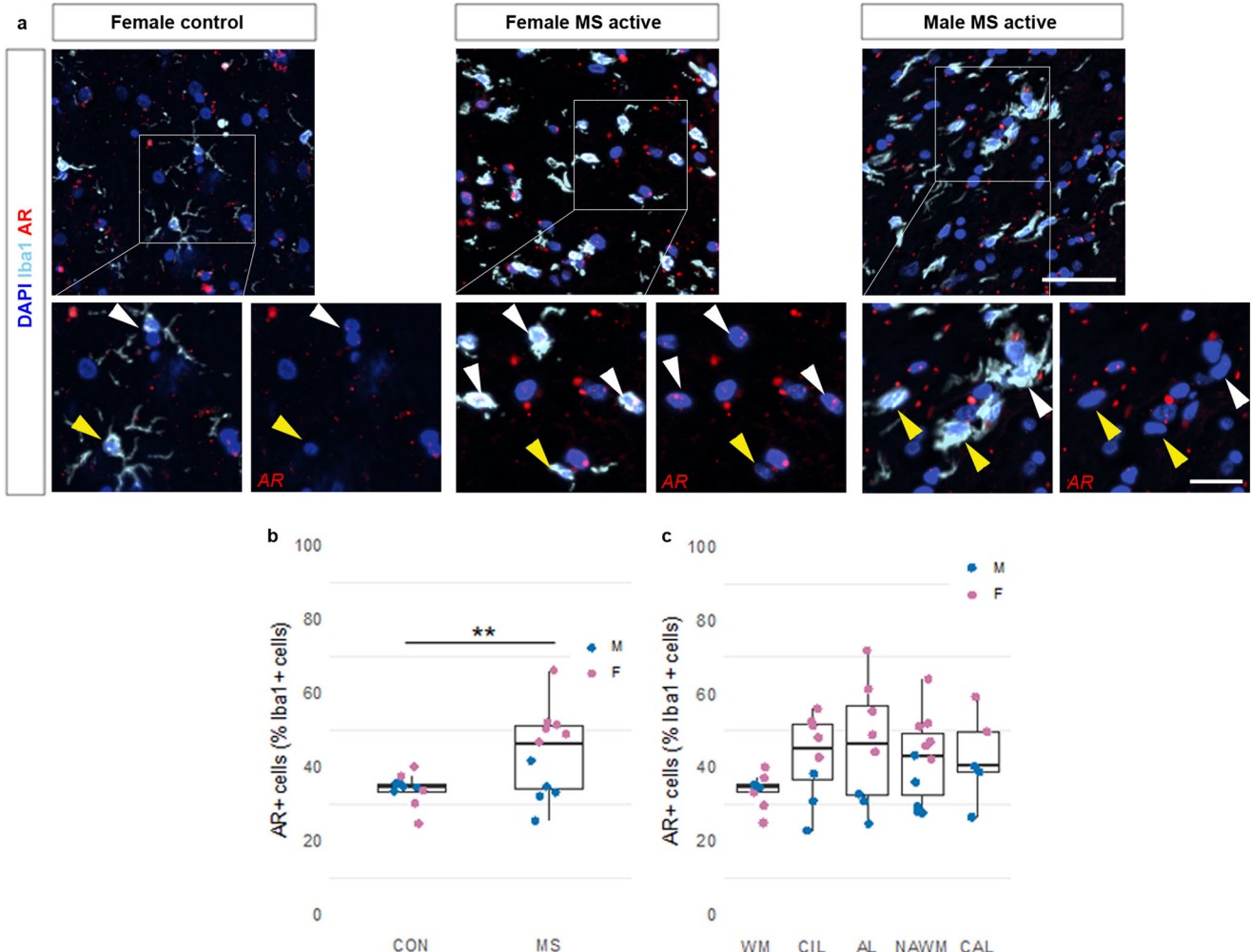

**Fig. 2 | AR mRNA expression in IBA1+ microglia/macrophages is up-regulated in the human female MS brain.** Post-mortem white matter samples from MS and non-neurological control donors were used to detect AR mRNA expression in fluorescently labeled IBA1+ microglia/macrophages. **a** Representative images showing AR mRNA expression (red) within Iba1+ microglia/macrophages (cyan) in white matter from a female control donor and in active white matter demyelinated lesions from a female and male MS donor. Sections were counterstained with DAPI (blue). Scale bar 50 μm. Magnified regions in inserts show AR+ (white arrowheads) and AR− (yellow arrowheads) microglia/macrophages. Scale bar 20 μm. **b** Quantification of RNAscope experiment shows a significantly greater proportion of Iba1+ cells expressing AR mRNA in MS samples compared to controls ($F_{1, 18} = 8.778$, $p = 0.00821$; post-hoc pairwise comparison: $t = 2.963$, $p = 0.0083$; control: median 34.70 (CI 30.00, 37.47), 25% percentile 32.50, 75% percentile 35.98, minimum 25.00, maximum 40.10; MS: median 46.60 (CI 31.92, 52.00), 25% percentile 33.33, 75% percentile 51.43, minimum 25.77, maximum 66.00), with a significant effect of sex, as significantly more Iba1+ cells express AR in females compared to males (linear mixed-effect model: $F_{1, 18} = 10.411$, p = 0.00459; post-

hoc pairwise comparison: $t = 3.231$, $p = 0.0046$). **c** Within MS donors, female samples show a significantly greater proportion of AR+Iba1+ cells compared to males ($F_{1, 8} = 28.579$, $p = 0.000494$). No significant differences were detected among different lesion types ($F_{3, 97} = 2.027$, $p = 0.1151$; WM: median 34.70 (CI 30.00, 37.47), 25% percentile 32.50, 75% percentile 35.98, minimum 25.00, maximum 40.10; CIL: median 45.41 (CI 22.94, 56.00), 25% percentile 32.89, 75% percentile 52.28, minimum 22.94, maximum 56.00; AL: median 46.60 (CI 25.00, 72.03), 25% percentile 31.54, 75% percentile 59.94, minimum 25.00, maximum 72.03; NAWM: median (43.33 (CI 28.26, 52.00), 25% percentile 29.41, 75% percentile 51.36, minimum 27.78, maximum 64.00; CAL: median 40.54 (CI 26.53, 59.21), 25% percentile 32.64, 75% percentile 54.57, minimum 26.53, maximum 59.21). Data were analyzed by linear mixed-effects models followed by ANOVA to determine main effects and Tukey post-hoc pairwise comparisons. Each data point represents the average quantification of 5–10 different regions of interest from the same case ($n = 10$ controls (5 M, 5 F), $n = 11$ MS (5 M, 6 F)). AL active lesion, CAL chronic active lesion, CIL chronic inactive lesion, NAWM normal-appearing white matter, WM white matter from control donors. Source data are provided as a Source Data file.

## DHT decreases deleterious T cells and cytokines only in EAE females

The lower severity of EAE disease observed in females upon androgen treatment could be due in part to the remyelinating effects of the hormones (shown in the LPC model) and/or reflect a lower level of demyelination related to a reduced immune response. We addressed the question by investigating both peripheral immunity and immune cell infiltration into the CNS. Our analysis was restricted to DHT in order to focus specifically on AR-mediated effects. We wondered also if a sex-dependent discrepancy might exist in the control of immune cells by AR-mediated androgen signaling. Ovariectomized EAE female

and castrated EAE male mice received DHT or the vehicle for 8 days from onset of the first neurological symptoms and were analyzed at this early time point when neurological scores become significantly different between the vehicle- and the hormone-treated group before occurrence of any potential compensatory mechanism (Fig. 8a–c).

Immune cells were analyzed by flow cytometry in both the secondary lymphoid organs and the spinal cord by using the gating strategies presented in Supplementary Figs. 4–7. In the spleen from female mice (Fig. 8d), the percentage of CD90+ T cells ($p = 0.0003$), namely CD4+ cells ($p = 0.047$), was significantly decreased. In the draining lymph nodes (Fig. 8e), known to be essential for the balancing

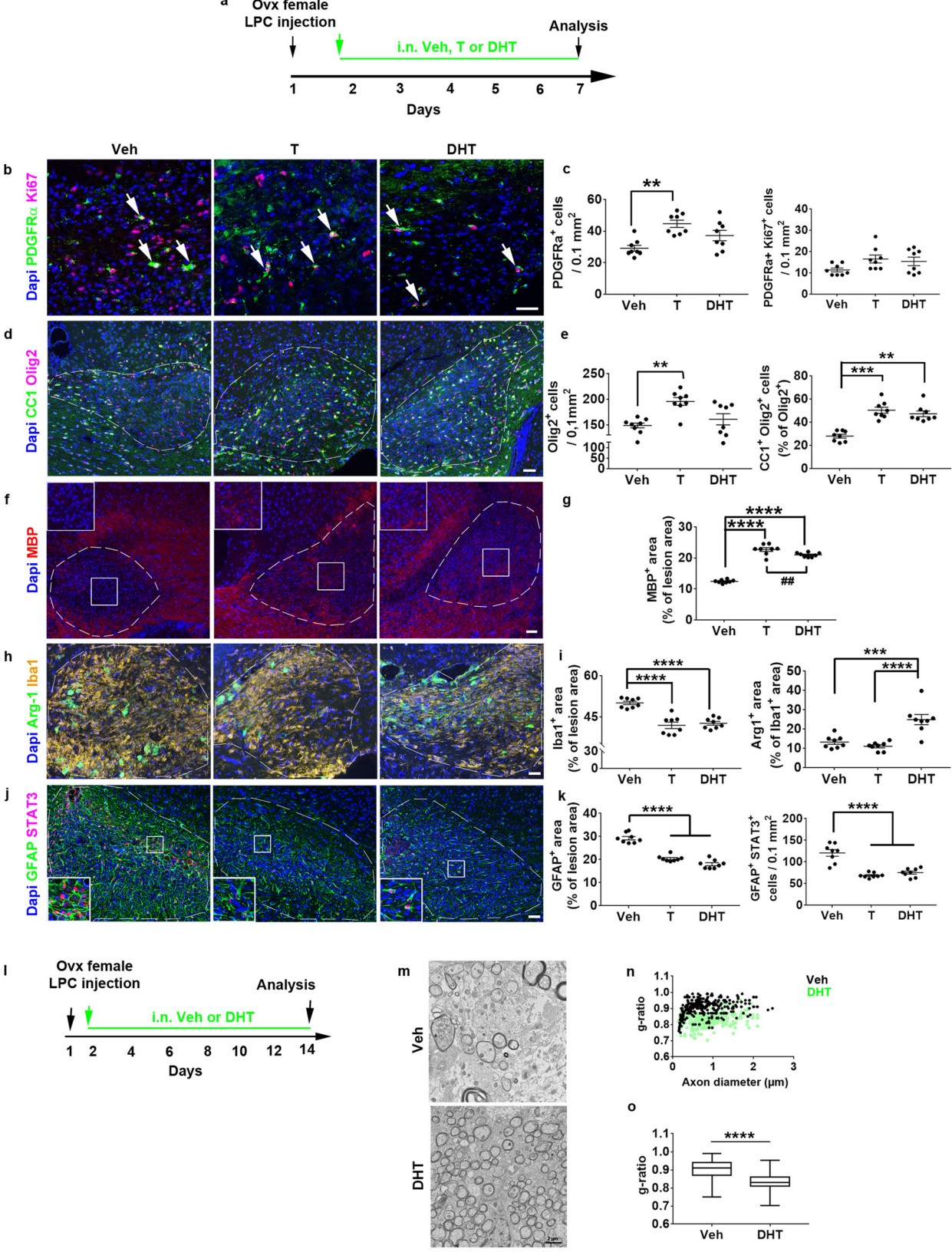

of tolerogenic versus detrimental responses in the CNS via the dendritic cells[27], we detected a lower proportion of CD11c[+] dendritic cells (likely deleterious ones; $p = 0.0157$) as well as a strong decrease in the percentage of CD44[hi] CD45RB[hi] effector/memory CD4[+] T cells expressing high membrane level of the prominent activation markers CD44 and CD45RB ($p = 0.0044$). T cell effectors including Tbet[+]/Th1

($p = 0.03$) and RoRγt[+]/Th17 ($p = 0.04$) cells, known to be deleterious in EAE, were also decreased as well as the levels of the two pro-inflammatory cytokines IFN-γ ($p = 0.005$) and TNF-α ($p = 0.031$). In the spinal cord (Fig. 8f), the decrease of the percentage of leukocytes labeled by the pan-leukocyte marker CD45 was consistent with the decrease of CD90[+] T cells ($p = 0.005$), the lower number of cellular foci

**Fig. 3 | Testosterone and DHT induce a potent regeneration of myelin in female mice. a** Scheme of the experimental protocol. **b–g** Visualization of OPC proliferation in (**b**, **c**), OPC differentiation in (**d**, **e**) and MBP expression in (**f**, **g**) evaluated 7 days after LPC injection into the corpus callosum of ovariectomized females daily treated with the drug vehicle (Veh), testosterone (T) or dihydrotestosterone (DHT). In (**b**), the white arrows indicate Ki67⁺ PDGFRα⁺ proliferating OPCs. **h–k** Immunostaining of local inflammatory cells using Iba1 and Arg-1 antibodies for the detection of the microglial population and the cell subset expressing the anti-inflammatory marker Arg-1 in (**h**, **i**) as well as GFAP and STAT3 antibodies, as markers of astrocytes and their reactive state in (**j**, **k**). The dashed lines in (**d**, **f**, **h**, **j**) indicate the lesion. The boxed area in (**f**, **j**) is magnified in the inset. Scale bars: 50 μm unless indicated. Data in (**c**, **e**, **g**, **i**, **k**) are presented as mean values ± SEM from n = 8 mice/group examined over 2 independent experiments (3–5 slices/animal). **l–o** Scheme of the experimental protocol in (**l**).

Electron microscopy analysis of the spinal cords from Vehicle and DHT-treated EAE females in (**m**) and determination of the g-ratio values plotted according to axon diameter in (**n**; 100 axons per animal, n = 3/group) as well as the mean value of g-ratios in each group in (**o**; 100 axons per animal, n = 3 mice/group). The upper, middle and lower horizontal lines of the boxplots represent the upper, median and lower quartile, respectively. Whiskers depict the smallest or largest values within 1.5-fold of the interquartile range. *P* values were calculated by using the one-way ANOVA test together with Tukey's (**c**, **i**, **k**) or Holm-Sidak's (**g**) multiple comparisons test, Kruskal-Wallis test together with Dunn's.multiple comparisons test **e**, two-tailed Mann-Whitney test (**o**). Brown-Forsythe correction was used for (**e** left, **i** left, **k** right). **\*\***$p \leq 0.01$; **\*\*\***$p \leq 0.001$; **\*\*\*\***$p \leq 0.0001$ compared to the control (Veh). ##$p = 0.0049$ compared to the indicated condition. Source data are provided as a Source Data file.

visualized at periphery of spinal cord and the lower density of infiltrated CD3-expressing T cells (Supplementary Fig. 8). Gating of the spinal cord myeloid cells indicated that percentages of the whole population of phagocytes, with as much CD45⁺ CD11b⁺ CD44⁻ microglia as CD45⁺ CD11b⁺ CD44⁺ macrophages, remained unmodified. In addition, the non-activated (resting, rMG) microglia remained the most abundant phenotype compared to the activated one (aMG). However, DHT accentuated this distribution by significantly increasing the proportion of resting cells ($p = 0.041$). As observed above in the lymph nodes, pro-inflammatory cytokine levels were also decreased, namely IL-1β ($p = 0.017$) and IFN-γ ($p = 0.0094$). As a whole, in EAE females, DHT decreases the proportion of CD4⁺ T cells, more specifically the deleterious effectors Th1 and Th17 in the lymph nodes. It reduces also the proportion of activated microglia to the advantage of resting cells consistent with the significant reduction of several pro-inflammatory cytokines in the CNS.

In male mice, DHT failed to regulate any of the immune cells or cytokine levels regulated in females suggesting an almost exclusive effect in the thymus where testosterone induces negative selection of CD4⁺ T cells[13]. Nevertheless, one exception was the increase of CD11c⁺ dendritic cells in the lymph nodes ($p = 0.047$) suggesting that unlike females, male dendritic cells may be tolerogenic (Fig. 8e). Our data also indicated notable differences in the percentages of immune cells between females and males (Supplementary Fig. 9a–c). Indeed, in the secondary lymphoid organs, CD4⁺ T cells, effector/memory CD4⁺ CD44hi CD45hi cells ($p = 0.0022$) and the CD4⁺ effectors Tbet⁺ ($p = 0.0014$) and RORγt⁺ ($p = 0.0016$) were detected in significantly higher proportions in vehicle-treated females compared to males while DHT treatment in females led these cells to reach the proportions that they display in males (Supplementary Fig. 9b). In the spinal cord, the most important sexual dimorphism regarded microglia and macrophages. Vehicle-treated females displayed as much microglia ($46 \pm 4\%$) as macrophages ($53 \pm 4\%$) whereas males displayed predominant microglia ($86 \pm 1\%$) compared to macrophages ($9 \pm 1\%$) (Supplementary Fig. 9c). In the same line, resting microglia largely predominate in females whereas similar proportions of resting- and activated- microglia could be detected in males. It should be also noted that despite comparable levels of the pro-inflammatory cytokine IL1-β in vehicle-treated EAE females and males, the level reached upon DHT-treatment was significantly lower in females ($p = 0.0008$) than in males. Altogether, these data provide evidence for major discrepancies regarding both immune cells and pro-inflammatory cytokines in EAE female and male animals. They also support a strong anti-inflammatory activity of DHT in EAE females, but not males.

### Sex-dependent regulation of local inflammatory cells in EAE spinal cord

Given the anti-inflammatory activity of DHT observed only in females and because local inflammatory cells also drive the level of neuroinflammation in the spinal cord, we visualized parenchymal immune cells and astrocytes in EAE vehicle and DHT-treated animals when the neurological scores start to significantly differ (Fig. 9a). In vehicle-treated females and males (Fig. 9b, c, i, j), Iba1-expressing microglia/macrophages were abundantly detected in the white matter at a significantly higher level in males (Supplementary Fig. 10a, $p = 0.0013$) whereas DHT treatment strongly decreased the labeling in both sexes but still maintained a higher level in males (Supplementary Fig. 10h, $p = 0.0054$). Iba1⁺ cells co-expressed the anti-inflammatory marker Arg-1 in males and females in the vehicle condition (reflecting spontaneous remyelination) with a notable scattering of the cells throughout the whole white matter in females compared to their restricted localization at the periphery of the white matter in males (Fig. 9b, c, i, j). Iba1⁺Arg-1⁺ staining in vehicle-treated females were detected at a significantly lower level than in males (Supplementary Fig. 10b; $p = 0.017$) suggesting a spontaneous response to immune-mediated demyelination depending on the sex but not on sexual hormones since animals were gonadectomized. DHT reduced Arg-1 staining in parallel with Iba1 in the whole white matter leading to only a few Arg-1⁺ spots consistent with the detection of a restricted number of demyelinated areas, as indicated by the higher level of MBP staining observed in the white matter of DHT-treated females ($p < 0.0001$) and males ($p = 0.0008$) (Fig. 9h, o) compared to the vehicle. We characterized also microglia inside each remaining lesion rather than in the whole white matter. There, Iba1 and Arg-1 expression were still higher in males than in females in the vehicle condition (Supplementary Fig. 10c, d; $p = 0.0002$ and $p = 0.0001$, respectively). However, DHT increased Arg-1 expression in female mice ($p < 0.0001$) but did not regulate it in males (Fig. 9d, e, k, l and Supplementary Fig. 10j, k) in agreement with our LPC data.

Unlike microglia, GFAP⁺ astroglia expression was significantly higher in vehicle-treated female- than male mice in both the white ($p = 0.0090$) and gray ($p = 0.0002$) matter (Supplementary Fig. 10e, f). DHT led to a comparable regulation of astrocytes, i.e., the decrease of GFAP staining in the gray matter and its increase in the white matter, but it maintained the higher GFAP staining previously observed in females under vehicle condition (Fig. 9f, g, m, n and Supplementary Fig. 10e, f, l, m). Conversely, while demyelination level was higher in females than in males under the vehicle ($p = 0.0002$), DHT led to a more potent increase of MBP staining in females ($p = 0.0392$; Supplementary Fig. 10g, n). Thus, DHT is involved in the global decrease of microglia/macrophages in the spinal cord white matter as well as in the balance of astrogliosis between the gray and white matter in both sexes. However, a sexual dimorphism exists in the response of microglia/macrophages inside the few lesions persisting under DHT treatment since DHT clearly regulates microglia phenotype towards the expression of the anti-inflammatory Arg-1 only in females consistent with a more potent remyelinating effect induced by DHT in female mice.

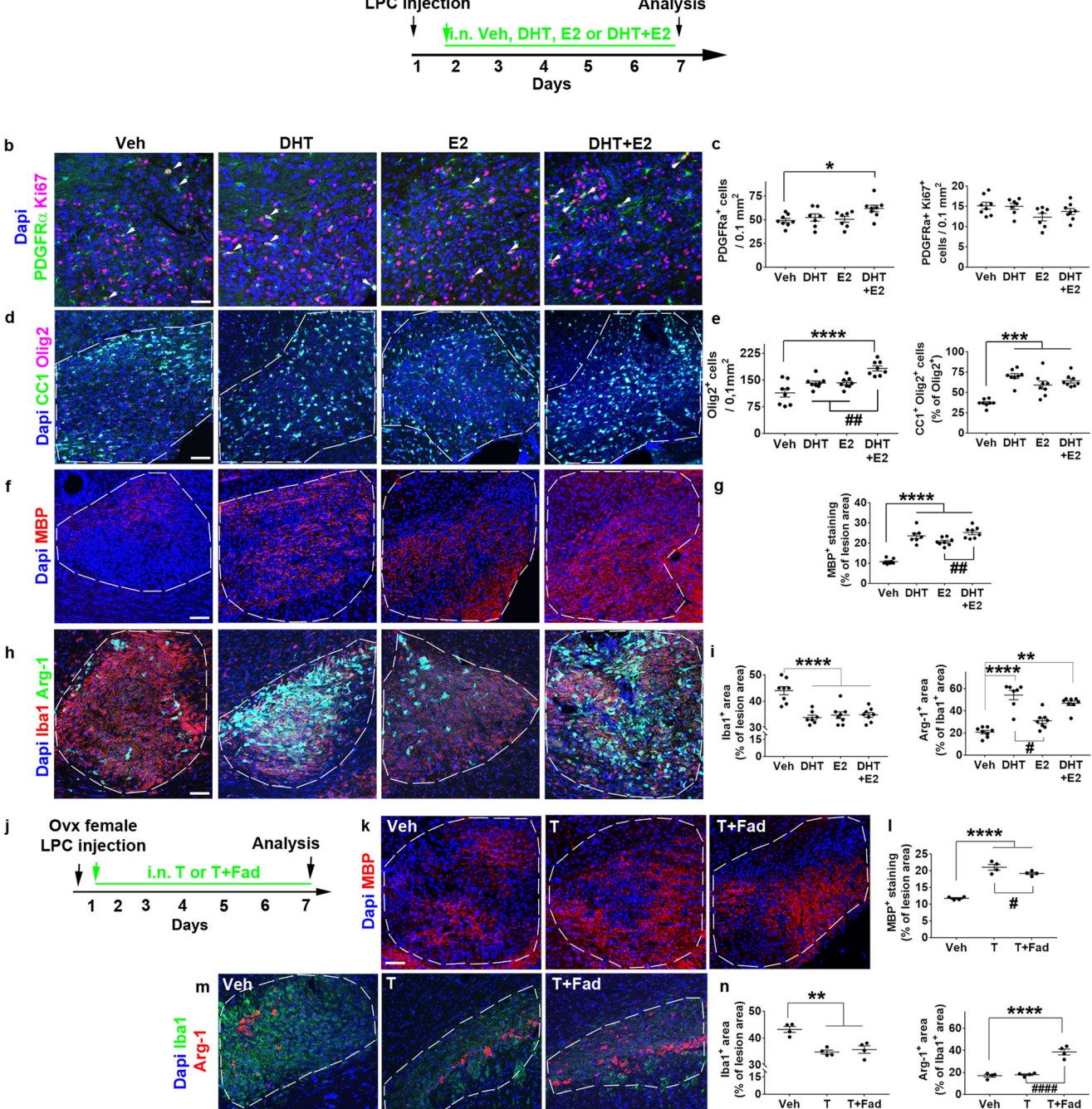

**Fig. 4 | The combination of androgens and estrogens in LPC-demyelinated female animals leads to a regeneration process more efficient than the one induced by each molecule used alone. a** Scheme of the experimental paradigm. **b–i** Visualization of OPC proliferation in **b**, **c**, OPC differentiation **d**, **e** and MBP immunostaining **f**, **g** as well as quantifications carried out 7 days after stereotaxic injection of LPC into the corpus callosum of ovariectomized female mice daily treated with the drug vehicle (Veh), dihydrotestosterone (DHT), estradiol (E2) or the combination of these molecules (DHT + E2). **h, i** Immunostaining of microglial cells by Iba1 and Arg-1 antibodies for the detection of the whole microglial population and the cell subset expressing the anti-inflammatory marker Arg-1. **j–n** Scheme of the protocol used for pharmacologically inhibiting the conversion of testosterone to estradiol by using the aromatase inhibitor, fadrozole (Fad) in (**j**). MBP in (**k, l**) and Iba1/Arg-1 in (**m, n**) immunostaining experiments were performed

and quantified on slices from the different groups of LPC-demyelinated animals. In (**b**), the white arrows indicate Ki67+ PDGFRα+ proliferating OPCs. The dashed lines in (**d, f, h, k, m**) delineate the lesion. Scale bars (μm): 50 in (**b**), 100 in (**d, f, h, k, m**). Data are presented as mean values ± SEM from $n = 8$ mice/group in (**c, e, g, i**) examined over two independent experiments and $n = 4$ mice/group in (**l, n**) examined in a single experiment (3–4 slices/per animal). *P* values were calculated by using the one-way ANOVA test together with Tukey's (**c**) or Holm-Sidak's (**e, g, l, n**) multiple comparisons test or Kruskal-Wallis test together with Dunn's multiple comparisons test (**i**). Brown-Forsythe correction was used for (**e** left, **l**). $*p \le 0.05$; $**p \le 0.01$; $***p \le 0.001$; $****p \le 0.0001$ compared to the control (Veh). $\#p \le 0.05$; $\#\#p \le 0.01$; $\#\#\#\#p \le 0.0001$ compared to the indicated condition. Source data are provided as a Source Data file.

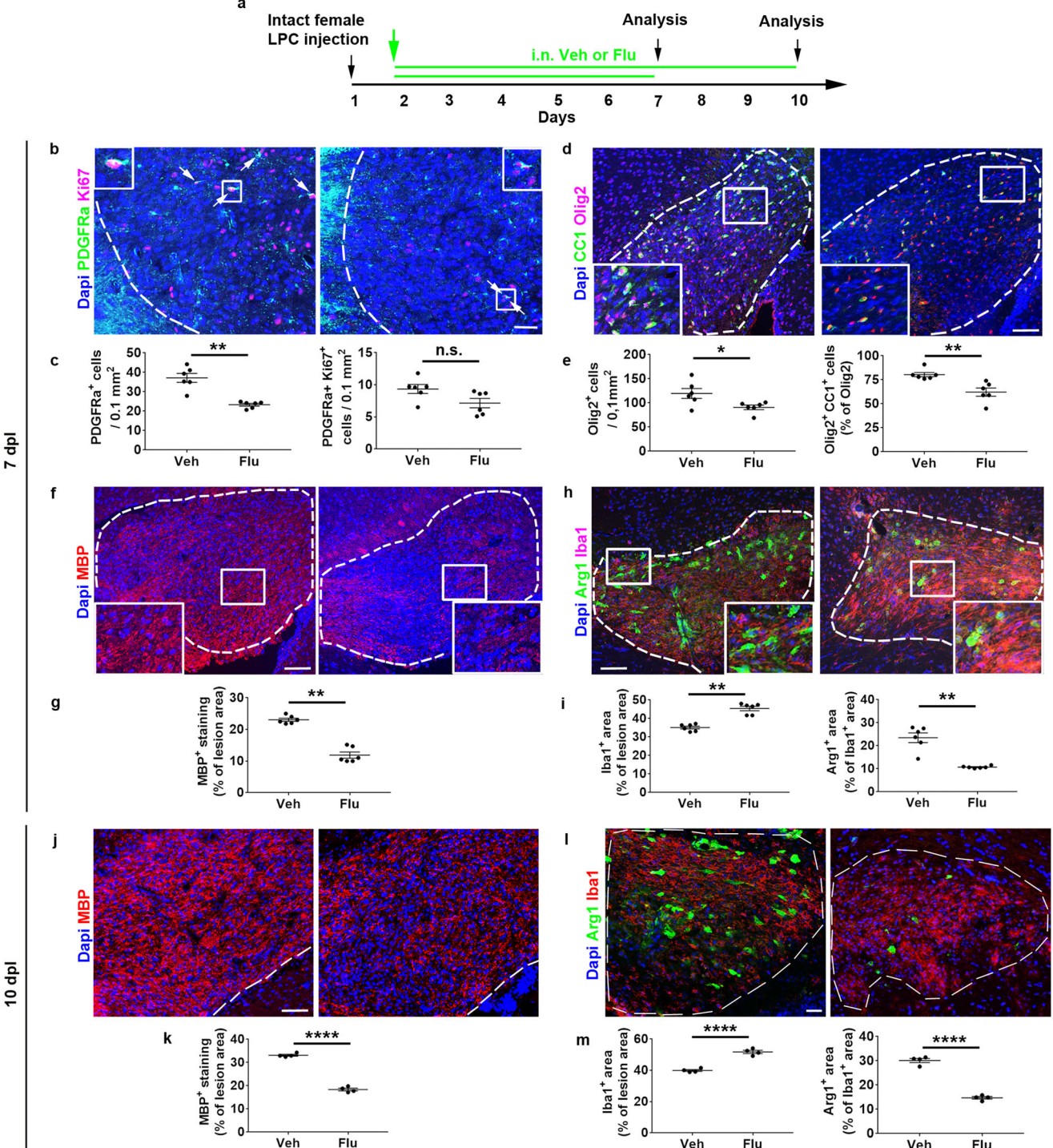

**Fig. 5 | AR blockade alters spontaneous regeneration in female mice. a** Scheme of the experimental paradigm. Visualization and quantification of OPC proliferation in (**b**, **c**), OPC differentiation in (**d**, **e**) and MBP immunostaining in (**f**, **g**) at 7 days after stereotaxic injection of LPC into the corpus callosum of ovariectomized female mice daily treated with the drug vehicle (Veh) or the AR antagonist flutamide (Flu). In (**b**), the white arrows indicate Ki67⁺ PDGFRα⁺ proliferating OPCs.
**h, i** Immunostaining of microglial cells by using Iba1 and Arg-1 antibodies for the detection of the whole microglial population and the cell subset expressing the anti-inflammatory marker Arg-1. The dashed lines delineate the lesions. The boxed areas are magnified in the insets. (**j**–**m**) Visualization and quantification of MBP in (**j**, **k**) and Iba1/Arg-1 in (**l**, **m**) immunostaining at 10 dpl. Scale bars (μm): 50 in (**b**, **j**), 100 in (**d**, **f**, **h**, **l**). Data are presented as mean values ± SEM from $n = 6$ mice/group in (**c**, **e**, **g**, **i**) and $n = 4$ mice/group in (**k**, **m**) (3–4 slices/per animal). $P$ values were calculated by using the unpaired two-tailed t-test (**c**, **k**, **m**) or two-tailed Mann-Whitney (**e**, **g**, **i**). Welch's correction was used for (**c** left, **i** right). *$p \leq 0.05$; **$p \leq 0.01$; ***$p \leq 0.001$; ****$p \leq 0.0001$ compared to the control (Veh); n.s., non-significant. Source data are provided as a Source Data file.

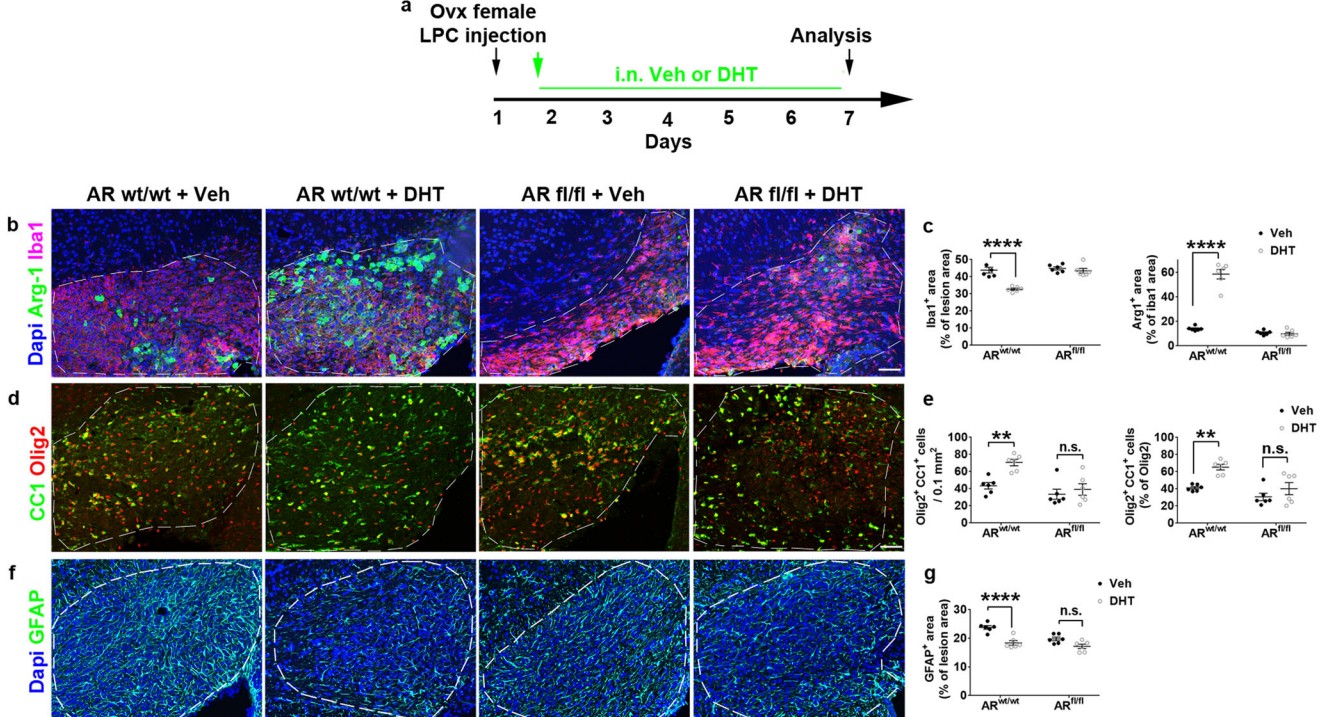

**Fig. 6 | Microglial AR is required for DHT-induced control of microglia response to demyelination. a** Scheme of the experimental paradigm. **b**–**g** Visualization and quantification of Arg-1 expression in Iba1-expressing microglial cells in **b**, **c**, OPC differentiation in **d**, **e** and GFAP immunostaining of astrocytes in **f**, **g** at 7 days after stereotaxic injection of LPC into the corpus callosum of ovariectomized female mice expressing (AR wt/wt) or not (AR fl/fl) AR in microglia and treated with the drug vehicle (Veh) or DHT. The dashed lines delineate the lesions. Scale bars: 50 μm. Data are presented as mean values ± SEM from $n = 6$ mice/condition examined in two independent experiments (3 slices/per animal). $P$ values (**c**, **e**, **g**,) were calculated by using the two-way ANOVA test together with Tukey's multiple comparisons test. **p = 0.0069 (**e** left), $p = 0.0061$ (**e** right); ****$p < 0.0001$ **c**, $p = 0.0001$ **g**; n.s. not significant. Source data are provided as a Source Data file.

## Sexually dimorphic molecular mechanisms induced by DHT in EAE animals

The sexual dimorphism mentioned above led us to investigate the molecular mechanisms putatively involved. We performed a transcriptomic comparison, by bulk RNA sequencing (RNA-Seq) of the spinal cords derived from EAE female and male mice treated or not (control) with DHT (Fig. 10a). After dataset normalization (Supplementary Fig. 11), we found that both DHT-treated females and males were clearly separate from their respective controls by principal component and clustering analyses indicating a clear effect of DTH treatment (Fig. 10b, c and Supplementary Fig. 12a–d). A large number of differentially expressed genes (DEGs) was found in DHT-treated compared to control, both in females (3285 up- and 4185 down-regulated) and males (2061 up- and 1720 down-regulated) by using stringent statistical criteria (FDR < 0.05) indicating strong gene expression changes upon DHT treatment in both sexes Supplementary Fig. 11b; Supplementary Dataset 1). To assess the impact of DTH-treatment in oligodendrogenesis and myelination, we used OligoScore (https://oligoscore.icm-institute.org/), a resource using a knowledge-driven scoring procedure for gene sets involved in oligodendrogenesis and (re)myelination, as described in Methods. Given that EAE model is characterized by the co-existence of demyelinating and remyelinating plaques (as in MS), genes either promoting or inhibiting the oligodendroglial processes could be detected among the DEGs (Supplementary Fig. 13a–j; Supplementary Datasets 2–5). This analysis indicated that myelination was the main process impacted by DHT treatment, both in females and males (Fig. 10d, e), whereas the global changes on other processes of oligodendrogenesis remained rather faint, except on proliferation for which gene changes resulted in an inhibitory effect specifically in females. Thus, DTH treatment mostly results in the promotion of myelin biogenesis in both sexes with

however a sex-dependent signature since besides the regulation of 25 shared genes, DHT specifically controlled 69 and 25 genes in female and male mice, respectively (Supplementary Fig. 13k, l).

Gene ontology (GO) analysis of DEGs, showed that in both sexes, upon DTH-treatment, the top processes involving up-regulated genes were enriched in terms promoting neuronal activity, confirming our above-mentioned immunofluorescence and functional analyses. However, the top processes enriched in down-regulated genes showed sexual dimorphism, in females related to the immune system and inflammation and in males related to lipid metabolic processes (Fig. 10f–i, Supplementary Dataset 6 and Supplementary Fig. 12e). These molecular features are thus fully consistent with the remyelinating effects and the functional improvement of female and male animals induced by DHT and also corroborate the differential capacity of DHT to regulate the inflammatory process in female and male demyelinated tissues.

Given the sex-dependent discrepancies identified above regarding the local inflammatory cells, we further analyzed the RNA-Seq data focusing on these cells. We used gene sets characteristic of the main subpopulations of microglia constituting microglia diversity during aging or neurodegeneration, including homeostatic, activated, disease associated (DAM), and white matter associated (WAM) microglia[28–35]. Two subsets of genes were down-regulated in both DHT-treated females and males compared to their respective controls either without significant difference between female and male controls (including homeostatic genes *Axl, Cd68, Csf1r, Cx3cr1* and *Tmem119*; DAM genes *Axl, Trem2, Ctsl, Fth1,* and *Lyz2*; WAM genes *Anxa5, C1qb, Cd63, Cd74, Cst7, Ctsz, Fam20c, Fth1, Ftl1, H2-D1 and Lyz2*) or with significant difference between female and male controls (including homeostatic genes *ApoE, C1qc, Mertk, Rxra* and *Trem2*; DAM genes *ApoE, Ctsb,Ctsd, Lpl, Timp2* and *Trem2*; WAM genes *ApoE, Atp6v0c* and *Ctsb*;

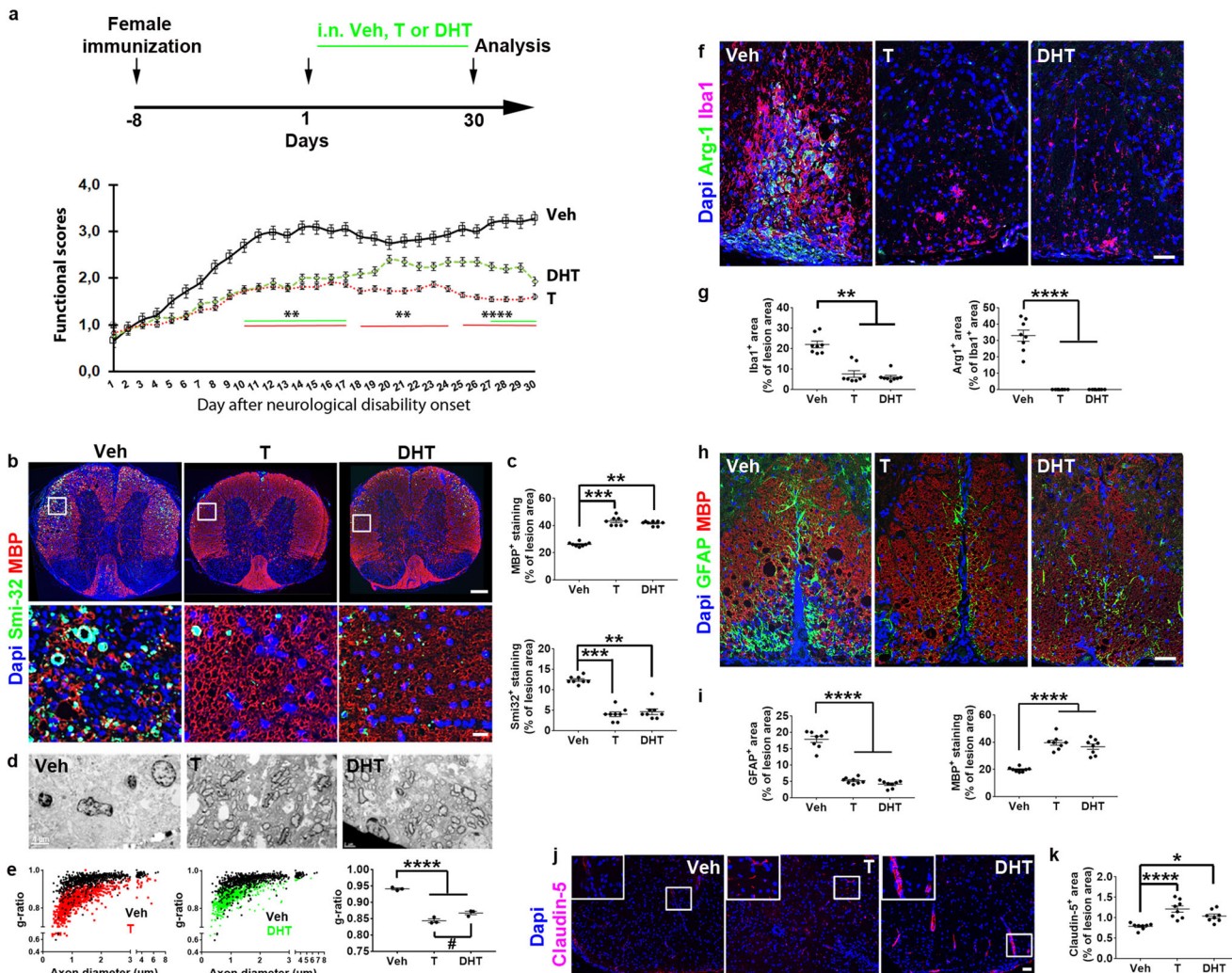

**Fig. 7 | Therapeutic administration of androgens mitigates the course of EAE in female mice. a** Functional scores derived from EAE ovariectomized female mice treated with the drug vehicle (Veh), testosterone (T) or dihydrotestosterone (DHT) at onset of the first neurological symptoms (day 1) for 30 days (two-way ANOVA: treatment: F(2, 1307) = 298.2, p < 0.0001; time: F(29, 1307) = 33.23, p < 0.0001). **b, c** Smi-32 and MBP IHF performed on spinal cord slices derived from animals of each group. The boxed areas are magnified in the bottom panels. Determination of the fluorescent area is shown in the histograms on the right. **d, e** Electron microscopy analysis of the spinal cords and determination of the g-ratios plotted according to axon diameter or represented by their mean value. **f, g** Visualization and quantification of microglia immunostained with Iba1 (red) and Arg-1 (green) as markers of the whole microglia population and the cell subset that express the anti-inflammatory molecule Arg-1, respectively. **h, i** Visualization and quantification of astrocytes by using GFAP. **j, k** Detection of the tight junction protein Claudin-5 in each animal group. The boxed areas are magnified in the insets. Data are the mean ± SEM from n = 12 animals/condition in (**a**) or from n = 8 animals/condition in **g, i, k** examined in a single experiment (3–4 slices/per animal). 600–900 axons from n = 3 mice in (**e**) were evaluated. P values were calculated by using the Kruskal-Wallis test together with Dunn's.multiple comparisons test (**c, g**) or one-way ANOVA test together with Tukey's (**e, k**) or Holm-Sidak's **i** multiple comparisons test. Brown-Forsythe correction was used in (**k**). *p < 0.05; **p < 0.01; ***p < 0.001 ****p < 0.0001 versus the control condition. Scale bars (μm): 200 in (**b** top), 50 **f, h, j**, 25 (**b** bottom). Source data are provided as a Source Data file.

---

Supplementary Datasets 7 and 9) in agreement with the reduction of the inflammatory foci and demyelinated areas previously observed in spinal cord slices. However, another gene subset (34 genes out of 103) was exclusively down-regulated in DHT-treated females without significant differences between female and male controls (including microglia genes *Cd33, Fcgr2b, Tlr4, Tnf*; microglia DAM genes *B2m, Ccl6,Cd9,Clec7a, Csf1, Csf2, Itgax*; microglia WAM genes *Anxa2, C1qb, Capg, Cd52, Crip1, Ctss, Cybb, H2-K1, Ifitm3, Lgals1, Lgals3*, Spp*1, Tspo, Vim*; microglia activated genes *Il1b, Rpl14, Rpl21, Rpl35, Rpsa, Tmsb4x*; microglia remyelination genes *Ank, Fn1, Psat1*; pink highlighted in Supplementary Dataset 9; tab DEGs Microglia). Several out of these genes are notable such as the genes encoding TLR4 whose activation leads to the production of pro-inflammatory cytokines[36] or the pro-inflammatory cytokine TNF-α, but also *Csf2* and *Il1b*, which encode the well-known pro-inflammatory GM-CSF and IL-1β, respectively.

Quantitative RT-PCR amplification confirmed that *Tnf* and *Csf2* were significantly down-regulated in DHT-treated females (Supplementary Fig. 14). To exclude bias putatively related to changes in microglia cell numbers between conditions, we first took advantage of tools available in the field by using a scRNA-Seq dataset from mouse EAE model[37] (GSE113973). Resulting dotplots revealed that 21 genes out of 31 displayed a high average and percent expression mostly in EAE microglia (Supplementary Fig. 15). By combining all microglial gene sets (105 genes), we performed gene set enrichment analysis (GSEA) with these genes ordered by their changes in expression either in female or male comparisons (DHT-treated versus non-treated). In line with previous results, this GSEA analysis showed large enrichment of many gene sets in DHT-treated females but almost none in males, with many of the suppressed gene sets (genes being downregulated) related to immune and inflammatory processes, including 'lymphocyte mediated

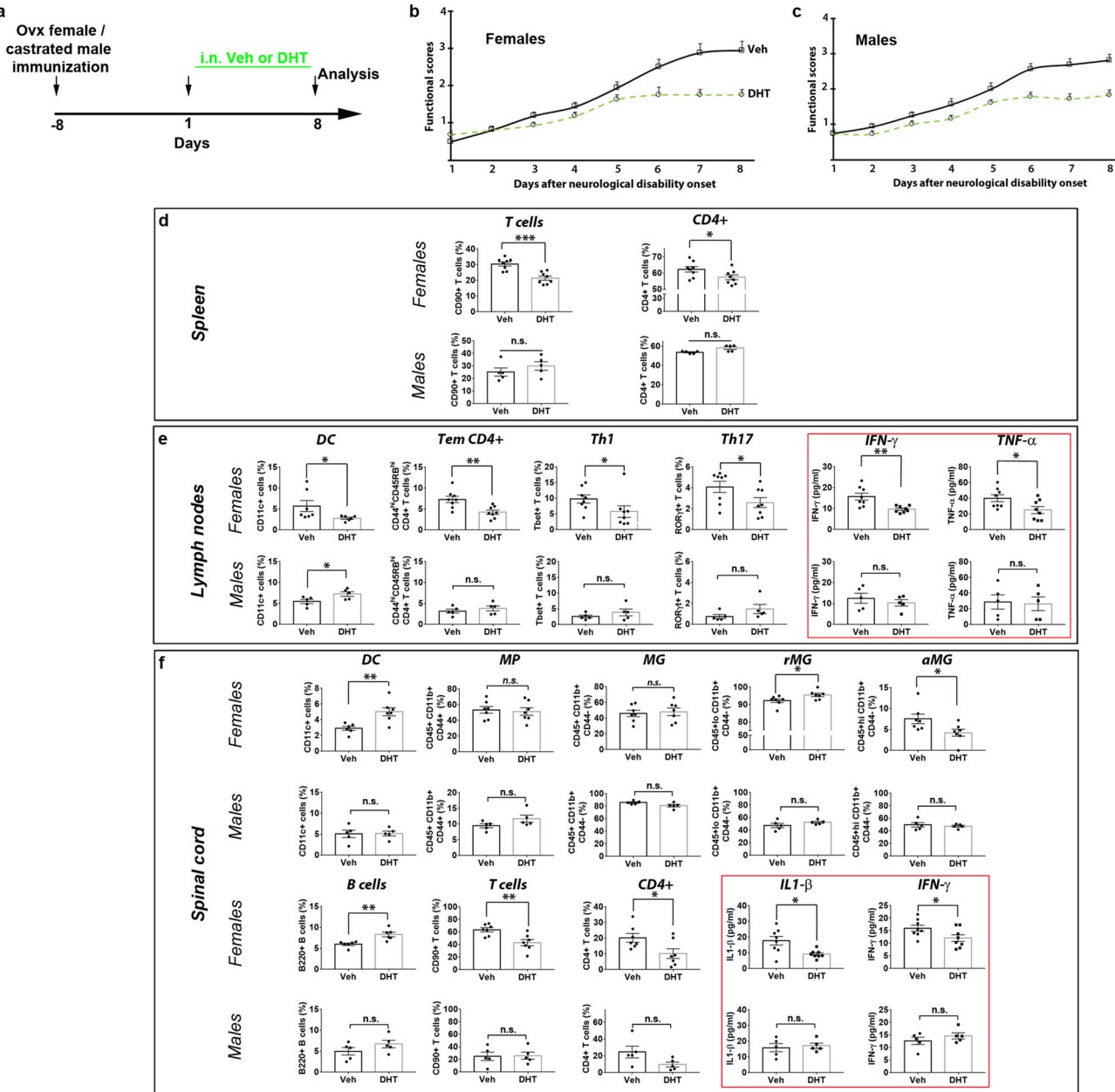

**Fig. 8 | The immune response triggered by DHT in EAE animals is strikingly different in female compared to male mice. a** Scheme of the experimental protocol. **b**, **c** Scoring of neurological disabilities in EAE female and male mice (*n* = 8/ group) daily treated with DHT or the drug vehicle (Veh) at onset of neurological symptoms (day 1) for 8 days. **d**–**f** The spleen (*n* = 8 females, 5 males) in **d**, lymph nodes in **e** and spinal cord in **f** (*n* = 7 females, 5 males) from each animal group examined in a single experiment were harvested in order to perform flow cytometry analysis and dosage of cytokines. Data are presented as mean values ± SEM. The gating strategies for flow cytometry analysis are shown in Supplementary

Figs. 4–7. Only the cell types regulated by DHT are shown. In the lymphoid organs and spinal cord, immune cell types are expressed in percentage of all cells and CD45⁺ leukocytes, respectively. Similarly, a panel of 9 cytokines has been assessed (as described in Methods). Only cytokines regulated by DHT are shown in red boxes. *P* values (**d**, **e**, **f**) were calculated by using the unpaired two-tailed t-test or Mann-Whitney tests. Welch's correction was used for IFNγ **e** and IL1-β **f** in females. *$p \leq 0.05$; **$p \leq 0.01$; ***$p \leq 0.001$ compared to the control (Veh); ns non-significant. Source data are provided as a Source Data file.

immunity', 'response to stress', 'defense response', 'immune system process', 'immune response', and 'cytokine production' (Supplementary Fig. 16; Supplementary Dataset 10). Therefore, these results agree with those obtained by immunofluorescence analyses supporting a strong anti-inflammatory activity of DHT in EAE female mice, but not males. To try to exclude bias putatively related to changes in cell numbers between conditions, we used Cibersortx, a machine learning method to determine cell type abundance and expression from bulk tissues[38], together with a single cell RNA-Seq dataset from mouse EAE model[37] (GSE113973). This deconvolution of our bulk-RNA-Seq datasets

suggested that while microglial clusters did not change in proportions upon DHT-treatment in males, DHT-treated females presented some changes in microglial clusters and EAE immune-OL/OPC clusters[37] (Supplementary Dataset 11), likely due to the abovementioned dysregulation of microglial/inflammatory genes. Indeed, 21 genes out of 31 downregulated genes only in DHT-treated females are expressed in the EAE microglial cells from this scRNA-Seq dataset (Supplementary Fig. 15). Altogether these results are in agreement with our flow cytometry data, which indicated that DHT did not modify the proportion of microglia/macrophages neither in female nor in male spinal cords

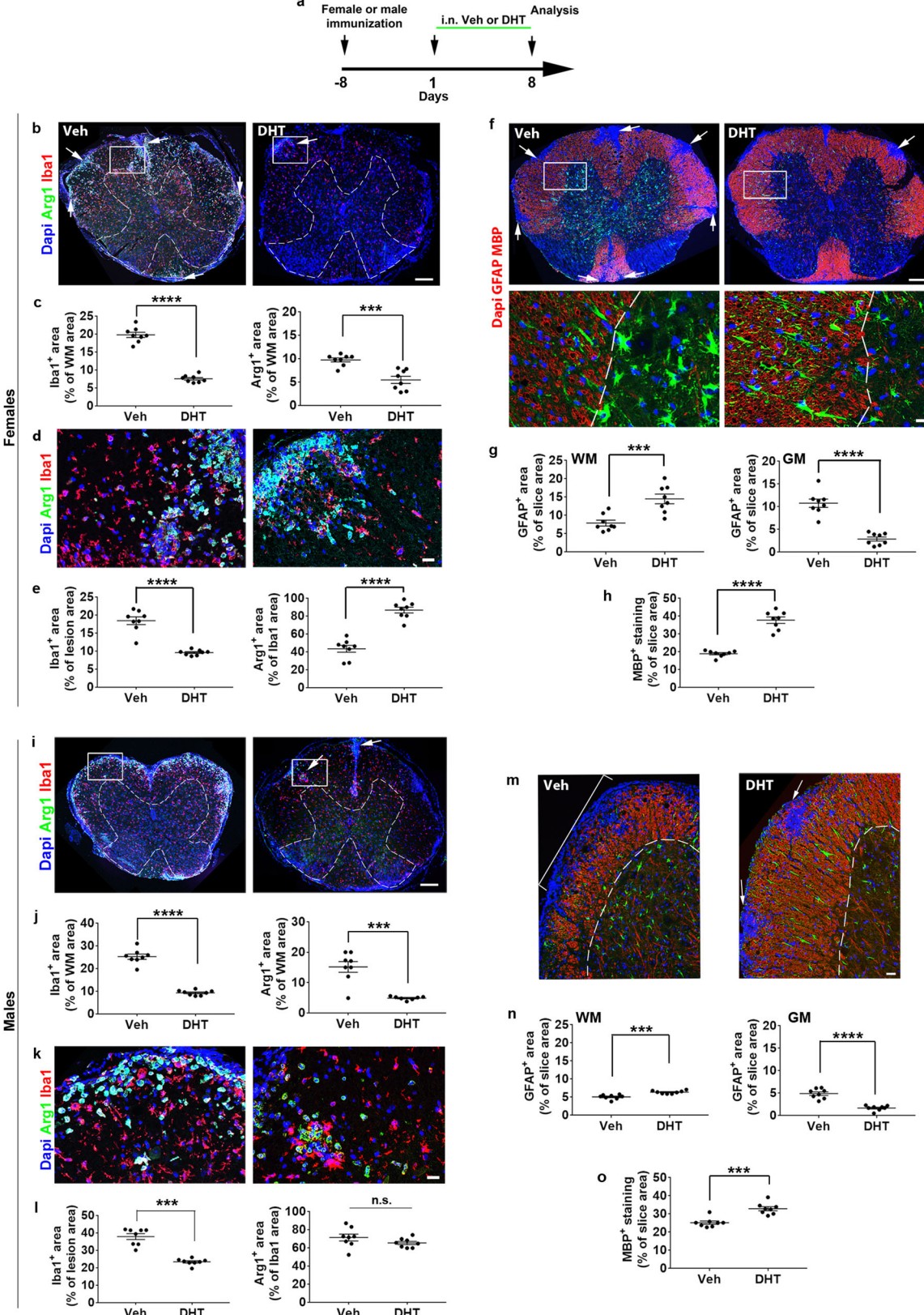

compared to their controls (Fig. 8f) and they support the existence of true molecular differences in the effects of DHT in female compared to male mice.

Similarly, we used sets of genes previously implicated in the characterization of different astrocyte subsets[39,40]. Among the 165 DEGs related to astrocytes, 62 were deregulated in both DHT-treated females and males. Most importantly, 83 were deregulated by DHT exclusively in females whereas only 20 were deregulated exclusively in males compared to their own controls. Additionally, most genes (21 out of 26) known to identify activated astrocytes, pro- and anti-inflammatory astroglial phenotypes were exclusively deregulated in DHT-treated females (Supplementary Datasets 8 and 9) further

**Fig. 9 | DHT controls differently the local inflammatory cells in EAE female and male mice. a** Scheme of the experimental protocol. Immunostaining of microglial and astroglial cells in the spinal cord from vehicle or DHT-treated female in (**b–h**) or male in (**i–o**) mice. **b, c** Visualization and quantification of microglia in the whole white matter of vehicle-treated females indicate numerous spots of Arg-1[+] cells extending deeply into the white matter (white arrows in **b**) strongly reduced under DHT treatment. **d, e** Visualization and quantification of Iba1 and Arg-1 staining at the level of an individual lesion indicating that Iba1 staining is still decreased whereas Arg-1[+] area is significantly higher under DHT treatment. **f, g** GFAP[+] astrogliosis is shown in whole spinal cord slices co-labeled by MBP antibody aimed at visualizing myelin. Numerous spots of demyelinated tissue are shown (white arrows) in the vehicle- compared to the DHT condition. Magnifications of the boxed areas show that DHT treatment is accompanied by the decrease of GFAP staining in the gray

matter (GM) and conversely its increase in the white matter (WM). **h** Quantification of MBP[+] area in the white matter. **i–l** Visualization and quantification of Iba1 and Arg-1 staining in the spinal cord from male in the whole white matter in (**i, j**) and at the level of individual lesions in **k, l. m–o** Visualization of GFAP and MBP staining in **m**. Quantification of GFAP[+] fluorescence in the white (WM) and gray (GM) matter in **n** and of MBP in the white matter in **o**. The boxed areas in **b, i** are magnified in **d, k**. Data are presented as mean values ± SEM from $n = 8$ mice/group examined in a single experiment (3–4 slices/per animal). $P$ values (**c, e, g, h, j, l, n, o**) were calculated by using the unpaired two-tailed t-test or Mann-Whitney test. Welch's correction was used for (**e** left, **j, h**). **$*p \leq 0.01$; ***$p \leq 0.001$; ****$p \leq 0.0001$; n.s, non-significant. Scale bars (μm): 200 in (**b, f** top, **i**), 50 in **m**, 25 in (**d, f** bottom, **k**). Source data are provided as a Source Data file.

suggesting the ability of DHT to molecularly control astrogliosis in female mice in a specifically different way compared to males.

## Discussion

Since testosterone is well-known to exert its effects on target cells via AR or ERs after its conversion to estradiol, beneficial effects of estrogens on the course of EAE[10] in males have not been so surprising when reported. In contrast, testosterone involvement in repairing processes in females is more unexpected. However, circulating levels of testosterone in women are far from being insignificant despite a 10–20-fold lower level than in men[41–43]. Additionally, exacerbation of demyelinating episodes during the postpartum were associated with a decrease in both female and male hormones[44]. Finally, low testosterone levels were previously associated with increased brain lesions and clinical disability in women with MS[45].

Our present data are consistent with those findings as shown by the graphical abstract (Supplementary Fig. 18). Indeed, besides the well-recognized expression of AR in cortical neurons, AR transcripts and protein are also highly expressed in demyelinated areas from female patients and mice. As previously shown in Sertoli cells[46], both nuclear and non-nuclear AR might participate in androgen signaling during remyelination in females, an observation consistent with the well characterized AR translocation from the cytoplasm to the cell nucleus upon ligand binding[47,48], and the existence of a nucleotide sequence suggesting also AR translocation to the membrane[49]. Although all cell types express AR in the female mouse lesions, AR expression in microglia is widely predominant and functionally critical since conditional AR mutants confer to androgens a direct role in the control of female microglia by promoting a pro-regenerative response to demyelination resulting in OPC differentiation and astrogliosis decrease. However, our data do not allow to exclude that the high AR expression in female neurons might also participate in DHT-induced beneficial response. Indeed, DHT up-regulates genes related to synaptic function and thus promotes neuronal electrical activity, a well-known inducer of myelination[50,51]. Moreover, the serine/threonine kinase mTor that we found exclusively up-regulated by DHT in females was previously proposed to maintain the non-reactive state of astrocytes in the cerebral cortex[52].

In male patients and mice, the quite undetectable *AR* expression in the demyelinated lesion supports a fully different remyelinating activity independent of microglia but putatively dependent on neuronal AR expression. These observations are consistent with our previous data showing that microglia is not implicated in AR-mediated remyelinating effect of testosterone[14,20] and with the ability of DHT to up-regulate in male mice (like in females) genes related to purine nucleoside biosynthetic processes known to be required before extra-synaptic release of adenosine, a critical mediator for triggering myelination[53]. Similarly, neuronal AR activation might also be involved in testosterone-induced increase of astrogliosis previously reported to be AR-dependent in male animals[15,20] since in response to neuron-derived active compounds, astrocytes display specific molecular

signatures leading to mechanisms of astrocyte-neuron communication including those implicated in migration[54].

Our work also uncovers other major sex-dependent molecular discrepancies regarding DHT effects upon CNS demyelination. Thus, only DHT-treated female mice are able to down-regulate genes known to identify activated-, pro- or anti-inflammatory astrocytes. One of these genes encodes the critical regulator of astrogliosis STAT3[55] whose down-regulation is consistent with the decrease of the number of GFAP[+]STAT3[+] reactive astrocytes induced by DHT in the LPC model in female animals, but not in males (present data and[20]). Moreover, only female mice respond to DHT by promoting clear anti-inflammatory effects including the reduction of the pro-inflammatory cytokines IL-1β/IFN-γ and the down-regulation of the genes encoding TLR4 and TNF-α, which are involved in the pro-inflammatory phenotype of reactive astrocytes[40]. The critical crosstalk existing between astrocytes, microglia and oligodendrocytes during remyelination[56] might thus be sex-dependent and likely contribute to the differential gene profiles characterizing female and male mouse remyelination as shown for instance for *mTor*. Indeed, known to regulate the initiation of myelination[57,58] mTor appears here to be one of the molecular targets of AR-dependent remyelinating effects of DHT in female mice but not in males.

Finally, if most genes down-regulated by DHT in female mice are related to inflammation, those predominantly down-regulated in males are related to lipid metabolic processes. However, this discrepancy does not mean that lipid metabolism is not controlled by DHT in females as evidenced by the comparable down-regulation of genes encoding proteins known to link lipid metabolism and remyelination[59]. This is true for RxRα, involved in the phagocytic removal of myelin debris deleterious for OPC differentiation[60] and for ApoE, impeding cholesterol accumulation upon myelin debris phagocytosis to prevent phagocytes from becoming inefficient[59]. Though androgens protect against autoimmunity by primarily acting at the level of the thymus in both males and females[13], the decrease of the encephalitogenic CD4[+] T cells Th1 and Th17[61] exclusively detected in females suggest a higher level of complexity in the peripheral immune response of the latter. The curative administration of DHT nevertheless interrupts the encephalitogenic process and decreases the need for high spontaneous remyelination in agreement with the down-regulation of *RxRα/ApoE* in both sexes. However, if we consider the anti-inflammatory effects of DHT occurring exclusively in females together with the fact that the peripheral immune response is thought to drive the relapsing-remitting form of MS whereas compartmentalized CNS immune reactions may be more involved in the progression of MS[62], we may wonder if the well-known worse prognosis of MS in men compared to women might be in part related to improper anti-inflammatory response in the demyelinated male CNS. Thus, besides suggesting the use of appropriate doses of androgens in demyelinated females, this work also uncovers the need for considering the sex-specific AR-mediated control of microglia/macrophage response to demyelination in the therapeutic management of MS.

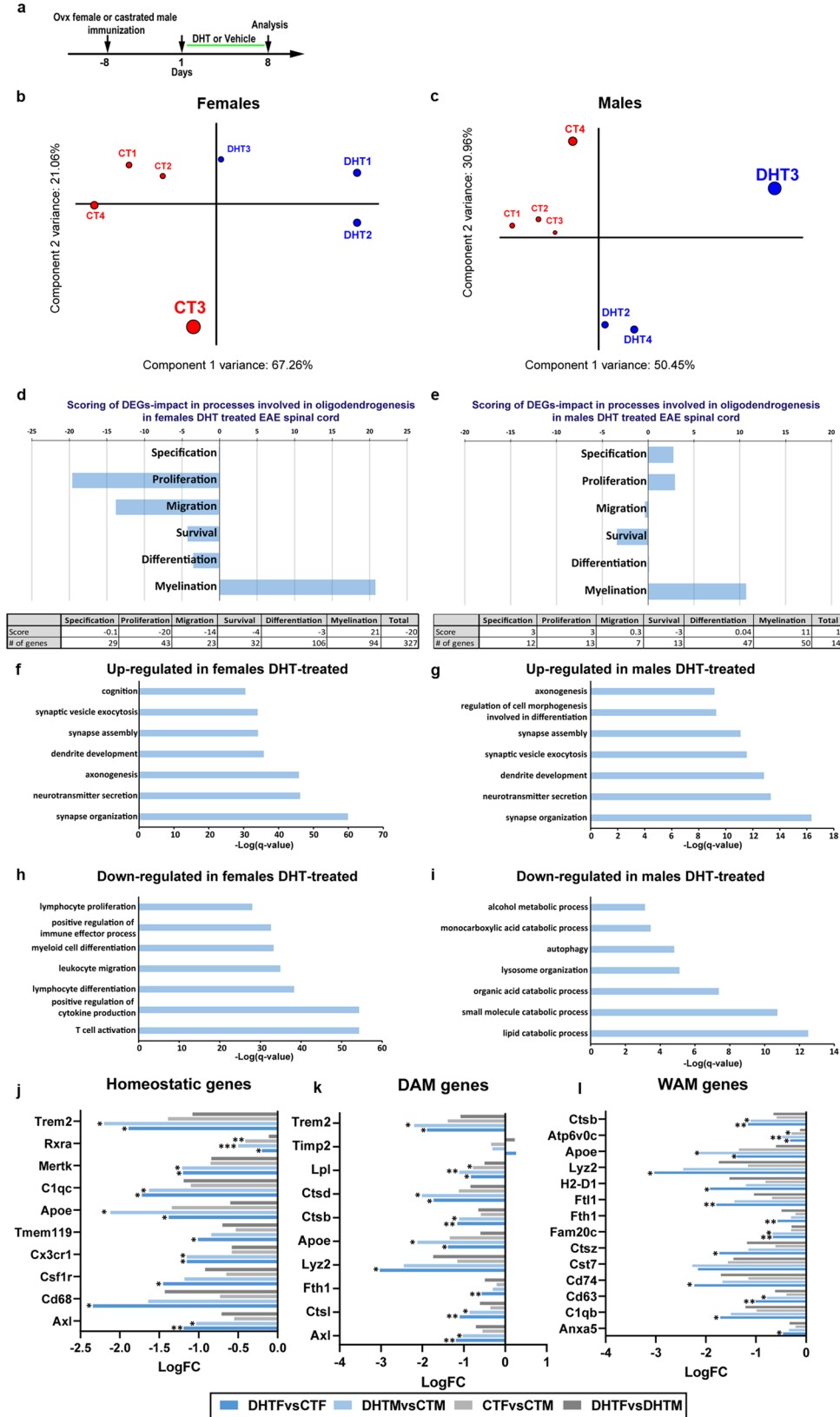

## Methods

### Animals

All procedures were performed according to the European Communities Council Directive (86/806/EEC) for the care and use of laboratory animals and were approved by the Regional Ethics Committee

CEEA26, Ministère de l'Education Nationale, de l'Enseignement et de la Recherche. Wild-type intact or gonadectomized C57Bl/6 male and female mice were purchased at the age of 8 to 12 weeks from Janvier Labs Breeding Center (France). $AR^{fl/fl}$[63] were maintained on a C57Bl/6 background. The mouse strain CX3CR1tm2.1(Cre/ERT2) (thereafter

**Fig. 10 | RNA-Seq analysis of the spinal cord derived from EAE mice therapeutically treated with DHT reveal major differences between female and male animals. a** Scheme of the experimental protocol. **b, c** PCA plot of two first components with their contribution to the variance depicturing clear differences between DHT-treated (DHT, n = 3) and control (CT, n = 4) samples in the first PCA component, which contributes for more than half of the variance of the experiment, in females in (**b**) and in males in (**c**) examined in two independent experiments. The size of the sample name and the circle indicate the relative contribution to the total variance. **d, e** Barplots and tables showing the contribution of oligodendroglial curated DEGs genes to promote (positive) or inhibit (negative) each process of oligodendrogenesis in females in **d** and in males in **e**. Note that DHT mainly promotes (re)myelination. **f, g** Dotplot representing the top 7 biological processes enriched in up-regulated genes in DHT-treated females in **f** and

DHT-treated males in **g** compared to their respective controls, showing similar up-regulation of synaptic and neuronal associated processes in both sexes. **h, i** Dotplot representing the top 7 biological processes enriched in down-regulated genes in DHT-treated females in **h** and DHT-treated males in **i**. Note that while down-regulated genes are implicated in immune processes in females in **h**, they are implicated in catabolism in males in **i**. **j–l** Histograms visualizing the deregulation of genes characterizing homeostatic in **j**, Disease-Associated in **k** and White matter-Associated in **l** microglia by DHT in EAE females or males. Multiple testing correction aimed at controling the false discovery rate (FDR, p-adjust) was performed by using the Benjamini-Hochberg method **f–i**. Fisher test was used as well as multiple testing correction aimed at controling the false discovery rate (FDR, p-adjust) performed by using the Benjamini–Hochberg method **j–l**. *FDR < 0.05; **FDR < 0.01; ***FDR < 0.001.

called CX3CR1CreER-YFP) expressing the YFP reporter under the promoter of the chemokine receptor CX3CR1[64,65] was provided by Jackson Laboratory. Animals prone to receive hormones were gonadectomized in order to exclude the confounding effects of endogenous gonadal steroid hormones and after validation that *AR* upregulation was still detected in ovariectomized animals and not related to the mechanical injury induced by the injection (Supplementary Fig. 17). All animals were housed in standard conditions: ambient temperature at 20 °C, relative humidity at 45–65%, 12 h light-dark cycle with food and water *ad libitum*.

### Drugs
Testosterone, dihydrotestosterone, estradiol, flutamide and fadrozole were provided by Sigma-Aldrich (France). Testosterone (0.20 mg/day under a volume of 2.5 μl in each nostril), dihydrotestosterone (0.04 mg/day under a volume of 2.5 μl in each nostril due to its much higher potency in transactivating AR target genes than testosterone[66]) and estradiol (0.0375 mg/day under a volume of 2.5 μl in each nostril) were administered daily per the intranasal route via a proprietary oleogel (MetP Pharma AG, Emmetten, Switzerland)[67]. Flutamide (20 mg/kg) and fadrozole (250 μg/kg) were administered daily per gavage. Tamoxifen (Sigma-Alrich; 30 mg/ml) was dissolved in corn oil (Sigma-Aldrich) and administered by gavage (3 mg/day for 5 days) 2 weeks before inclusion of the animals in any experimental protocol.

### LPC-induced focal demyelination
Demyelinating lesions were induced unilaterally by stereotaxic injections of 1.5 μl of a solution containing LPC 1% (Sigma-Aldrich) into the right corpus callosum at the following coordinates (to the bregma): anteroposterior (AP) +1 mm, lateral +1 mm, dorsoventral (DV) −2.2 mm for brain analyses performed at 7, 10 or 14 days postlesion (dpl) after animal perfusion with PFA 4%. The tissue was post-fixed for 4 hrs in fresh 4% PFA solution before being cryopreserved in 30% sucrose, frozen in liquid nitrogen and cryostat sectioned (14 μm). 7dpl was selected as the suitable time when the process of spontaneous remyelination is ongoing and corresponds to the end of OPC recruitment and the beginning of their differentiation into oligodendrocytes[20,68]. 10 and 14 dpl were used for MBP immunostaining and electron microscopy analysis of myelin, respectively.

### Autoimmune experimental encephalomyelitis
Ovariectomized females or castrated male mice at age of 9–10 weeks were maintained for one week for acclimatization prior to EAE. The pathology was induced by subcutaneous injection of an emulsion of MOG$_{35-55}$ peptide in complete Freund's adjuvant[69]. The mice that developed EAE were randomly assigned into vehicle, testosterone or DHT treatment in order to constitute groups with similar time of EAE onset and similar onset scores (n = 8–12 animals per group). The mice were scored blindly once a day starting at Day 8 post-immunization (onset of neurological disabilities for all animals) until Day 16 or 38 (as indicated) according to the following scale: 0.0 = no obvious changes

in motor function; 0.5 = tip of tail is limp; 1.0 = limp tail; 1.5 = limp tail and hind leg inhibition; 2.0 = limp tail and weakness of hind legs or signs of head tilting; 2.5 = limp tail and dragging of hind legs or signs of head tilting; 3.0 = limp tail and complete paralysis of hind legs or limb tail with paralysis of one front and one hind leg; 3.5 = limp tail and complete paralysis of hind legs and animal unable to right itself when placed on its side; 4.0 = limb tail, complete hind leg and partial front leg paralysis with minimal moving and feeding. Drugs were administered at the onset of clinical symptoms for 30 days. The drugs or the vehicle were daily administered via the intranasal route. The spinal cord/vertebrae were removed and lumbar spinal cord/vertebrae samples were either post-fixed in PFA 4% for 24 hrs and sectioned (7 μm) with a microtome for immunostaining or post-fixed in a mixture of PFA 2% and glutaraldehyde 2% for 5 days, then in cacodylate-buffered 1% osmium tetroxide for 1 h at 4 °C and in 2% uranyl acetate for 1 h at room temperature before embedding in epoxy resin and ultrathin sectioning for electron microscopy.

### Human tissues
Post-mortem brain samples from MS and non-neurological control donors were provided by a UK prospective donor scheme with full ethical approval from the UK Multiple Sclerosis Society Tissue Bank (MREC/02/2/39) and from the MRC Edinburgh Brain Bank (16/ES/0084). MS diagnosis was confirmed by neuropathological means by F. Roncaroli (Imperial College London) and Prof. Colin Smith (Center for Clinical Brain Sciences, University of Edinburgh) and clinical history was provided by R. Nicholas (Imperial College London) and Prof. Colin Smith. Supplementary Tables 1 and 2 include donor characteristics corresponding to the human samples used. Control samples were derived from donors between 44 and 88 years old for whom the cause of death was heart disease, pulmonary disease or cancer. MS samples were derived from donors between 44 and 72 years old for whom the cause of death was heart disease, pulmonary disease, cancer or sepsis. Tissue blocks were used as paraffin sections for RNAscope analysis and cut at 4 μm. White matter lesions were identified and characterized by Anna Williams[70] using Luxol Fast Blue staining and Oil Red O (for lipids phagocytosed by macrophages). Active lesions have indistinct borders and lipid-laden macrophages/microglia. Chronic active lesions have a ring of lipid-laden macrophages/microglia and a core with few immune cells[70].

### Immunostaining experiments
The primary antibodies were as follows: Olig2 (rabbit, AB9610, 1:500, Millipore; mouse, MABN50, 1:500, Millipore), MBP (rabbit, AB980, 1:750, Millipore), Adenomatus Polyposis Coli (APC/CC1) (mouse, OP80, 1:500, Calbiochem), GFAP (mouse, G3893, 1:1500, Sigma), Iba1 (rabbit, W1 W019-19741, 1:500, Wako), Arg-1 (goat, sc-18355, 1:100, Santa-Cruz), PDGFRα (rat, 558774, 1:500, BD Pharmingen), DHT (guinea-pig, GP-DHT1, 1 :200, Synabs), Neurofilament H (NF-H), Non-phosphorylated Smi-32 (mouse, 801701, 1 :300, Biolegend), Ki67 (mouse, 550609, 1:150, BD Pharmingen), Aromatase (rabbit, Ab18995,

1:200, Abcam), Claudin-5 (mouse, 35-2500, 1 :700, Invitrogen), pSTAT3 (Tyr705) (rabbit, 9145, 1: 500, Cell Signaling). AR is a home-made antibody (guinea-pig, Aa 283−298/Aa 406−420 from mus musculus AR Accession NP_038504.1; Eurogentec). The secondary antibodies were: goat anti-rabbit cyanine 3 conjugated (111165003, 1/250, Jackson Immunoresearch); goat anti-mouse Alexa 488 (A11029, 1:250, Thermo Fisher Scientific), goat anti-rabbit Alexa 633 (A21070, 1:750, Thermo Fisher Scientific), goat anti-rat Alexa 633 (A21094, 1:750, Thermo Fisher Scientific), goat anti-rabbit Alexa 488 (A32731, 1:350, Thermo Fisher Scientific); goat anti-guinea pig cyanine 3 conjugated (106165003, 1/500, Jackson Immunoresearch); donkey anti-goat Alexa 488 (A11055, 1:500, Thermo Fisher Scientific).

### High-resolution fluorescent in situ hybridization in human tissues

To detect single AR mRNA molecules within microglia/macrophages, the RNAscope Multiplex Fluorescence v2 Assay (Bio-Techne) was combined with IBA1 immunofluorescence on 4 μm-thick white matter paraffin sections. Briefly, sections were deparaffinised and antigen retrieval was performed using Co-Detection Target Retrieval buffer (Bio-Techne) in a steamer. Endogenous peroxidase activity was quenched with hydrogen peroxide and sections were incubated overnight at 4 °C with a monoclonal anti-Iba1 primary antibody (1:250; ab178846, Abcam). The next day, sections were fixed in 10% neutral buffered formalin to cross-link the primary antibody and the RNAscope assay was performed as per manufacturer's instructions. Briefly, sections were digested with protease and hybridized with a human AR probe (Hs-AR-02, Bio-Techne) for 2 h at 40 °C. Sequential amplifications were performed at 40 °C and the mRNA signal was developed by horseradish peroxidase incubation followed by incubation with Opal 570 (1:750; FP1488001KT, Akoya Biosciences). For Iba1 immunofluorescence, sections were then incubated with anti-rabbit Alexa Fluor 647 secondary antibody (1:750; A-21244, Thermo Fischer Scientific), counterstained with DAPI and mounted with ProLong Glass Antifade mounting medium (Invitrogen). The entire sections were imaged using the Opera Phenix Plus system (PerkinElmer) under a 20x water-immersion objective. Image analysis was performed in QuPath software[71]. Regions of interest were selected to contain different lesion types in MS samples, or at random in control samples. Iba1+ cells that contained AR+ puncta were manually counted in at least 3 different regions of interest per lesion type and expressed as percentage of the total number of Iba1+ cells.

### High-resolution fluorescent in situ hybridization in animal tissues

FISH was performed on frozen brain sections derived from 3 independent animals per group by using RNAscope Multiplex Fluorescent Reagent Kit v2, ACDBio according to the instructions of the provider (Advanced Cell Diagnostics). The probes (Biotechne) and Opal fluorophores (Akova Biosciences) were as follow: *AR* (316991), *Esr1* (478201) *Esr2* (316121), Opal-570 (FP1488001KT, 1 :1500), Opal-620 (FP1495001KT1 :1500).

### Image acquisition and analysis

Images were taken using the microscope analyzing system Axiovision 4.2 (Carl Zeiss, Inc.) and the confocal Zeiss LSM 510-Meta Confocor 2. Analyses were performed with ImageJ software. A total of 3–5 sections per mouse were analyzed. For the brains derived from the LPC-injected animals, the immunofluorescent-positive cells or areas were determined in one every other 5 sections throughout the whole demyelinated lesion per mouse and averaged for each animal. The lesion surface was determined by measuring the area of the nuclear densification (correlated with myelin loss visualized by MBP staining) one every other 5 slices through the whole demyelinated lesion. For the

human post-mortem tissue analysis, the entire sections were imaged using a ZEISS Axio Scan.Z1 slide scanner. All quantifications were performed using Zeiss Zen lite imaging software. For cell density quantification, 3–5 different areas of interest were marked out in control white matter, normal appearing white matter (NAWM) and lesion sites.

### Electron microscopy

Ultrathin sections of lumbar spinal cords were examined using transmission electron microscope (1011 JEOL) equipped with a Gatan digital camera. The g ratio (the ratio between the axon diameter and fiber diameter corresponding to myelin sheath + axon diameter) was estimated by measuring the minimum and maximum axon diameter and fiber diameter for each axon using ImageJ software. A total of 100 to 300 randomly chosen myelinated axons were evaluated for each animal.

### Flow cytometry

Spleen and lymph node cells were isolated using a digestion solution containing 1 mg ml/ Collagenase A, 100 μg/ml DNase and 1 U/ml Dispase (Roche) in Dulbecco's modified Eagle's medium (DMEM) 37 °C 20 min. The mixture obtained was filtered through a Falcon 70 μm nylon cell strainer. The single-cell suspensions were then centrifuged at 600 g and the resulting pellets resuspended in RPMI 1640 culture medium and viable cell numbers were determined using LUNA-FL dual fluorescence cell counter (Logos Biosystems). The spinal cords, which mostly contain non-immune cells unlike the lymphoid organs, were cut in small fragments, mechanically and enzymatically dissociated in a solution containing collagenase A 3 mg/ml, DNAse 100 μg/ml, Dispase 2 mg/ml at 37 °C for 35 min before being filtrated through Falcon® 70 μm Cell Strainer. The suspension was mixed with 30% Percoll and layered on top of a 70% Percoll solution for cell purification and then centrifuged at 500 g without brake for 30 min. The myelin top layer was removed. Immune cells were isolated from the interface and resuspended in RPMI 1640 culture medium. The concentration and viability of single cell suspensions were determined using automated cell counter (LUNA-FL dual fluorescence counter, Logos Biosystems) and acridine orange and propidium iodide staining (Logos Biosystems). Spleen, lymph node, and spinal cord mononuclear cell suspensions were phenotyped by flow cytometry using 50 ng/10^6 cells of the following fluorescent-conjugated monoclonal antibodies (mAb) directed against the cell surface markers CD90.2/Thy1.2 (clone 53−2.1), B220 (clone RA3-6B2), CD4 (clone GK1.5), CD8α (clone 53−6.7), CD44 (clone IM7), CD45RB (clone C363.16 A), CD11b (clone M1/70), CD11c (clone N4/8), Ly-6G (clone RB6-8C5), NK1.1 (clone PK136), CD45 (clone 30-F11), CD206 (clone MR5D3) and the transcription factors Foxp3 (clone FJK-16s), T-bet (clone eBio4BIO) and RORγt (clone B2D) (ThermoFisher Scientific, BD Bioscience). The use of a mAb to the mouse Fcγ receptor (clone 93, eBioscience) avoided non-specific antibody binding. At least 20,000 events were analyzed for each sample. Cell debris, dead cells, and doublets were gated out using the FSC and SSC parameters. Data acquisition was performed at the Flow Cytometry Core Facility IPSIT (Clamart, France). Flow cytometry data were analyzed using FlowJo (Treestar) software.

### Quantification of cytokines

Spleen lymph nodes and spinal cord were dissected and snap-frozen in liquid nitrogen. Samples were then stored at −80 °C until further processing. Frozen tissues were homogenized in cold RIPA lysis buffer (Biorad) according to the manufacturer's instructions in the presence of protease inhibitors (Sigma-Aldrich). The protein extract concentration was measured using the BCA method (Thermo Fisher Scientific) and the expression levels of cytokines (GM-CSF IFN-γ IL-1β IL-2

IL-4 IL-5 IL-10 TNF-α IL-17) secreted by immune cells from spinal cord, spleen and lymph nodes were determined using the Bio-Plex Pro Mouse Cytokine 8-Plex Immunoassay (Biorad) according to the manufacturer's instructions[20].

## Bulk RNA sequencing and analysis

The spinal cords from 4 animals for each group were dissected and frozen in liquid nitrogen for further processing. Total RNA was isolated with the Trizol Reagent protocol (ThermoFisher) from spinal cords and RNeasy Mini Kit (Qiagen) according to instructions of the provider. The RNA-Seq libraries were prepared using the NEBNext Ultra II Directional RNA Library Prep Kit (NEB) and sequenced with the Novaseq 6000 platform (ILLUMINA, $32 \times 10^6$ 100 bp pair-end reads per sample). Quality of raw data was evaluated with FastQC. Poor quality sequences were trimmed or removed with fastp tool, with default parameters, to retain only good quality paired reads. Illumina DRAGEN bio-IT Plateform (v3.6.3) was used for mapping on mm10 reference genome and quantification with gencode vM25 annotation gtf file. Library orientation, library composition and coverage along transcripts were checked with Picard tools. Following analyses were conducted with R software. Female and male datasets were first integrated by using normalization by housekeeping genes[72] present in the two datasets, as a previously demonstrated strategy to reduce unwanted variation from RNA-Seq data[73] (RUVSeq). The integrated data were then normalized with edgeR (v3.28.0) bioconductor packages, prior to differential analysis with glm framework likelihood ratio test from edgeR package workflow. Multiple hypothesis adjusted p-values were calculated with the Benjamini-Hochberg procedure to control FDR. For the differential expression analyses, low expressed genes were filtered, sex was used as covariable (when relevant) and the cut-offs applied were FDR < 0.05. Finally, gene ontology (GO) enrichment analysis of biological processes of the differentially expressed genes (DEGs) was conducted with clusterProfiler R package (v3.14.3). R script detailing these analyses has been deposited in https://github.com/ParrasLab/Androgen-signaling-and-remyelination-Nat-Commun-paper (https://doi.org/10.5281/zenodo.7560637).

## Scoring of differentially expressed genes for their impact in oligodendrogenesis

We used OligoScore (https://oligoscore.icm-institute.org/), a resource based on expert curation scoring strategy for gene signatures or transcriptomic studies related to oligodendrogenesis and (re)myelination. This resource currently implicates curation of 430 genes for which loss-of-function and gain-of-function studies have demonstrated their requirement in the main processes of oligodendrogenesis that we categorized in: specification, proliferation, migration, survival, differentiation and myelination. Gene activities are scored in each process from 1 to 3 (low, medium, strong) either positively or negatively (promoting or inhibiting, respectively), depending on the severity of gain- or loss-of-function phenotypes. The large number of references (~1000) used in the scoring are provided per gene and process.

## Gene set enrichment analysis (GSEA)

We used gseGO function of Cluster profiler R package to find gene sets enriched in the gene list of 105 microglial genes (Supplementary Dataset 10) ranged by the differential expression (logarithmic fold change, logFC) in DTH-treated versus non-treated females and males, respectively. We found 219 gene set enriched in females but only 2 in males. Dotplot and gseaplot functions were used for visualization of enriched gene sets. All gene sets enriched are provided in Supplementary Dataset 10. R script has been deposited in https://github.com/ParrasLab/Androgen-signaling-and-remyelination-Nat-Commun-paper (https://doi.org/10.5281/zenodo.7560637).

## scRNA-Seq analysis

EAEraw.RData object was obtained from Gonçalo Castelo-Branco's lab and processed in R (4.0) using the following packages: Seurat (3.0) for data processing and ggplot2 (v3.3.6) for graphical plots. Seurat objects were first generated using CreateSeuratObject function (min.cells = 5, min.features = 100). Normalized with sctransform function. Cell neighbors and clusters were found using FindNeighbors (dims = 1:30) and FindClusters (resolution = 0. 8) functions. RunPCA, and RunUMAP functions with default parameters. Clusters were annotated based on cell-subtype markers as detailed in the R script, which has been deposited in https://github.com/ParrasLab/Androgen-signaling-and-remyelination-Nat-Commun-paper (https://doi.org/10.5281/zenodo.7560637).

## Statistical analysis

Statistical analysis of mouse histological staining was performed with GraphPad Prism 7.0 software (La Jolla, CA). The significance of differences between means was evaluated by two-tailed, unpaired Student's t test for two independent group comparisons and ANOVA followed by Tukey's or Holm-Sidak's post tests for comparisons of more than two groups and/or several variables. In case of absence of distribution normality (analyzed via D'Agostino & Pearson normality test and Shapiro-Wilk normality test), non-parametric tests (Mann-Whitney two-tailed, Kruskal-Wallis with Dunn's post tests for comparison) were used. Appropriate corrections were done according to the determination of the variance of each sample. The values are the means ± SEM from the number of animals indicated in each plotted graph or as indicated in the corresponding legends. Significance of $p < 0.05$ was used for all analyses. $*p \leq 0.05$; $**p \leq 0.01$; $***p \leq 0.001$; $****p < 0.0001$. For human histological staining, data were analyzed using linear mixed-effects models on R Studio with the lmerTest package, adding diagnosis, sex and lesion type as main effects and accounting for multiple measurements from each sample by the random effects. To determine main effects ANOVAs were used (stats package), and post-hoc comparisons between groups were made using pairwise comparisons in the emmeans package with Tukey method. To ensure data met model assumptions, normal distribution was assessed by Shapiro-Wilk test, and homogeneity of variance by Levene's test. For transcriptomic analyses, multiple testing correction aimed at controlling the false discovery rate (FDR) was performed using the Benjamini-Hochberg method. Cutoff used for FDR was 5%. For differential expression, the workflow used edgeR's quasi-likelihood (QL) pipeline (edgeR-quasi).

## Reporting summary

Further information on research design is available in the Nature Portfolio Reporting Summary linked to this article.

# Data availability

All metadata associated with RNA sequencing generated in the present manuscript are available at https://www.ncbi.nlm.nih.gov/geo/query/acc.cgi?acc=GSE225254. The publicly available independent single-nuclei RNA sequencing database from MS donors is available at https://malhotralab.shinyapps.io/MS_broad/. Human MS data are available at https://ega-archive.org/studies/EGAS00001006345. scRNA-Seq dataset from mouse EAE model is available at https://www.ncbi.nlm.nih.gov/geo/query/acc.cgi?acc=GSE113973. Mm10 reference genome (org.Mm.eg.db_3.15.0.) is available at https://bioconductor.org (DOI: 0.18129/B9.bioc.BSgenome.Mmusculus.UCSC.mm10). The readers can expect to receive any raw data from their request. Source data are provided with this paper.

# Code availability

Analyses of images were performed with ImageJ-win64 v1.41 software (mouse tissues) at https://imagej.nih.gov/ or Zeiss Zen lite blue edition 2.3 software (human tissues) available at https://www.zeiss.fr. Flow

cytometry analysis used FlowJo v10.8.1 (Treestar) software available at https://www.flowjo.com. Statistical analysis was performed with GraphPad Prism 7.0 software (La Jolla, CA) available at https://www.graphpad.com. RNA-seq analyses were conducted with R software available https://www.r-project.org and including edgeR (v3.28.0) bioconductor packages for normalization, edgeR package workflow for differential analysis, clusterProfiler R package (v3.14.3) for gene ontology enrichment analysis, gseGO function of Cluster profiler R package used for gene Set Enrichment Analysis. EAE raw.RData GSE113973 were processed in R (4.0) using the packages: Seurat (3.0) for data processing and ggplot2 (v3.3.6) for graphical plots. The deconvolution of our bulk RNA-Seq datasets used the CIBERSORTx tool on the docker (v20.10.12) module Cibersortx/fractions available at https://cibersortx.stanford.edu. R script detailing these analyses has been deposited at https://github.com/ParrasLab/Androgen-signaling-and-remyelination-Nat-Commun-paper (https://doi.org/10.5281/zenodo.7560637). Scoring of differentially expressed genes for their impact in oligodendrogenesis used the OligoScore resource that has been deposited at https://oligoscore.icm-institute.org/.

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

## Acknowledgements

This work was supported by the French Multiple Sclerosis Foundation ARSEP [RAK17128LLA; RAK19176LLA; RAK21128LLA to E.T.]. A.Z. was funded by Mattern Foundation. A.K. was funded by grants from the French Government and ARSEP. A.W., L.Z. and F.T. were funded by the MS Society UK. We thank UMS44 (Le Kremlin-Bicêtre), A. Schmidt and Imaging platform (Hôpital Cochin, Paris), Y. Marie and Genotyping/ Sequencing core facility (ICM, Paris), D. Langui, A. Baskaran and ICM Quant platform (ICM, Paris) for technical assistance, ICM plateforms iSeq and DAC, particularly F-X. Lejeune for help in transcriptome normalization using housekeeping genes and C. Raoux for help in bulk RNA-seq deconvolution.

## Author contributions

Conceptualization, supervision, funding acquisition: E.T. Design of the experiments: E.T., A.Z., A.K. Achievement of the experiments: A.Z., A.K., T.H.-H., A.M., Co.M., L.Z., F.T. Data analysis: A.Z., A.K., T.H.-H., A.M., Co.M., L.Z., F.T., C.l.M., P.B., M.S., A.W., C.P., E.T. Writing of the original version of the manuscript: E.T. with contributions from the other authors.

## Competing interests

The authors declare no competing interests.
