## [Peer Review File · Nature Communications]

Androgens show sex-dependent differences in myelination in
immune and non-immune murine models of CNS
demyelinationREVIEWER COMMENTS

Reviewer #1 (Remarks to the Author):

The work by Zahaf et al. presents new results on the action of androgens via the AR on inflammation and remyelinating processes in MS in females and the differences compared to males.

General comments:

Comparison of males vs females is done mostly by discussion in the text, while results are almost always presented separately in the plots. This presentation of results does not provide the reader with the ability to have a proper comparison in mind. Further, there is also not always proper statistics for the comparison of the results between males and females.

Major comments:

Overall, the study shows experiments comparing control versus experimental groups with 4 to 5 animals per group. In some of the experiments, the values are showing high intragroup variance. Since the statements might be rather impactful, validation cohorts are necessary throughout the manuscript. Even better would be to provide evidence in other mouse strains for those experiments that do not rely on strain specific settings, e.g. knockout animals or the EAE model. Nevertheless, in these cases validation experiments using the same mouse strain showing the same findings are required.

The human data are not adequate. While it is certainly an unfortunate situation, if a reagent is not available anymore during the performance of a project, as the data are presented at the moment they cannot be really interpreted, since the number of patients studied is too small to make general statements (as they are currently in the manuscript). The link to the human is critical, to avoid a murine model artifact concerning the role of AR in MS. Also, the impact of the findings in the murine system without a link to human disease is an important asset of the manuscript, but requires to be significantly improved. With the lack of the antibody, an alternative approach could be spatial transcriptomics with subcellular resolution. This actually would also allow to measure the other molecules presented in the murine model.

Along these lines the bulk transcriptomics is somewhat confusing. The use of gene terms derived from a study that is still in preparation (line 417) is not appropriate without exactly knowing what is really done by this approach. Further, the text is not very clear about the similarities and differences of male versus female animals in the experimental setting. In addition, since very often bulk differences are more likely due to changes in cell numbers between conditions rather than true transcriptional changes, some of the results might be blurred by such cell type ratio differences. This needs to be further addressed with respective tools available to the expert in the field. Also as the data are presented, it has not been clear, whether the male and female data were analyzed as one dataset or independent datasets. If treated as different datasets, comparisons as they have been described in the paper are not valid, since confounding factors are not adequately addressed. Further, a validation experiment is missing, which could be best by e.g. single cell RNA-seq, since this would better show, whether the transcriptome differences per cell type are indeed different between male and females. Lastly, the data need to be made available prior to publication (at least so that the reviewer sees this) and the respective links need to be added to the manuscript accordingly.

Further important comments:

Line 135: This is indeed a problem, since the numbers of patients are definitely too low to draw any conclusions. The authors need to make any attempt to better quantify this in a larger group of patients, see suggestions above.

Line 150: Findings reported in Fig 2 seem to be performed in a single experiment with 4 mice per group only. A validation experiment, best even in a different murine strain would be required to increase the validity and generalizability of the data, see major comment above in general.

Line 180: "The differential effects induced by testosterone and DHT on PLP expression and microglia response upon demyelination suggested that exogenous testosterone may induce its effects via both AR and/or ER after its aromatase-mediated conversion to estradiol (E2)". To test this hypothesis, the authors performed an experiment on ovariectomized females injecting DHT, E2 or DHT+E2. However, this does not directly test the hypothesis on the aromatase activity. For this, the appropriate test will be to inhibit aromatase activity in the CNS.

Line 268: Why should there be a linear relationship. What is the basis for this assumption? Even visual inspection of the data points in panel E might be better explained by other models

Fig 7 + 8 : The authors compare the DHT effect on EAE progression in females vs males. At the given time point the values for the control treatment for some parameters (e.g. Th17, Tbet+, TNF in the lymph node) are at higher values in females compared to males. Can the authors please clarify if this difference is significant? What is the value of comparing similar treatments of DHT on two different stages of inflammation in the model?

Fig 7 and Fig 8 : in the text the authors compare the results presented in the two figures, though separating the results to two figures does not allow proper comparison for the reader.

Line 412: The knowledge driven scoring strategy for the pathways is not very clear from the methods, and therefore limits the ability to assess the quality of work and moreover will not allow reproducibility of the results. The authors should provide the list with their scoring and properly define the way the scoring was attributed to each gene. Moreover, different numbers of genes overlap with the DE genes in females and males, does this have an impact on the results of the scoring done?

Minor comments:

Markers mentioned are not always explained for their biological relevance. Would be good to be consistent with a brief explanation to provide a friendly reading for people who are not regularly working on CNS. Examples: MBP, line 195 + line line 214, only explained at line 254.

Line 20, Figure 1B: For this reviewer the image was not clear. Maybe it's better to present it in an overlay. Moreover, it seems panel E should fit right after panel B.

Paragraph 1 of results, missing a summarising sentence.

Figure S1: for people with no experience in MS, the detection of lesions in the images are not clear. Can help to have arrows indicating the lesions in the image.

Figure. 7G-H: flow-cytometry, what is the percentage of (%)? of immune cells populations of leukocytes?

Line 446-473: the discussion on specific genes will benefit adding plots to supplementary data or as part of the main figure.

Reviewer #2 (Remarks to the Author):

This study addresses the role of androgens in remyelination and neuroinflammation in females, primarily using two model systems: LPC-induced demyelination of the corpus callosum and C57BL/6 EAE. They report that androgen receptor (AR) is unexpectedly expressed at greater levels in demyelinated lesions of female vs male LPC and MS lesions and further that androgens impact myelin repair (in LPC) and neuroinflammation (in EAE) in female mice. In LPC female mice, AR impacts macrophage/microglia phenotype, although macrophage/microglia-specific effects are not required for augmentation of oligodendrocyte differentiation. In EAE, clinical effect of androgens is similar between female and male mice, although aspects of the central and peripheral immune responses and bulk transcriptomics differ between males and females. Overall they conclude that androgens are required for remyelination in females and have gender-specific effects in both LPC and EAE models.

The strongest conclusion supported by the data is that androgens play a role in myelin repair in female mice, as evidenced by Fig 4 using flutamide to block endogenous AR. This is a consequential finding that may have clinical relevance. The authors also demonstrate that androgen effects at the cellular level differ between male and female mice in the 2 models, which is not surprising but nonetheless of interest with regard to the growing recognition of the importance of sex differences in MS. The overall amount of work included in the manuscript is impressive. The biggest weakness is that the findings are largely descriptive rather than mechanistic, as even the cellular targets that are important for AR effects in females remain uncertain. The key cellular target identified in the LPC model (macrophages and microglia) do not mediate the effects of androgens on remyelination, leaving their role and the key cellular targets unidentified. A minor concern is that there is no unifying theme between the LPC and EAE experiments, such that the manuscript feels like two separate stories.

Specific comments:

1. Given that the most consequential finding is the role of androgens in myelin repair in females, this data should be the most convincing in order to best justify the conclusion.

A few concerns arise in this regard:

- All treatments began on the same day as LPC injection, such that the effects of treatment could reflect protection of OPC/oligos from LPC-induced injury rather than a specific effect on remyelination. Ideally treatment would begin on day 2 or 3, but it would suffice to do a control experiment in which mice treated on day 0 are sacrificed on day 2 to ensure similar extent of demyelination in treated vs vehicle mice.**
- It seems odd that g-ratios were performed in EAE (in which remyelination is minimal) but not LPC. Quantification of remyelination by EM g-ratio should be performed in the LPC model**
- LPC mice are only analyzed at day 7 post-lesion, which is an early time point when oligodendrocyte differentiation has just begun. It's difficult to feel confident in conclusions drawn solely from such an early time point, and a subsequent time point would yield greater insight into the effect of androgens on microglia responses and myelin repair, at least in the flutamide model. As reported (doi: 10.1038/s41593-019-0418-z), microglia phenotype changes over time following LPC demyelination in corpus callosum.**

2. Given the authors' focus on macrophages/microglia, a disappointing aspect of the story is that macrophage/microglia-specific knockout of AR has no impact on the observed effects on OPC and oligodendrocyte differentiation, even though expression of AR on other CNS cells appears minimal in female mice. Yet this finding is glossed over in the manuscript. Could the impact of androgens on myelin repair be mediated by peripheral, infiltrating cells?

3. The human data is difficult to interpret, largely because the extent of samples from which it is derived is not explained clearly. Supplemental Table 7 provides descriptions of the donors, but it's not clear if all these donors were included in the IHC experiments and how many of each type of lesion were included (i.e., the sample size from which numbers were derived). This should be clarified. Further, does "% of CD68+AR+" cells mean the % of CD68+ cells that are double positive for AR+? This is unclear and should be clarified. Finally, these experiments were not completed because an antibody was discontinued. From the methods, it appears 2 other antibodies were tested unsuccessfully. Given the translational importance of this data, are no other methods possible to complete the study? Perhaps in situ hybridization as was used for the LPC mice?

4. For all experiments, why are the female mice ovariectomized? The expression of AR was determined in non-ovariectomized LPC mice (Fig 1), but all subsequent interventional experiments are performed in ovariectomized mice. Might not the expression of AR and other physiologic effects of androgens be altered in female mice after ovariectomy?

Minor point:

1. The number of sections examined per animal for microscopy studies is specified in the methods, but this should be stated in figure legends to make evaluation of rigor easier for the reader.

Reviewer #3 (Remarks to the Author):

This is an exhaustive study demonstrating a plethora of data which, in summary, suggest that androgens play in the context of inflammatory, demyelinating disorders, such as Multiple Sclerosis, a major role in females that is critically different from their role in males. To address this interesting hypothesis, the authors use a well characterized model for remyelination (id est the LPC model) and a model of auto-immune driven inflammatory demyelination (id est the EAE model). As there are fundamental biological differences between each gender, it is pivotal to address divergent effects of drugs and potential treatment strategies. This, the topic of the presented study is of high relevance. As a major limitation, the presented study addresses two distinct but fundamentally different aspects of the MS pathology, remyelination and auto-immune driven inflammation. It would have been more convincing to focus on one aspects and try to understand the cellular mechanisms in more detail. Nevertheless, while the study is worth to be published there are several major and minor aspects that need the full attention of the authors. In particular, the following aspects should be addressed:

1. In the material and methods section it is stated that "drugs were administered at the onset of clinical symptoms until Day 30 after immunization." Was this done per individual animal or was drug treatment started at the same days for the entire cohort? Please specify.

2. Some minor typos should be corrected such as "The RNA-seq libraries were prepared using either the NEBNext Ultra II Directional RNA Library Prep 811 Kit (NEB) and sequenced with the Novaseq" (either should be deleted)

3. The authors state that "In case of absence of distribution normality, non-parametric tests (Mann-Whitney two-tailed, Kruskal-Wallis with Dunn's post tests for comparison) were used." Please state how normal data distribution was evaluated.

4. The authors state that "At 7 days post-lesion (dpi), when the process of spontaneous remyelination is ongoing and corresponds to the end of OPC recruitment and the beginning of their differentiation..." Please either provide appropriate citations for this statement or demonstrated it with the used samples.

5. Figure 1B demonstrates a LPC-induced lesions with ongoing remyelination, as stated by the authors. Is this true for the entire lesions or for the lesion rim? Was there any

difference of AR-expression throughout the lesion? Beyond, it is stated that "we observed a strong AR upregulation in the lesion from females while AR transcripts could be detected at a much lower level in the lesion from males". The demonstrated images are not convincing and quantification should be performed. Is the AR-expression induction due to LPC-induced demyelination, or due to the mechanical, needle-induced injury. Vehicle-treated mice would be required to answer this important question.

6. A key step in androgen action is AR nuclear translocation. Can the authors provide evidence that the AR is indeed expressed in the nuclear compartment? The high-power insert in Fig1C rather suggest a perinuclear expression pattern, especially in IBA1+ cells.

7. The authors should clearly state which control experiments were performed to demonstrate the specificity of their stains.

8. The studies using MS tissues appear to be preliminary, and I am not sure whether they add much to the paper with this limited number of investigated cases. In case no reliable antibodies are available the authors could dissect different lesions of cyrosections and perform mRNA expression analyses.

9. Again, in figure 2 it remains unclear whether the observed effects of testosterone and DHT on astrocytes and microglia are linked to the LPC-induced demyelination or the mechanical injury induced during the LPC-application. Sham-operated groups would be required.

10. In the text it is stated that aromatase to be upregulated in the lesion from female mice. The corresponding figure demonstrates indeed aromatase expression but whether this is due to the LPC-induced demyelination remains unclear. Again, sham-operated mice would be supportive.

11. The authors state that "flutamide-treated animals displayed a significant decrease in the percentage of OPCs that are able to differentiate into CC1+ oligodendrocytes compared to the vehicle condition". This sentence appears misleading. As the percentage of OLIG2/CC1 double positive cells in relation to the entire OLIG2 cell population is lower in flutamide-treated mice, this would mean less cells mature under flutamide treatment. Please rephrase.

12. In some cases, myelination is estimated by anti-PLP, in other by anti-MBP stains. The authors should either comment on this discrepancy or perform both stains for all the subexperiments.

13. The authors state that microglia are the main AR-expressing cells in the CNS. Did the authors consider that a significant proportion of these cells are IBA1+ recruited monocytes in the LPC model.

14. Figure 6K should be Claudin5+ area instead of Claudin+ area.

15. The presented results using the EAE model are somewhat irritating. First, the authors should try to focus on the most relevant findings and move the less relevant ones into the supplements. Second, their FACS analysis clearly demonstrate that the observed protective effects of DHT are at least in part due to immunosuppressive functions. For example, proportions of CD4+ T cells as well as the proinflammatory Th1 and Th17 cells are lower in the secondary lymphoid organs in DHT-treated compared to Vehicle-treated mice. However, the main focus of this manuscript so far was induction of remyelination. It is not clear to me how these data help to strengthen the so far observed pro-myelinating effects. I rather would suggest to verify the pro-myelinating effects in another model of remyelination (such as the cuprizone model) or to start treatment during the chronic phase of the EAE disease when the lesions are fully established.

16. I am not sure if the NGS data add much to the manuscript. Since bulk RNA sequencing was performed, it is hard to assign the observed expressional changes to a specific cell type.

REVIEWER COMMENTS

We thank the three Reviewers for their constructive comments. By performing additional work, we have addressed the important concerns that have been raised. As you will read in our detailed point by point answers below, we have performed RNAscope experiments by using an AR probe on human tissues that confirm our preliminary data using AR antibody. We also provide an interactive web browser link available in Biorxiv and related to a work that we submitted to publication after submission of the present manuscript. These Biorxiv data correspond to the analysis of cell-type specific gene expression levels and transcriptomic changes in MS versus control tissues. They corroborate the existence of AR sex differences in human. Moreover, regarding the need to show further mechanistic insights, we first performed new experiments in order to increase our cohorts of animals including those belonging to the conditional mouse strain AR fl/fl ; CX3CR1-CreERT2. These new data now clearly show that AR expressed in female microglia is also required for the differentiation of OPCs into CC1+ oligodendrocytes and for the decrease of astrogliosis thus identifying microglia AR as a major target for the remyelinating effects of androgens exclusively in females. Second, we provide validation for several cellular and molecular targets of AR-mediated effects of androgens specific to females, in particular by using bioinformatics tools available in the field. We think that our additional work addressing the major and more minor concerns raised by the Reviewers allowed us to improve our manuscript. All corrections are visible in red throughout the manuscript.

Reviewer #1 (Remarks to the Author): The work by Zahaf et al. presents new results on the action of androgens via the AR on inflammation and remyelinating processes in MS in females and the differences compared to males.

General comments: Comparison of males vs females is done mostly by discussion in the text, while results are almost always presented separately in the plots. This presentation of results does not provide the reader with the ability to have a proper comparison in mind. Further, there is also not always proper statistics for the comparison of the results between males and females.

REPLY: We agree with the Reviewer that the new data comparing males vs females provided in the present work, are inadequately illustrated. In particular, this is true for Figures 7 and 8 (immune cell sorting / cytokine analysis), Figure 9 (parenchymal inflammatory cells) and Figure 10 (RNA-Seq data). **First**, Figures 7 and 8 have been replaced in the present Revised version 1 (R1) by a single figure (Figure 8-R1), which gathers cell sorting and cytokine data from females and males. This new presentation provides evidence for statistically significant differences in the regulation of several types of immune cells and cytokines in DHT-treated females, but not in DHT-treated males. In addition, we provide as Supplementary Figure 9-R1 the direct comparison between female and male immune cell type proportions under the Vehicle- or DHT conditions. **Second**, data characterizing microglia/macrophages and astrocytes in the spinal cord from EAE females and males were already gathered in a single figure in the initial version of the manuscript (Figure 9). This figure is still called Figure 9-R1. However, we have now added Supplementary Figure 10-R1), which provides proper statistics for the direct comparison of microglia/macrophages and astrocytes in females versus males under either vehicle- or DHT-treatment. **Third**, regarding RNA-Seq data, we provide now single Supplementary Tables that display the data obtained in females versus males in particular for all deregulated genes (Supplementary Table 3-R1), ontology analysis (Supplementary Table 4-R1), genes specifically deregulated in microglia and astrocytes (Supplementary Tables 5-7-R1).

Major comments: Overall, the study shows experiments comparing control versus experimental groups with 4 to 5 animals per group. In some of the experiments, the values are showing high intragroup variance. Since the statements might be rather impactful, validation cohorts are necessary throughout the manuscript. Even better would be to provide evidence in other mouse strains for those experiments that do not rely on strain specific settings, e.g. knockout animals or the EAE model. Nevertheless, in these cases validation experiments using the same mouse strain showing the same findings are required.

REPLY: As requested by the Reviewer, we have performed validation experiments for all the experimental protocols using either LPC or EAE demyelination in order to increase our animal cohorts. Consequently, we have edited the corresponding data (in Results and Figures) throughout the whole manuscript. In addition, we have used another mouse strain (129X1/SvJ) and provide now evidence that the pharmacological blockade of AR by flutamide impedes myelin regeneration in 129X1/SvJ strain as it does in C57BL/6 mice used in our work. Data associated with 129X1/SvJ strain are now available in Supplementary Figure 3-R1.

The human data are not adequate. While it is certainly an unfortunate situation, if a reagent is not available anymore during the performance of a project, as the data are presented at the moment they cannot be really interpreted, since the number of patients studied is too small to make general statements (as they are currently in the manuscript). The link to the human is critical, to avoid a murine model artifact concerning the role of AR in MS. Also, the impact of the findings in the murine system without a link to human disease is an important asset of the manuscript, but requires to be significantly improved. With the lack of the antibody, an alternative approach could be spatial transcriptomics with subcellular resolution. This actually would also allow to measure the other molecules presented in the murine model.

REPLY : The Reviewer is right. Since the submission of our manuscript, we have submitted another manuscript entitled ‘Single nuclei RNA-Seq stratifies multiple sclerosis patients into three distinct white matter glia responses’ by Macnair and collaborators presently available as a Biorxiv (<https://www.biorxiv.org/content/10.1101/2022.04.06.487263v1.article-info>). An interactive web browser to analyse cell-type specific expression levels of genes and transcriptomic changes in MS versus control tissue is available at https://malhotralab.shinyapps.io/MS_broad/ (for broad cell types) and at https://malhotralab.shinyapps.io/MS_fine/ (for fine cell types). AR sex difference can be observed and confirm our preliminary data using AR antibody (Supplementary Figure 1-R1) as shown now in the plot provided as Figure 2d-R1.

In addition, we have now performed new RNAscope experiments by using an AR RNAscope probe on human tissues. The data have now been included in the new Figure 2-R1. Results are described (lines 97-105). Corresponding Methods have been added (lines 549-575).

Along these lines the bulk transcriptomics is somewhat confusing. The use of gene terms derived from a study that is still in preparation (line 417) is not appropriate without exactly knowing what is really done by this approach.

REPLY : We agree with the Reviewer that several aspects of our bulk transcriptomic analysis require to be clarified. This is namely the case for the scoring procedure aimed at assessing the impact of DHT on the process of oligodendrogenesis. Therefore, we now provide four .xlsx tables and histograms with the detailed analyses based on our curation strategy of scoring the deregulated genes implicated in different aspects (processes) of oligodendrogenesis. These data

are for the exclusive use of the Reviewers. They are called ‘Additional data for the Reviewers_Tables 1-4). Additional data for the Reviewers_Tables 1 and 2 (Tab 1) correspond to all genes (n=391) selected according to the approach now described in Methods (lines 660-667) and for which the fold changes between DHT-treated data versus Vehicle-treated data are shown for females and males, respectively. Additional data for the Reviewers_Tables 3 and 4 (Tab 1) correspond to the deregulated genes (211 in females and 95 in males) for which the value of the False Discovery Rate (FDR) correction (that is considered to be an indicator of the strength of a study) is < 0.05 . The full curation dataset with the corresponding references for each gene will be published in another study in preparation by the laboratory of Dr Carlos Parras and made accessible through a website resource for the scientific community. Tab 2 of each .xlsx file presents graphs derived from the above data. The four graphs visualize in females and males: 1) The total score characterizing each oligodendroglial process; 2) The number of genes involved in each process; 3) The score of genes promoting and inhibiting each process; 4) The number of genes promoting and inhibiting each process.

All these analyses and the corresponding graphs have been now performed after normalization of the female and male data according to the request of the Reviewer (see below). Consequently, we have updated the ‘oligodendrogenesis’ histograms (Figure 10d,e-R1), added the new Supplementary Figure 12-R1; see below for details) and edited the Results (lines 323-334).

Further, the text is not very clear about the similarities and differences of male versus female animals in the experimental setting.

REPLY : In order to better describe the similarities and differences of male versus female animals in the experimental setting, as mentioned above, we provide now :

- **RNA-Seq data** by systematically comparing females and males in a single table with statistics analyses of the differences between DHT and Vehicle (control) females, DHT and Vehicle males, DHT females and the normalized female/male Vehicle, DHT males and the normalized female/male Vehicle, DHT females and DHT males, Vehicle females and Vehicle males. Thus, ‘Supplementary Table 3-R1_DEGs now replaces ‘Tables 1 and 2_DEGs_DHT_cpm_EdgeR_FDR005’ from the initial version of the manuscript. ‘Supplementary Table 4-R1 now replaces ‘Tables 3 and 4_GO_DHT_EdgeR_FDR005’ from the initial version of the manuscript. Supplementary Tables 5-R1 and Table 6-R1 replace Table 5_microglia_genes and Table 6_astroglia_genes, respectively. The new ‘Supplementary Table 7-R1’ shows genes deregulated either in both females and males or only in females and only in males.
- **Data for oligodendroglial processes** in the Supplementary Figure 12-R1 displaying the scores and number of genes promoting or inhibiting these processes in females and in males as well as the list of genes deregulated in the myelination process according to the sex of the animals.
- **Cell sorting and cytokine data** in a single figure (Figure 8-R1) allowing a direct comparison between female and male DHT effects as well as in the Supplementary Figure 9-R1 comparing the proportions of immune cells and the levels of cytokines under Vehicle or DHT conditions in females versus males.
- **A direct comparison of microgliosis and astrogliosis** in females and males from the EAE model at 14 dpi, including statistics between Vehicle-treated females and males

on one side and DHT-treated females and males on the other side as presented in Supplementary Figure 10.

Also as the data are presented, it has not been clear, whether the male and female data were analyzed as one dataset or independent datasets. If treated as different datasets, comparisons as they have been described in the paper are not valid, since confounding factors are not adequately addressed.

REPLY: The experiments of DHT administration were done first for females and then for males, given the amount of efforts and resources needed for these EAE experiments. In order to take into account the remark of the Reviewer, we have now compared male and female data as one dataset. For this purpose, both datasets have been integrated using a normalization by recently identified mouse housekeeping genes (Li et al., 2017) present in the two datasets, as a previously demonstrated strategy to reduce unwanted variation from RNA-Seq data (RUVSeq) (Risso et al., 2014). We provide now - only for the Reviewers - graphs visualizing the normalization of female and male data as well as the comparison of the differentially expressed genes in different group-comparisons, which show comparable numbers of DEGs in particular between females treated or not with DHT, and between males treated or not with DHT (Additional data for the Reviewers-Figure 1a,b).

Consequently, we have now :

- described the step of normalization in the revised version of the manuscript (Methods, lines 647-649)

- updated all panels in Figure 10-R1 and Supplementary Figure 12-R1.

- updated Supplementary Tables 3 and 4 reporting Gene ontology (GO) analysis of differentially expressed genes as well as Supplementary Tables 5 and 6 depicting microglia- and astroglia-related DEGs. These tables are now provided as ‘Supplementary Table 4_GO_DHT_females-males_EdgeR_FDR005_R1’, ‘Supplementary Table 5_microglia_genes_R1’ and ‘Supplementary Table 6_astroglia_genes_R1’.

- accordingly edited the Results (lines 315-343).

After this normalization, we still found that both DHT-treated females and males were clearly separate from their respective controls by principal component and clustering analyses indicating a clear effect of DTH treatment. The top processes involving up-regulated genes were still enriched in terms promoting neuronal activity and function while those enriched in down-regulated genes showed sexual dimorphism, in females related to the immune system / inflammation and in males related to lipid metabolic processes. Thus, this normalization step confirms that gene deregulation is indeed caused by the treatment, but not by other confounding factors not adequately addressed.

In addition, since very often bulk differences are more likely due to changes in cell numbers between conditions rather than true transcriptional changes, some of the results might be blurred by such cell type ratio differences. This needs to be further addressed with respective tools available to the expert in the field.

REPLY : We agree with the Reviewer to tell that bulk differences might be due to changes in cell numbers between conditions rather than true transcriptional changes. However, we provide now several arguments that do not support this hypothesis.

Regarding microglia/macrophages-related genes for which the differences were major, our cell sorting experiments indicate that the percentages of the whole population of phagocytes including CD45⁺ CD11b⁺ CD44⁻ microglia and CD45⁺ CD11b⁺ CD44⁺ macrophages remained unmodified in the spinal cord from DHT-treated females and males compared to their respective control (Fig. 8f-R1). This observation supports the idea that the down-regulation of genes observed in DHT-treated females is likely related to true transcriptional changes.

In addition, accordingly to the Reviewer comment, we have used a scRNA-Seq dataset from mouse EAE model (Falcao et al., 2018; GSE113973) to show in the EAE microglial cells (microE cluster, see Additional data for the Reviewer_Figure 2), the specific expression of genes reported as markers of the different activated microglia profiles presently found to be exclusively down-regulated in DHT-treated females but not in DHT-treated males compared to their respective control. In this purpose, we have listed the p-values of DEGs from each prototypical class of microglia and identified 33 genes out of 101 that were down-regulated only in DHT-treated females compared to their control without being deregulated in Vehicle-treated females versus Vehicle-treated males (Supplementary Table 7-R1 ; pink-highlighted genes). Then, we established the dotplots of those 33 genes. 13 out of 33 (including *Capg*, *Ccl6*, *Cd33*, *cd52*, *Clec7a*, *Ctss*, *Cybb*, *Fcgr2b*, *Il1b*, *Itgax*, *Lgals3*, *Tlr4*, *Tnf*) displayed a high average expression mostly in microglial cells (Supplementary Figure 14) validating again that their exclusive downregulation in DHT-treated females was likely not related to any decrease in cell number. The manuscript has been accordingly edited (lines 348-365).

Regarding prototypical classes of astroglia (Supplementary Table 7-R1). 83 genes out of a total number of 165 DEGs related to astroglia were down (50) or up (33) -regulated only in DHT-treated females (pink-highlighted) compared to female controls whereas only 20 were down (16) or up (4)-regulated only in DHT-treated males (blue-highlighted) compared to male controls. None of those 83 and 20 DEGs displayed differential expression between female and male controls suggesting that the deregulation was caused by the treatment, but not by the sex of the animal. Additionally, 62 genes were deregulated in both DHT-treated females and DHT-treated males with (29) or without (33) a significant difference between female and male controls. Additionally, the high majority of genes (21 out of 26) known to identify activated astrocytes, pro- (A1) and anti- (A2) inflammatory astroglial phenotypes (including *Ggta1*, *Psm8*, *Serping1*, *Srgn*, *Cd109*, *Cd14*, *C1cf1*, *Emp1*, *S100a10*, *Sphk1*, *Tm4sf1*, *Actn1*, *Bgn*, *C1ql1*, *C4b*, *Ifitm3*, *Igfbp7*, *Ntrk2*, *S100a10*, *S100a11*, *S1pr3*) was exclusively down- (19) or up- (2) regulated in DHT-treated females compared to female controls. Although there are not astroglia in the EAE dataset from Falcao et al (Falcao et al., 2018) (GSE113973) allowing us to establish the dotplots of those genes as done above for microglia, this observation suggests the ability of DHT to control astrogliosis in females in a specifically different way compared to males (Supplementary Table 7-R1). The manuscript has been accordingly edited in the Results (lines 367-373).

Further, a validation experiment is missing, which could be best by e.g. single cell RNA-Seq, since this would better show, whether the transcriptome differences per cell type are indeed different between male and females.

Actually, our bulk RNA-Seq experiment validates and extends the phenotypic characterization made by different approaches throughout the manuscript including immunofluorescence or cytokine determination. Indeed, as a whole, bulk RNA-Seq data revealed the substantially higher ability of DHT to down-regulate activated profiles of microglia and astrocytes in demyelinated females than in demyelinated males. In agreement with this finding, 1) AR expression is detected mostly in microglia and to a lower extent in astrocytes in the

demyelinated lesions from females, but not in the demyelinated lesions from males in which AR expression is quite below the detection threshold (Fig. 1-R1); 2) Among the genes related to microglia and shown to be downregulated in RNA-Seq analysis of DHT-treated females but not DHT-treated males (Supplementary Tables 5-R1 and 7-R1), the downregulation of *Ilib* in females is validated by the decrease of IL-1 β levels in the spinal cord from DHT-treated females, but not DHT-treated males (Fig. 8f-R1). Additionally, we have also validated the female-specific downregulation of *Tnf* (exclusively detected in microglia in the EAE model as shown in Supplementary Figure 14-R1) and *Csf2* genes by using quantitative RT-PCR amplification of DHT-treated female mRNA (Supplementary Figure 13-R1). 3) In the same line, the downregulation of the well-known marker of reactive astrocytes STAT3 is only observed in RNA-Seq analysis from DHT-treated females, which is validated by the visualization of GFAP+STAT3+ immunofluorescence decrease in spinal cord slices from DHT-treated females (Figure 3j-k-R1). Still validating this RNA-Seq data, we previously published that in the presence of the aromatase inhibitor fadrozole, testosterone was unable to decrease GFAP+ STAT3+ labelling in LPC demyelinated lesions from males (Laouarem et al, 2021). For all these reasons, we consider that the request of a third source of experimental evidence to reinforce the results obtained, such as the particularly expensive and difficult experiments like new scRNA-Seq datasets of EAE models would not be fully justified.

Lastly, the data need to be made available prior to publication (at least so that the reviewer sees this) and the respective links need to be added to the manuscript accordingly.

REPLY: The data have been made available to the Reviewers on March 18th, 2022 as indicated by the following message from NCBI and transferred to The Editorial Board of the Journal the same day :

Sujet : Reviewer link created for BioProject PRJNA816168
Date : Fri, 18 Mar 2022 02:47:39 -0400 (EDT)
De : sra@ncbi.nlm.nih.gov
Pour : elisabeth.traiffort@inserm.fr
Dear Elisabeth Traiffort,

Your BioProject's metadata is available at :
<https://dataview.ncbi.nlm.nih.gov/object/PRJNA816168?reviewer=m32uflgbbafp5c4ern5ukvb7g4> in read-only format. You may forward this email to your publisher to share with your reviewer(s) or send them the URL above. It will remain active and reflect all metadata associated with your BioProject until your BioProject is released to the public.

The link above has been now added to the manuscript (lines 689-691).

Further important comments:

Line 135: This is indeed a problem, since the numbers of patients are definitely too low to draw any conclusions. The authors need to make any attempt to better quantify this in a larger group of patients, see suggestions above.

REPLY : The Reviewer is right. Since the submission of our manuscript, we have submitted another manuscript entitled 'Single nuclei RNA-Seq stratifies multiple sclerosis patients into three distinct white matter glia responses' by Macnair and collaborators presently available as a Biorxiv (<https://www.biorxiv.org/content/10.1101/2022.04.06.487263v1.article-info>). An interactive web browser to analyse cell-type specific expression levels of genes and

transcriptomic changes in MS versus control tissue is available at https://malhotralab.shinyapps.io/MS_broad/ (for broad cell types) and at https://malhotralab.shinyapps.io/MS_fine/ (for fine cell types). AR sex difference can be observed and confirm our preliminary data using AR antibody (Supplementary Figure S1-R1) as shown now in the plot provided as Figure 2d-R1.

In addition, we have now performed new RNAscope experiments by using an AR RNAscope probe on human tissues. The data have now been include in an additional figure in the revised version (Figure 2-R1). Results are now described (lines 97-105). Corresponding Methods have been added (lines 549-575).

Line 150: Findings reported in Fig 2 seem to be performed in a single experiment with 4 mice per group only. A validation experiment, best even in a different murine strain would be required to increase the validity and generalizability of the data, see major comment above in general.

REPLY: The experiment presented in Figure 2 has been repeated with a new cohort of animals and allowed us to pool animals data leading to n=8 per condition. Figure 2 has thus been edited and is now called Figure 3-R1. As indicated above, we have also performed validation experiments for all the experimental protocols using either LPC or EAE demyelination in order to increase our cohorts until 6-8 animals / condition. Consequently, we have edited the corresponding histograms. In addition, we have used another mouse strain (129X1/SvJ) and provide now evidence that the pharmacological blockade of AR by flutamide impedes myelin regeneration in both C57BL/6 and 129X1/SvJ strains. These data are now available in Supplementary Figure 3-R1 and are mentioned at line 171-172.

Line 180: “The differential effects induced by testosterone and DHT on PLP expression and microglia response upon demyelination suggested that exogenous testosterone may induce its effects via both AR and/or ER after its aromatase-mediated conversion to estradiol (E2)”. To test this hypothesis, the authors performed an experiment on ovariectomized females injecting DHT, E2 or DHT+E2. However, this does not directly test the hypothesis on the aromatase activity. For this, the appropriate test will be to inhibit aromatase activity in the CNS.

REPLY: As recommended by the Reviewer, we have performed a new experiment by using the aromatase inhibitor fadrozole, a molecule that we previously used in a recent publication (Laouarem et al, *Glia*, 2021). These new data are included in Figure 4-R1 (panels j-n). They confirm the results derived from the experiments using DHT and E2. Indeed, inhibiting the conversion of testosterone to estradiol by fadrozole maintains the increase of MBP staining compared to the vehicle even though the increase is found significantly lower than the one induced by testosterone alone corroborating that DHT and E2 may display additive effects on MBP expression in the lesion. Moreover, in the presence of fadrozole, testosterone still decreases Iba1+ microglia staining, but also increases Arg-1+ staining in a consistent manner with our previous results (Fig. 4h, i-R1) showing that only DHT (not E2) induces Arg-1 expression in the LPC lesion. The results are included in the manuscript (lines 157-163).

Line 268: Why should there be a linear relationship. What is the basis for this assumption? Even visual inspection of the data points in panel E might be better explained by other models.

REPLY: The Reviewer is right. Our representation of the g-ratio according to axon diameter was not appropriate. We have withdrawn the sentence regarding the linear relationship (line 204), changed the panel dedicated to the analysis of the g-ratio (Figure 7e-R1). The graphs now

clearly visualize that the g-ratio values are much lower under testosterone or DHT treatments than in the vehicle condition. We also provide the histogram representing the mean g-ratio value in each group of animals.

Fig 7 + 8 : The authors compare the DHT effect on EAE progression in females vs males. At the given time point the values for the control treatment for some parameters (e.g. Th17, Tbet+, TNF in the lymph node) are at higher values in females compared to males. Can the authors please clarify if this difference is significant? What is the value of comparing similar treatments of DHT on two different stages of inflammation in the model?

REPLY: Our choice to characterize the effects of DHT at both 14 dpi (Figure 8-R1 and Figure 9-R1) and 30 dpi (Figure 7-R1) has been guided by the idea to detect both precocious effects of the hormones at the time when the neurological scores become significantly different between the vehicle group and the hormone-treated group in order to avoid potential compensatory mechanisms occurring with time and, later effects when the scores have reached their respective plateau. We have now mentioned more clearly this point in our text (lines 228-230). The Reviewer is right regarding the higher value of some parameters in vehicle-treated females compared to males. Now, we provide the new Supplementary Figure 9-R1 that, indeed, shows significantly higher levels of Th1 and Th17 cells in the lymph nodes from vehicle-treated females compared to males. The difference disappears in the presence of DHT. All sex-dependent differences in immune cells are now described and commented (lines 259-271).

Fig 7 and Fig 8 : in the text the authors compare the results presented in the two figures, though separating the results to two figures does not allow proper comparison for the reader.

REPLY: As recommended by the Reviewer, we have replaced Fig. 7 and 8 by a single figure in which both female and male data are gathered. As mentioned above, Supplementary Figure 9 also allows proper comparison for the reader.

Line 412: The knowledge driven scoring strategy for the pathways is not very clear from the methods, and therefore limits the ability to assess the quality of work and moreover will not allow reproducibility of the results. The authors should provide the list with their scoring and properly define the way the scoring was attributed to each gene. Moreover, different numbers of genes overlap with the DE genes in females and males, does this have an impact on the results of the scoring done?

REPLY: We agree with the Reviewer that several aspects of our bulk transcriptomic analysis require to be clarified. This is namely true for the scoring procedure aimed at assessing the impact of DHT on the process of oligodendrogenesis. Therefore, we now provide four .xlsx tables and histograms with the detailed analyses based on our curation strategy of scoring the deregulated genes implicated in different aspects (processes) of oligodendrogenesis for the exclusive use of the Reviewers (Additional data for the Reviewers_Tables 1-4). Tables 1 and 2 (Tab 1) correspond to all genes (n=391) selected according to the approach now described in Methods (lines 660-667) and for which the fold changes between DHT-treated versus Vehicle-treated data are shown for females and males, respectively. Tables 3 and 4 (Tab 1) correspond to the deregulated genes (211 in females and 95 in males) for which the value of the False Discovery Rate (FDR) correction (that is considered to be an indicator of the strength of a study) is < 0.05 . The full curation dataset with the corresponding references for each gene will be published in another study in preparation by the laboratory of Dr Carlos Parras and made accessible as a resource for the scientific community. Tab 2 of each .xlsx file presents graphs derived from the above data. The four graphs visualize in females and males: 1) The total score

characterizing each oligodendroglial process; 2) The number of genes involved in each process; 3) The score of genes promoting and inhibiting each process; 4) The number of genes promoting and inhibiting each process.

All these analyses and the corresponding graphs have been performed after normalization of the female and male data according to the request of the Reviewer (see above). Consequently, we have updated the 'oligodendrogenesis' histograms (Figure 10-R1), added a new figure (Supplementary Figure 12-R1; see above for details) and edited the Results (lines 323-334).

Minor comments:

Markers mentioned are not always explained for their biological relevance. Would be good to be consistent with a brief explanation to provide a friendly reading for people who are not regularly working on CNS. Examples: MBP, line 195 + line line 214, only explained at line 254.

REPLY: We have now added the useful explanation for MBP (line 127), Iba1 (line 82), GFAP and Olig2 (line 84-85) that were indeed lacking.

Line 20, Figure 1B: For this reviewer the image was not clear. Maybe it's better to present it in an overlay. Moreover, it seems panel E should fit right after panel B.

REPLY : We have now modified Figure 1b by using an overlay and we have placed panel E right after panel B.

Paragraph 1 of results, missing a summarising sentence.

REPLY: A sentence has been added (lines 92-94).

Figure S1: for people with no experience in MS, the detection of lesions in the images are not clear. Can help to have arrows indicating the lesions in the image.

REPLY: We have now added further information to the legend of Supplementary Figure 1-R1 to help clarify what we are drawing attention to. The pictures are taken so that they encompass the lesions, so arrows are not helpful. However, this should be clearer.

Figure. 7G-H: flow-cytometry, what is the percentage of (%)? of immune cells populations of leukocytes?

REPLY: Spinal cord, in contrast with spleens and lymph nodes, contain mainly non-immune cells. We therefore used the CD45 leukocyte marker to discriminate between CD45+ leukocyte and CD45- non-immune cells and determined the percentage of CD45+ immune cells for each condition (Fig. 8f-R1). In contrast, in spleens and lymph nodes (Figure 8d, e-R1), excepted the capsule, all cells are leukocytes. Thus, we considered that CD45 markers was not necessary for this experiment since 100% of cells in suspension expressed CD45. This point has been written more clearly in the Methods (line 608) and in Figure 8 legend (line 1097).

Line 446-473: the discussion on specific genes will benefit adding plots to supplementary data or as part of the main figure.

REPLY: According to the Reviewer's comment, we have now provided a new 'xlsx' file (Supplementary Table 7-R1) reporting the DEGs related to the different microglia (Tab 1) and

astroglia (Tab 2) profiles showing those specifically deregulated upon DHT treatment only in females or only in males or in both females and males. In addition, we have added histograms including genes characteristic of different classes of microglia (homeostatic, DAM, WAM) in Figure 10-R1 (panels j, k, l).

Reviewer #2 (Remarks to the Author):

This study addresses the role of androgens in remyelination and neuroinflammation in females, primarily using two model systems: LPC-induced demyelination of the corpus callosum and C57BL/6 EAE. They report that androgen receptor (AR) is unexpectedly expressed at greater levels in demyelinated lesions of female vs male LPC and MS lesions and further that androgens impact myelin repair (in LPC) and neuroinflammation (in EAE) in female mice. In LPC female mice, AR impacts macrophage/microglia phenotype, although macrophage/microglia-specific effects are not required for augmentation of oligodendrocyte differentiation. In EAE, clinical effect of androgens is similar between female and male mice, although aspects of the central and peripheral immune responses and bulk transcriptomics differ between males and females. Overall they conclude that androgens are required for remyelination in females and have gender-specific effects in both LPC and EAE models.

The strongest conclusion supported by the data is that androgens play a role in myelin repair in female mice, as evidenced by Fig 4 using flutamide to block endogenous AR. This is a consequential finding that may have clinical relevance. The authors also demonstrate that androgen effects at the cellular level differ between male and female mice in the 2 models, which is not surprising but nonetheless of interest with regard to the growing recognition of the importance of sex differences in MS. The overall amount of work included in the manuscript is impressive.

The biggest weakness is that the findings are largely descriptive rather than mechanistic, as even the cellular targets that are important for AR effects in females remain uncertain. The key cellular target identified in the LPC model (macrophages and microglia) do not mediate the effects of androgens on remyelination, leaving their role and the key cellular targets unidentified.

REPLY: We agree with the Reviewer that our investigation of the key cellular target identified in the LPC model in females was really disappointing. However, as previously recommended by Reviewer 1 and because some data were showing intragroup variance (in particular for the determination of the ability of OPC to differentiate into CC1⁺ oligodendrocytes in the presence of DHT in the conditional mutant), we performed validation cohorts for all EAE and LPC experiments including the one using the mouse strain allowing the conditional removal of AR from microglia. Now, the data indicate that the removal of AR from microglia in females clearly prevents DHT to induce OPC differentiation and astrogliosis decrease whereas the first cohort led to show only lower but non-significant effects of DHT in the mutant compared to the wild-type animals. This new data identifies microglia as a key cellular target of AR-mediated effects of androgens in females. In addition, we have now clearly listed genes known to characterize various classes of microglia and astrocytes which here appear to be selectively deregulated under DHT treatment only in females and not in males towards an anti-inflammatory effect. These mechanistic data have been now included in Fig. 6-R1 and Supplementary Table 7-R1 and have been described in the text (lines 181-185; 348-373).

A minor concern is that there is no unifying theme between the LPC and EAE experiments, such that the manuscript feels like two separate stories.

REPLY: We understand the comment of the Reviewer and have included more clearly in the text the requirement for considering remyelination in the context of peripheral immune cell infiltration as it occurs in MS (lines 188-189).

Specific comments:

1. Given that the most consequential finding is the role of androgens in myelin repair in females, this data should be the most convincing in order to best justify the conclusion. A few concerns arise in this regard:

- All treatments began on the same day as LPC injection, such that the effects of treatment could reflect protection of OPC/oligos from LPC-induced injury rather than a specific effect on remyelination. Ideally treatment would begin on day 2 or 3, but it would suffice to do a control experiment in which mice treated on day 0 are sacrificed on day 2 to ensure similar extent of demyelination in treated vs vehicle mice.

REPLY: Actually, LPC injection has been done 15 hrs after LPC injection in order to use a 'therapeutic administration of the hormones' as done in the EAE model where hormone administration starts at the onset of neurological disabilities. However, the Reviewer is fully right, we did not correctly mention this point neither in the text nor in the schemes describing the LPC protocols. We have now edited the text (line 121) and the different schemes visualizing the experimental protocols in Figures 3-R1 to 6-R1.

- It seems odd that g-ratios were performed in EAE (in which remyelination is minimal) but not LPC. Quantification of remyelination by EM g-ratio should be performed in the LPC model

REPLY: The reviewer is right. We have now performed a new experiment in which animals have been analyzed at 14 dpl in order to visualize axons and myelin sheaths at the ultrastructural level. The data have been added in Figure 3-R1 (panels l-o) and described in lines 134-139. They indicate a highly significant decrease of the g-ratio values in the DHT-treated mice.

- LPC mice are only analyzed at day 7 post-lesion, which is an early time point when oligodendrocyte differentiation has just begun. It's difficult to feel confident in conclusions drawn solely from such an early time point, and a subsequent time point would yield greater insight into the effect of androgens on microglia responses and myelin repair, at least in the flutamide model. As reported (doi: 10.1038/s41593-019-0418-z), microglia phenotype changes over time following LPC demyelination in corpus callosum.

REPLY: As requested by the Reviewer, we performed a new LPC experiment including flutamide-treated mice analyzed for MBP and Iba1/Arg-1 staining at 10 dpl when remyelination is ongoing. The data are included in Figure 5-R1 (panels j-m). Results show that at 10 dpl, flutamide consistently prevents the increase of MBP and Arg-1 expression in the demyelinated area as now indicated in the text (lines 170-172).

2. Given the authors' focus on macrophages/microglia, a disappointing aspect of the story is that macrophage/microglia-specific knockout of AR has no impact on the observed effects on OPC and oligodendrocyte differentiation, even though expression of AR on other CNS cells

appears minimal in female mice. Yet this finding is glossed over in the manuscript. Could the impact of androgens on myelin repair be mediated by peripheral, infiltrating cells?

REPLY: As mentioned in our first reply above, we agree with the Reviewer to say that this aspect of the paper was quite disappointing. As for our other experiments, we increased our animal cohorts in order to more strongly validate the data. The experiment presented in Figure 5 comprised only n=3 animals because of some difficulties in the production of animals. Since the first submission of our manuscript, we succeeded in boosting the animal production and thus performed a new experiment. The pooled data are now presented in Figure 6-R1. Although DHT induced only a lower effect on the percentage of Olig2+ CC1+ cells in the mutant compared to the wild-type animals in the initial version of the manuscript, its effect is now found to be clearly prevented on OPC differentiation supporting the hypothesis that in females the expression of AR in microglia is not only involved in the ability of this cell type to express the anti-inflammatory marker Arg-1, but also in the capacity of OPCs to differentiate into CC1+ oligodendrocytes. The effect of DHT on astrogliosis is also clearly impaired in the mutant. Therefore, we have edited the corresponding text (lines 181-185).

3. The human data is difficult to interpret, largely because the extent of samples from which it is derived is not explained clearly. Supplemental Table 7 provides descriptions of the donors, but it's not clear if all these donors were included in the IHC experiments and how many of each type of lesion were included (i.e., the sample size from which numbers were derived). This should be clarified. Further, does “% of CD68+AR+” cells mean the % of CD68+ cells that are double positive for AR+? This is unclear and should be clarified. Finally, these experiments were not completed because an antibody was discontinued. From the methods, it appears 2 other antibodies were tested unsuccessfully. Given the translational importance of this data, are no other methods possible to complete the study? Perhaps in situ hybridization as was used for the LPC mice?

REPLY: Actually, the number of each type of lesions was indicated in the last column of Supplementary Tables 7 in the first version of the manuscript. However, we have now performed RNAscope analysis of AR in another series of tissues derived from MS patients (4 females and 4 males) and non-MS patients (2 females and 2 males). The data are presented in the new Figure 2-R1. The full description of the patients and corresponding samples (including the type of lesions that were examined are shown in Supplementary Tables 1 and 2. RNAscope data corroborate the data obtained by immunostaining as now described (lines 97-105) in the revised manuscript. In addition, since the submission of our manuscript, we have submitted another manuscript entitled ‘Single nuclei RNA-Seq stratifies multiple sclerosis patients into three distinct white matter glia responses’ by Macnair and collaborators presently available as a Biorxiv (<https://www.biorxiv.org/content/10.1101/2022.04.06.487263v1.article-info>). An interactive web browser to analyse cell-type specific expression levels of genes and transcriptomic changes in MS versus control tissue is available at https://malhotralab.shinyapps.io/MS_broad/ (for broad cell types) and at https://malhotralab.shinyapps.io/MS_fine/ (for fine cell types). AR sex difference can be observed and confirm our preliminary data using AR antibody as shown now in the plot provided as Figure 2d-R1.

4. For all experiments, why are the female mice ovariectomized? The expression of AR was determined in non-ovariectomized LPC mice (Fig 1), but all subsequent interventional experiments are performed in ovariectomized mice. Might not the expression of AR and other physiologic effects of androgens be altered in female mice after ovariectomy?

REPLY: We have chosen to assess the effects of the sexual hormones in gonadectomized females in order to exclude the confounding effects of endogenous gonadal steroid hormones as previously done for our work regarding male animals. We have nevertheless considered the comment of the Reviewer and therefore performed RNAscope analysis for evaluating AR transcription in the LPC lesion from ovariectomized female mice. As shown in Supplementary Figure 15, AR transcription is still detected in Iba1+ cells in the lesion from ovariectomized animals. The text has been modified accordingly (lines 4562454).

Minor point:

1. The number of sections examined per animal for microscopy studies is specified in the methods, but this should be stated in figure legends to make evaluation of rigor easier for the reader.

REPLY: We have now added the number of slices in the different figure legends.

Reviewer #3 (Remarks to the Author):

This is an exhaustive study demonstrating a plethora of data which, in summary, suggest that androgens play in the context of inflammatory, demyelinating disorders, such as Multiple Sclerosis, a major role in females that is critically different from their role in males. To address this interesting hypothesis, the authors use a well characterized model for remyelination (id est the LPC model) and a model of auto-immune driven inflammatory demyelination (id est the EAE model). As there are fundamental biological differences between each gender, it is pivotal to address divergent effects of drugs and potential treatment strategies. This, the topic of the presented study is of high relevance. As a major limitation, the presented study addresses two distinct but fundamentally different aspects of the MS pathology, remyelination and auto-immune driven inflammation. It would have been more convincing to focus on one aspects and try to understand the cellular mechanisms in more detail. Nevertheless, while the study is worth to be published there are several major and minor aspects that need the full attention of the authors. In particular, the following aspects should be addressed:

1. In the material and methods section it is stated that “drugs were administered at the onset of clinical symptoms until Day 30 after immunization.” Was this done per individual animal or was drug treatment started at the same days for the entire cohort? Please specify.

REPLY: The onset of clinical symptoms occurred after the same delay in all animals. Thus, drug treatment was started on the same day for each entire cohort. This is now specified in the Methods (lines 489-490).

2. Some minor typos should be corrected such as “The RNA-seq libraries were prepared using either the NEBNext Ultra II Directional RNA Library Prep 811 Kit (NEB) and sequenced with the Novaseq” (either should be deleted).

REPLY: The text has been edited (line 640).

3. The authors state that “In case of absence of distribution normality, non-parametric tests (Mann-Whitney two-tailed, Kruskal-Wallis with Dunn’s post tests for comparison) were used.” Please state how normal data distribution was evaluated.

REPLY: Two tests have been used including D'Agostino & Pearson normality test and Shapiro-Wilk normality test now indicated in lines 681.

4. The authors state that “At 7 days post-lesion (dpl), when the process of spontaneous remyelination is ongoing and corresponds to the end of OPC recruitment and the beginning of their differentiation...” Please either provide appropriate citations for this statement or demonstrated it with the used samples.

REPLY: Prof. Robin Franklin and collaborators have previously described that ‘5, 10, and 14 days post-lesion (dpl), corresponded to the timing of peak OPC recruitment, initiation of OPC differentiation, and myelin sheath formation, respectively’ (Fancy et al, 2009). However, in our own experiments (Laouarem et al, *Glia*, 2021, Figure 7), we found that at 7 dpl, we could still detect OPC proliferation and already visualize OPC differentiation, hence the sentence used above. Both references have now been added (lines 477).

5. Figure 1B demonstrates a LPC-induced lesions with ongoing remyelination, as stated by the authors. Is this true for the entire lesions or for the lesion rim? Was there any difference of AR-expression throughout the lesion? Beyond, it is stated that “we observed a strong AR upregulation in the lesion from females while AR transcripts could be detected at a much lower level in the lesion from males”. The demonstrated images are not convincing and quantification should be performed. Is the AR-expression induction due to LPC-induced demyelination, or due to the mechanical, needle-induced injury. Vehicle-treated mice would be required to answer this important question.

REPLY: We agree with the Reviewer regarding the fact that images in Figure 1B were not convincing enough. Therefore, we have edited the figure and now show separate channels and overlays in order to better visualize nuclei concentration reflecting the LPC lesion and the high AR transcript signals in the lesion from females but not males. In contrast, the cortex express AR in both females and males. The requested quantification was already provided in the first version, but the histogram was located far from panel 1B. Therefore, we have also modified the panel organization (Figure 1-R1 panels b-d).

The Reviewer is right when he/she speaks of the existence of a rim corresponding to the progressive recruitment of new oligodendrocytes from the outside of the lesion. AR expression was nevertheless found homogeneously distributed throughout the lesion in agreement with its main localization in microglia and not in OPCs/oligodendrocytes.

Finally, we took into consideration the remark of the Reviewer suggesting that AR expression induction might be due to the mechanical, needle-induced injury. In order to investigate this hypothesis, we performed a new experiment including vehicle-treated mice. RNAscope was performed in slices from intact females, which have received either LPC or the vehicle. As shown in Supplementary Figure 15, animals receiving the vehicle display inflammation (as shown by Iba1 expression) but no induction of AR expression. This point is now indicated in line 454.

6. A key step in androgen action is AR nuclear translocation. Can the authors provide evidence that the AR is indeed expressed in the nuclear compartment? The high-power insert in Fig1C rather suggest a perinuclear expression pattern, especially in IBA1+ cells.

REPLY: The Reviewer is right, the images in Figure 1C showed a predominant perinuclear expression pattern. Now, we provide images at a higher magnification leading to detect besides

the perinuclear localization, an immunofluorescent signal in the nucleus (Figure 1f-R1). Because of this unexpected observation, we have also used an antibody directed to DHT ligand (Figure 1g-R1), which leads to a clear nuclear labeling in a wide majority of Iba1+ cells. We can also notice that besides the nuclear DHT+ signals (white arrows in Figure 1g-R1), a perinuclear signal can also be observed (yellow arrowheads in Figure 1g-R1). Our data do not allow to exclude that our polyclonal antibody mostly detect the ligand-unbound AR, which is likely present in the cytoplasm. However, in other cell types out of the CNS, the existence of classical and non-classical pathways of androgen action have been proposed to co-exist. This is true for the Sertoli cells where ligand-bound AR monomers can either migrate to the inner side of the cell membrane and interact with Src, thus activating the non-classical/non-genomic pathway of androgen action or alternatively translocate to the nucleus and form homodimers that can interact with androgen response elements or with other transcription factors, thus activating the classical genomic pathway (Edelsztein and Rey, 2019). The visualization of both extra and intra-nuclear staining by using the DHT-antibody might support the latter hypothesis. However, this remains to be investigated. A few sentences of the discussion indicate these hypothesis (lines 385-389).

7. The authors should clearly state which control experiments were performed to demonstrate the specificity of their stains.

REPLY: Additional data for the Reviewers-Figure 3 provides images obtained with the AR antibody in the ipsilateral and contralateral sides of the LPC lesion showing the exclusive labeling of cortical neurons in the contralateral side compared to the labeling of both the cortical neurons (at a higher level) and the callosal lesion in the ipsilateral side. Additional data for the Reviewers-Figure 3 also shows that the secondary antibody does not lead to any non-specific labeling.

8. The studies using MS tissues appear to be preliminary, and I am not sure whether they add much to the paper with this limited number of investigated cases. In case no reliable antibodies are available the authors could dissect different lesions of cyrosections and perform mRNA expression analyses.

REPLY: We have now performed RNAscope analysis of AR in another series of tissues derived from MS patients (4 females and 4 males) and non-MS patients (2 females and 2 males). The data are presented in the new Figure 2-R1. The full description of the patients and corresponding samples (including the type of lesions that were examined are shown in Supplementary Tables 1 and 2. RNAscope data corroborate the data obtained by immunostaining as now described (lines 97-105) in the revised manuscript. In addition, since the submission of our manuscript, we have submitted another manuscript entitled ‘Single nuclei RNA-Seq stratifies multiple sclerosis patients into three distinct white matter glia responses’ by Macnair and collaborators presently available as a Biorxiv (<https://www.biorxiv.org/content/10.1101/2022.04.06.487263v1.article-info>). An interactive web browser to analyse cell-type specific expression levels of genes and transcriptomic changes in MS versus control tissue is available at https://malhotralab.shinyapps.io/MS_broad/ (for broad cell types) and at https://malhotralab.shinyapps.io/MS_fine/ (for fine cell types). AR sex difference can be observed and they confirm our preliminary data using AR antibody as shown now in the plot provided as Figure 2d-R1.

9. Again, in figure 2 it remains unclear whether the observed effects of testosterone and DHT on astrocytes and microglia are linked to the LPC-induced demyelination or the mechanical injury induced during the LPC-application. Sham-operated groups would be required.

REPLY: In order to investigate the hypothesis, we performed GFAP and Iba1 staining on slices derived from intact females stereotaxically injected with PBS. As shown in Additional data for the Reviewers-Figure 4, the mechanical injury does not promote any substantial astrogliosis and/or microgliosis comparable to the one induced by LPC as shown in the present manuscript.

10. In the text it is stated that aromatase to be upregulated in the lesion from female mice. The corresponding figure demonstrates indeed aromatase expression but whether this is due to the LPC-induced demyelination remains unclear. Again, sham-operated mice would be supportive.

REPLY: In order to investigate the hypothesis, we performed aromatase staining on slices derived from intact females stereotaxically injected with PBS. As shown in Supplementary Figure 2, the mechanical injury does not promote a substantial increase in aromatase expression comparable to the one induced by LPC. This observation is indicated in line 146.

11. The authors state that “flutamide-treated animals displayed a significant decrease in the percentage of OPCs that are able to differentiate into CC1+ oligodendrocytes compared to the vehicle condition”. This sentence appears misleading. As the percentage of OLIG2/CC1 double positive cells in relation to the entire OLIG2 cell population is lower in flutamide-treated mice, this would mean less cells mature under flutamide treatment. Please rephrase.

REPLY: The Reviewer is right, we have rephrased (lines 167-168).

12. In some cases, myelination is estimated by anti-PLP, in other by anti-MBP stains. The authors should either comment on this discrepancy or perform both stains for all the subexperiments.

REPLY: We agree with the Reviewer and thus have performed MBP staining for the panel f in Figure 3-R1 instead of PLP. The result obtained with PLP is similar to the one obtained with MBP.

13. The authors state that microglia are the main AR-expressing cells in the CNS. Did the authors consider that a significant proportion of these cells are IBA1+ recruited monocytes in the LPC model.

REPLY: Indeed, we cannot exclude that a restricted number of macrophages could have infiltrated the LPC-induced lesion as previously shown for T cells (Ghasemlou et al, 2007) and as proposed by (El Wali et al, 2020). Consequently, we have edited the text (line 84).

14. Figure 6K should be Claudin5+ area instead of Claudin+ area.

REPLY: The figure 7k-R1 has been edited.

15. The presented results using the EAE model are somewhat irritating. First, the authors should try to focus on the most relevant findings and move the less relevant ones into the supplements. Second, their FACS analysis clearly demonstrate that the observed protective effects of DHT are at least in part due to immunosuppressive functions. For example, proportions of CD4+ T cells as well as the proinflammatory Th1 and Th17 cells are lower in the secondary lymphoid organs in DHT-treated compared to Vehicle-treated mice. However, the main focus of this manuscript so far was induction of remyelination. It is not clear to me how these data help to strengthen the so far observed pro-myelinating effects. I rather would suggest to verify the pro-myelinating effects in another model of remyelination (such as the cuprizone model) or to start treatment during the chronic phase of the EAE disease when the lesions are fully established.

REPLY: We understand the comment of the Reviewer. However, the discrepancy observed at the level of the inflammatory cells (microglia, macrophages, astrocytes) between females and males let us consider that those cells have a major importance in the ability of androgens to promote remyelination and thus should not be set aside. The higher infiltration of macrophages in females compared to males may also be a critical point in the remyelination process. We have now included more clearly in the text the requirement for considering remyelination in the context of peripheral immune cell infiltration as it occurs in MS (lines 188-189). Regarding the use of another model allowing to support once more the ability of androgens to induce remyelination, we preferred the LPC model since the cuprizone model had been already used to show the ability of androgens (even though it was not DHT but testosterone) to promote MBP staining increase in females (Hussain et al, 2013).

16. I am not sure if the NGS data add much to the manuscript. Since bulk RNA sequencing was performed, it is hard to assign the observed expressional changes to a specific cell type.

REPLY: Actually, our bulk RNA-Seq experiment validates and extends the phenotypic characterization made by different approaches throughout the manuscript including immunofluorescence or cytokine determination. Indeed, as a whole, bulk RNA-Seq data revealed the substantially higher ability of DHT to downregulate activated profiles of microglia and astrocytes in demyelinated females than in demyelinated males. In agreement with this finding, 1) AR expression is detected mostly in microglia and to a lower extent in astrocytes in the demyelinated lesions from females, but not in the demyelinated lesions from males in which AR expression is quite below the detection threshold (Fig. 1-R1); 2) Among the genes related to microglia activation and shown to be downregulated in our RNA-Seq analysis of DHT-treated females but not DHT-treated males (Supplementary Tables 5-R1 and 7-R1), the downregulation of *Illb* in females is validated by the decrease of IL-1 β levels in the spinal cord from DHT-treated females, but not DHT-treated males (Fig. 8f-R1). Additionally, we have also validated the downregulation of *Tnf* (exclusively detected in microglia in the EAE model as shown in Supplementary Figure 14-R1) and *Csf2* genes by using quantitative RT-PCR amplification of mRNA from DHT-treated females Supplementary Fig. 14-R1). 3) In the same line, the downregulation of the well-known marker of reactive astrocytes STAT3 is only observed in RNA-Seq analysis from DHT-treated females, which is validated by the visualization of GFAP+STAT3+ immunofluorescence decrease in spinal cord slices from DHT-treated females (Fig. 3j, k-R1). Still validating this RNA-Seq data, we previously published that in the presence of the aromatase inhibitor fadrozole, testosterone was unable to decrease GFAP+ STAT3+ labelling in LPC demyelinated lesions from males (Laouarem et al, 2021). For all these reasons, we consider that our RNA-Seq data are consistent with our slice immunolabeling and flow cytometry experiments. All these complementary data make our statements more convincing.

Additional references :

- Edelsztein NY, Rey RA (2019) Importance of the Androgen Receptor Signaling in Gene Transactivation and Transrepression for Pubertal Maturation of the Testis. *Cells* 8.
- Falcao AM, van Bruggen D, Marques S, Meijer M, Jakel S, Agirre E, Samudyata, Floriddia EM, Vanichkina DP, Ffrench-Constant C, Williams A, Guerreiro-Cacais AO, Castelo-Branco G (2018) Disease-specific oligodendrocyte lineage cells arise in multiple sclerosis. *Nature medicine* 24:1837-1844.
- Li B, Qing T, Zhu J, Wen Z, Yu Y, Fukumura R, Zheng Y, Gondo Y, Shi L (2017) A Comprehensive Mouse Transcriptomic BodyMap across 17 Tissues by RNA-seq. *Scientific reports* 7:4200.
- Risso D, Ngai J, Speed TP, Dudoit S (2014) Normalization of RNA-seq data using factor analysis of control genes or samples. *Nat Biotechnol* 32:896-902.

REVIEWER COMMENTS

Reviewer #1 (Remarks to the Author):

Comments after first revision:

The authors have made improvements to the manuscript that subsided most of the concerns raised. Nonetheless, the following points should be improved:

Major:

1. Human cohort:

- Despite the increase in data size and the new measurements provided, the cohorts, especially control, are still very small to address the within control/disease group sex comparison.

2. RNAseq analysis:

- Despite the additional information of the scoring method, it is still unclear how the scoring was performed. The authors explain in the rebuttal the scoring is based on a work that is in preparation in another lab, and provide partial information for the review of this manuscript. It is not possible to assess the quality of the RNAseq analysis work in this manner, without knowing the papers that were used for the curation, and the full details of the scoring scheme. It is therefore an absolute requirement that the authors only provide results based on already published and established methods, such as GO enrichment, or provide additional analysis that can be properly reviewed and accepted. At the moment the scoring method given cannot be reproduced or verified. It is also disturbing that the RNA-seq data being presented in this manuscript are supposed to be published in a different paper, for which it remains unclear, whether this will happen in a timely fashion. Following FAIR principles the data have to be published with this manuscript or the authors wait until the other paper is published and then reference it. Choose either way, but as suggested, it cannot be accepted.
- The authors used different gene sets that represent the main subpopulations of microglia. In order to assess the gene set behavior in the different groups, the authors are requested to perform the results of an enrichment analysis for these gene sets in the different groups and indicate clearly in the text the changes that were observed. At the moment the methodology is not optimal and the description in the text is unclear.
- Analysis of scRNA-seq data: not clear which tool was used for the analysis. Please provide further information.
- The current results presented from the scRNA-seq data do not resolve the cell number bias that might be present. Therefore, the authors should provide better evidence, such as deconvolution analysis using the scRNA-seq from Falcao et al.).
- Why are the RNA-seq analyses for male and female (rebuttal, page 4 first paragraph) are only for the reviewers? The should be included in the paper.

Minor:

3. Lines 265-266: vehicle-treated females displayed as much microglia as macrophages whereas males displayed predominant microglia compared to macrophages – since this is not directly shown in the same plot, readers will benefit by specifying the values in the text to make this comparison clearer.
4. Line 105: please add: “compared to males” – please provide the statistical test performed and values.
5. Supplementary table 3- please provide an index for abbreviation of groups.
6. Line 300: out of place +
7. Figure S12: indicate in legend the y and x axis more clearly. Without the figure provided only to reviewers this plot is not very clear.

Reviewer #2 (Remarks to the Author):

The authors have performed an impressive amount of new work which, in my view,

sufficiently addresses the major concerns from the original manuscript. As a result, the conclusions of the paper are now adequately supported by the presented data. In particular, my concerns about the interpretability of the human data and the cellular target of androgens have been beautifully addressed. I believe this is now a well-supported piece of scholarship that will be of clear interest to the field of neuroimmunology.

Reviewer #3 (Remarks to the Author):

1. The authors claim that "Because of this unexpected observation, we have also used an antibody directed to DHT ligand (Figure 1g-R1), which leads to a clear nuclear labeling in a wide majority of Iba1+ cells. We can also notice that besides the nuclear DHT+ signals (white arrows in Figure 1g-R1), a perinuclear signal can also be observed (yellow arrowheads in Figure 1g-R1)." I would recommend to use, for this claim, a higher magnification, essentially the same provided in figure 1g. Beyond, nuclear AR localization in figure 1g is still not convincing for me. Unbiased quantification of the proposed co-localization would be an elegant option.

2. The authors state "Additional data for the Reviewers-Figure 3 also shows that the secondary antibody does not lead to any non-specific labeling." Please indicate where in figure 3 this is shown.

3. Early in the p2p response the authors state that "As shown in Supplementary Figure 15, animals receiving the vehicle display inflammation (as shown by Iba1 expression) but no induction of AR expression. This point is now indicated in line 454." Later on, it is stated that "As shown in Additional data for the Reviewers-Figure 4, the mechanical injury does not promote any substantial astrogliosis and/or microgliosis comparable to the one induced by LPC as shown in the present manuscript:" These contradictors statements are somewhat confusing. Please clarify.

4. The authors state that "At 7dpl, flutamide significantly decreased the number of OPCs and Olig2+ cells as well as the percentage of Olig2+ cells differentiated into CC1+ oligodendrocytes (Fig. 5b-e)". One cannot state that olig2+/cc1+ cells are derived from olig2+/cc1- cells. Thus, the last statement of the sentence is speculative. Beyond, if there are less OPC, less OLIG2+ cells and less mature oligodendrocytes, would that mean that flutamide treatment induced OPC death? If so, is there any data supporting such a scenario?

5. The authors correctly state that "testosterone or DHT-treated females displayed significantly lower scores throughout the whole experiment". In their rebuttal letter, the authors argue that peripheral immune cells might play a role during myelin repair in their model. While this might well be true, this claim is not substantiated by data. Beyond, DHT and Testosterone-treated EAE mice show a milder EAE disease score days after the occurrence of first symptoms, which is maybe a bit too fast for remyelination-mediated effects.

REVIEWER COMMENTS

We thank the three Reviewers for their constructive comments. In our detailed point by point answers below, we have addressed the concerns raised by Reviewers 1 and 3. All corrections are visible in red throughout R2 revised version of the manuscript.

Reviewer #1 (Remarks to the Author):

1.Human cohort: Despite the increase in data size and the new measurements provided, the cohorts, especially control, are still very small to address the within control/disease group sex comparison.

REPLY: Indeed, the number of human tissues was still low. Therefore, we performed new RNAscope experiments including n=5 male controls and n=5 female controls, n=5 male MS and n=6 female MS. Figure 2 and the corresponding legend as well as Supplementary Table 1 have been accordingly updated.

2.RNAseq analysis:

Despite the additional information of the scoring method, it is still unclear how the scoring was performed. The authors explain in the rebuttal the scoring is based on a work that is in preparation in another lab, and provide partial information for the review of this manuscript. It is not possible to assess the quality of the RNAseq analysis work in this manner, without knowing the papers that were used for the curation, and the full details of the scoring scheme. It is therefore an absolute requirement that the authors only provide results based on already published and established methods, such as GO enrichment, or provide additional analysis that can be properly reviewed and accepted. At the moment the scoring method given cannot be reproduced or verified. It is also disturbing that the RNA-seq data being presented in this manuscript are supposed to be published in a different paper, for which it remains unclear, whether this will happen in a timely fashion. Following FAIR principles the data have to be published with this manuscript or the authors wait until the other paper is published and then reference it. Choose either way, but as suggested, it cannot be accepted.

REPLY: In agreement with the Reviewer's comment, we have decided to release now the OligoScore (<https://oligoscore-staging.icm-institute.org/>) resource, an open resource to the community, to provide access to it in this paper, and have described the approach in Methods section (lines 668-677). Furthermore, we now provide as supplementary data, four Excel tables and histograms with the detailed analyses based on OligoScore curation strategy of scoring the deregulated genes for their implication in the different processes of oligodendrogenesis (Supplementary Tables 4-7). Tables 4 and 5 (Tab 1) contain all curated genes (n=391, at the moment when the analysis was performed) and for which the fold changes between DHT-treated versus Vehicle-treated data are shown for females and males. Tables 6 and 7 (Tab 1) contain only the deregulated genes (FDR correction is < 0.05) being curated (211 in females and 95 in males). These Supplementary Tables are now cited in the results line 329.

- The authors used different gene sets that represent the main subpopulations of microglia. In order to assess the gene set behavior in the different groups, the authors are requested to perform the results of an enrichment analysis for these gene sets in the different groups and indicate clearly in the text the changes that were observed. At the moment the methodology is not optimal and the description in the text is unclear.

REPLY : Following the suggestion of the reviewer, we have performed gene set enrichment analysis (GSEA), using Cluster Profiler in R, with all the microglial genes combined (105 genes) ordered accordingly to the logFC in females (DTHFvsCTF) or males (DTHMvsCTM) comparisons. This GSEA analysis shows an enrichment of many (219) gene sets in DTH-treated females (compared to their controls), most of them (208) suppressed (genes being downregulated), and many related to immune and inflammatory processes, including 'lymphocyte mediated immunity', 'response to stress', 'defense response', 'immune system process', 'immune response', and 'cytokine production'. On the contrary, in males only two GSEA processes were activated ('translation' and 'peptide biosynthetic process') and none were suppressed.

We have added the following paragraph in the results (lines 374-383) and the data in Supplementary Fig. 16 and Supplementary Table 12:

'By combining all microglial gene sets (105 genes), we performed gene set enrichment analysis (GSEA) with these genes ordered by their changes in expression either in female or male comparisons (DTH-treated vs. non treated). In line with previous results, this GSEA analysis showed large enrichment of many gene sets in DTH-treated females but almost none in males, with many of the suppressed gene sets (genes being downregulated) related to immune and inflammatory processes, including 'lymphocyte mediated immunity', 'response to stress', 'defense response', 'immune system process', 'immune response', and 'cytokine production' (Supplementary figure 16; Supplementary Table 12).'

We have also listed the sets of genes down-regulated in microglia in order to make the text clearer (lines 352-358 and 361-364).

Finally, we have described the approach (lines 678-685) in the Methods as follows:

'Gene set enrichment analysis (GSEA). We used *gseGO* function of *Cluster profiler* R package to find gene sets enriched in the gene list of 105 microglial genes (Supplementary Table 12) ranged by the differential expression (logarithmic fold change, logFC) in DTH-treated vs non- treated females and males, respectively. We found 219 gene set enriched in females but only 2 in males. Dotplot and gseaplot were used for visualization of enriched gene sets. All gene sets enriched are provided in Supplementary Table 12. R script has been deposited in <https://github.com/ParrasLab/Androgen-signaling-and-remyelination-Nat-Commun-paper>.'

- Analysis of **scRNA-seq data**: not clear which tool was used for the analysis. Please provide further information.

REPLY : We have now provided further information in the Methods (lines 686-694) as follows :

'scRNA-seq analysis. EAERaw.RData object was obtained from Gonçalo Castelo-Branco's lab and processed in R (4.0) using the following packages: *Seurat* (3.0) for data processing and *ggplot2* for graphical plots. Seurat objects were first generated using *CreateSeuratObject* function (min.cells = 5, min.features = 100). Normalized with *sctransform* function. Cell neighbors and clusters were found using *FindNeighbors* (dims = 1:30) and *FindClusters* (resolution = 0.8) functions. *RunPCA*, and *RunUMAP* functions with default parameters. Clusters were annotated based on cell-subtype markers as detailed in the R script, which has been deposited in <https://github.com/ParrasLab/Androgen-signaling-and-remyelination-Nat-Commun-paper>.

- The current results presented from **the scRNA-seq data** do not resolve the cell number bias that might be present. Therefore, the authors should provide better evidence, such as deconvolution analysis using the scRNA-seq from Falcao et al.

REPLY: In agreement with the suggestion of the reviewer, we have used CIBERSORTx (<https://doi.org/10.1038/s41587-019-0114-2>) and two scRNA-seq datasets: Falcao & Castelo-Branco (GSE113973, not having neurons or astrocytes in it) and the Meijer & Castelo-Branco (GSE166179, not having neurons but containing an astrocyte cluster and only one microglial cluster downloaded from <https://cells.ucsc.edu/>). We obtained similar results, indicating similar number of microglial cells/clusters in males (DTH-treated or not), while in DTH-treated females the deconvolution varied cell proportions in microglial and immune OL/OPC clusters, likely due to the downregulation of the immune related genes. In the present version of the paper, we present the results obtained by deconvolution with Falcao & Castelo-Branco's dataset and refer to it as follows:

In the results section (lines 383-393):

'To try to exclude bias putatively related to changes in cell numbers between conditions, we used CIBERSORTx, a machine learning method to determine cell type abundance and expression from bulk tissues (<https://doi.org/10.1038/s41587-019-0114-2>), together with a single cell RNA-Seq dataset from mouse EAE model (Falcao et al, 2018 Nat Med. 2019 Apr 26; 24(12): 1837–1844. doi: [10.1038/s41591-018-0236-y](https://doi.org/10.1038/s41591-018-0236-y)) (GSE113973). This deconvolution of our bulk-RNA-Seq datasets suggested that while microglial clusters did not change in proportions upon DTH-treatment in males, DTH-treated females presented some changes in microglial clusters and EAE immune-OL/OPC clusters model (Falcao et al, 2018 Nat Med. 2019 Apr 26; 24(12): 1837–1844. doi: [10.1038/s41591-018-0236-y](https://doi.org/10.1038/s41591-018-0236-y)) (Supplementary Table 13), likely due to the abovementioned dysregulation of microglial/inflammatory genes. Indeed, 21 genes out of 31 downregulated genes only in DTH-treated females are expressed in the EAE microglial cells from this scRNA-Seq dataset (Supplementary Fig. 15)'.

And in the Supplementary Methods :

Bulk RNA-seq deconvolution. We used CIBERSORTx tool (<https://doi.org/10.1038/s41587-019-0114-2>) on the docker module Cibersortx/fractions, with 100 permutations as input parameter, in order to deconvolute our bulk RNA-Seq datasets obtained from EAE spinal cord samples. The signature of scRNA-Seq matrix was generated according to the book methods described by Steen et al (https://doi.org/10.1007/978-1-0716-0301-7_7) with the GSE113973 public scRNA-Seq dataset from mouse EAE model. The deconvolution analysis was performed on two mixture files corresponding to the RNA-Seq count matrices generated as described before, containing females and males' comparisons, 'DHTFvsCTF' and 'DHTMvsCTM', respectively. The results obtained are estimated as the proportions of each cell types in each RNA-Seq sample inferred from the prior knowledge of the scRNA-Seq sample. R script has been deposited in <https://github.com/ParrasLab/Androgen-signaling-and-remyelination-Nat-Commun-paper>.

- Why are the RNA-seq analyses for male and female (rebuttal, page 4 first paragraph) only for the reviewers? They should be included in the paper.

REPLY: We now included the Additional data for the Reviewers - Figure 1a,b as 'Supplementary Figure 11'. This figure has been mentioned in the text (line 316).

Minor:

3. Lines 265-266: vehicle-treated females displayed as much microglia as macrophages whereas males displayed predominant microglia compared to macrophages – since this is not directly shown in the same plot, readers will benefit by specifying the values in the text to.

REPLY : In agreement with the Reviewer request, we have now specified the values in the text (lines 266-268).

0. Line 105: please add: “compared to males” – please provide the statistical test performed and values.

REPLY : According to the request of the Editor, we have withdrawn Figure 2d (described in line 105) and moved this panel as Supplementary Figure 1 panel I in support of our own data concerning the higher expression of AR in female microglia compared to males. This graph was built by analyzing the freely available and searchable data using the shiny app provided in the paper as an open resource to the community (https://malhotralab.shinyapps.io/MS_broad/). Since statistics were not tested as there are not enough cells to do it robustly, the sentence regarding Supplementary Figure 1 panel I was modified as follows : ‘Moreover, AR mRNA expression in microglia from MS and control donors from a publicly available single-nuclei RNA sequencing database appeared higher in MS female samples compared to males (Supplementary Fig. 1 panel I)’ (lines 112-115).

1. Supplementary table 3- please provide an index for abbreviation of groups.

REPLY: Done.

2. Line 300: out of place +

REPLY : Done.

3. Figure S12: indicate in legend the y and x axis more clearly. Without the figure provided only to reviewers this plot is not very clear.

REPLY: As mentioned above (point 1 of Comment 2), we have now included in the Supplementary data of the revised version R2 the figures that were previously provided only to reviewers in the revised version R1. This should clarify Supplementary Figure 12-R1 (which has become Supplementary Figure 13-R2). In the Supplementary Figure 13-R2 (like in Figure 10) all the bar plots have the title of the x-axis just above the axis. In the ‘y-axis’, if one want to consider this an axis, it is written the oligodendrogenesis processes labeled in each bar (the same in all graphics). The legend of Supplementary Figure 13-R2 has been accordingly edited.

Reviewer #2 : The authors have performed an impressive amount of new work which, in my view, sufficiently addresses the major concerns from the original manuscript. As a result, the conclusions of the paper are now adequately supported by the presented data. In particular, my concerns about the interpretability of the human data and the cellular target of androgens have been beautifully addressed. I believe this is now a well-supported piece of scholarship that will be of clear interest to the field of neuroimmunology.

We thank the Reviewer for his/her comments regarding the R1 revised version.

Reviewer #3 (Remarks to the Author):

1. The authors claim that “Because of this unexpected observation, we have also used an antibody directed to DHT ligand (Figure 1g-R1), which leads to a clear nuclear labeling in a wide majority

of Iba1+ cells. We can also notice that besides the nuclear DHT+ signals (white arrows in Figure 1g-R1), a perinuclear signal can also be observed (yellow arrowheads in Figure 1g-R1).” I would recommend to use, for this claim, a higher magnification, essentially the same provided in figure 1g. Beyond, nuclear AR localization in figure 1g is still not convincing for me. Unbiased quantification of the proposed co-localization would be an elegant option.

REPLY: As recommended by the reviewer, we provided a higher magnification for the colocalization of DHT and Iba1. These new images allow the visualization of both nuclear (white arrows) and perinuclear (white arrowheads) DHT immunostaining in Iba1-expressing cells. We have also indicated DHT-expressing cells, which do not co-express Iba1 (yellow arrowheads) and cells expressing neither DHT nor Iba1 (yellow arrows). In addition, as also recommended by the reviewer, we quantified the percentage of Iba1-expressing cells displaying a nuclear versus a perinuclear DHT labeling (Fig. 1hR2), which indicates a higher proportion of nuclear DHT staining. Fig. 1h has been added in the results (line 87) and the figure legend has been edited accordingly.

2. The authors state “Additional data for the Reviewers-Figure 3 also shows that the secondary antibody does not lead to any non-specific labeling.” Please indicate where in figure 3 this is shown.

REPLY: In the right panel of the Additional Figure 3 for the reviewers, the secondary antibody (AbII) shown in the ipsilateral side does not lead to any fluorescent signal neither in the lesion (delineated by the dashed line) nor in the cerebral cortex (Cx) whereas the AR antibody similarly used in the ipsilateral side (left panel) lead to clear signals detected both in the lesion and above the lesion in the cerebral cortex.

3. Early in the p2p response the authors state that “As shown in Supplementary Figure 15, animals receiving the vehicle display inflammation (as shown by Iba1 expression) but no induction of AR expression. This point is now indicated in line 454.” Later on, it is stated that “As shown in Additional data for the Reviewers-Figure 4, the mechanical injury does not promote any substantial astrogliosis and/or microgliosis comparable to the one induced by LPC as shown in the present manuscript:” These contradictors statements are somewhat confusing. Please clarify.

REPLY: The Reviewer is right. Our text was not very clear. What we mean is that the stereotactic injection of the vehicle - instead of LPC - leads to a limited Iba1+ inflammatory process (due to the mechanical injury) as shown in supplementary Figure 15-R1 (which has become Supplementary Figure 17-R2) right panel as well as in the Additional Figure 4 for the reviewers, right panel. In both figures, Iba1+ immunofluorescent signal observed in the vehicle-treated animals is much lower than the Iba1+ signal observed after LPC injection (left and middle panels in Supplementary Figure 17-R2).

4. The authors state that “At 7dpl, flutamide significantly decreased the number of OPCs and Olig2+ cells as well as the percentage of Olig2+ cells differentiated into CC1+ oligodendrocytes (Fig. 5b-e)”. One cannot state that olig2+/cc1+ cells are derived from olig2+/cc1- cells. Thus, the last statement of the sentence is speculative. Beyond, if there are less OPC, less OLIG2+ cells and less more mature oligodendrocytes, would that mean that flutamide treatment induced OPC death? If so, is there any data supporting such a scenario?

REPLY: We edited our text in order to take into account the comment of the Reviewer and thus removed the statement that is speculative (lines 168-170). Regarding the hypothesis that

flutamide might induce OPC death, there is no supporting data. 7 dpl is not the appropriate time point to investigate this hypothesis. However, since DHT was previously reported to increase the survival of new neurons in the dentate gyrus, an effect blocked by flutamide (Hamson et al, 2013, Endocrinology 154 DOI:10.1210/en.2013-1129), this hypothesis would merit to be investigated in the future via a more accurate analysis of DHT effect on the expression of survival markers in the presence or absence of flutamide at much earlier time points after LPC injection.

5. The authors correctly state that “testosterone or DHT-treated females displayed significantly lower scores throughout the whole experiment”. In their rebuttal letter, the authors argue that peripheral immune cells might play a role during myelin repair in their model. While this might well be true, this claim is not substantiated by data. Beyond, DHT and Testosterone-treated EAE mice show a milder EAE disease score days after the occurrence of first symptoms, which is maybe a bit too fast for remyelination-mediated effects.

REPLY: Actually, we wanted to say that even though EAE is not suitable for studying the remyelination process, it is important to consider demyelination / remyelination both in the context of immune-mediated and nonimmune-mediated animal models of CNS demyelination. In the former, phagocytes and T cells are known to be included in a vicious circle where the pro-inflammatory phagocytes promote the activation of T cells while conversely T cells induce myeloid cells to become pro-inflammatory giving rise to a microenvironment detrimental for myelin regeneration (Codarri et al, Nat Immunol, 2011; <https://doi.org/10.1038/ni.2027>). In addition, it was proposed that some subsets of macrophages actively participate in the destructive demyelination process (Croxford et al, Trends Immunol, 2015; doi: 10.1016/j.it.2015.08.004). This is the reason why we introduced a short sentence in our R1 revision indicating that ‘remyelination cannot be considered independently of the peripheral immune process characterizing MS’ (lines 188-189). We agree with the reviewer to say that the milder EAE disease scores induced by DHT and testosterone at 8 days after the onset of the neurological symptoms is likely the result of the decrease in deleterious T cells and cytokines (as shown by our FACS analyses and cytokine dosages). However, these milder scores may also be related to remyelination since our GO analyses identified DHT-induced upregulation of genes involved in (re)myelination.

REVIEWERS' COMMENTS

Reviewer #1 (Remarks to the Author):

The manuscript presents very important findings concerning sex differences in context of androgen signaling and demyelination within the CNS. The revisions have made this manuscript much stronger. Along these lines, this revision further improved the manuscript, and now only very few points need to be addressed.

1.Line 74: DHT mentioned here the first time. Abbreviation needs to be introduced here

2.The manuscript would benefit greatly from a schematic figure (graphical abstract) summarizing the many findings that differ between females and males.

Reviewer #3 (Remarks to the Author):

The authors have addressed my concerns, I can, thus, recommend publication of this nice work.

REVIEWER COMMENTS_R1

Reviewer #1 (Remarks to the Author): The work by Zahaf et al. presents new results on the action of androgens via the AR on inflammation and remyelinating processes in MS in females and the differences compared to males.

General comments: Comparison of males vs females is done mostly by discussion in the text, while results are almost always presented separately in the plots. This presentation of results does not provide the reader with the ability to have a proper comparison in mind. Further, there is also not always proper statistics for the comparison of the results between males and females.

REPLY: We agree with the Reviewer that the new data comparing males vs females provided in the present work, are inadequately illustrated. In particular, this is true for Figures 7 and 8 (immune cell sorting / cytokine analysis), Figure 9 (parenchymal inflammatory cells) and Figure 10 (RNA-Seq data). **First**, Figures 7 and 8 have been replaced in the present Revised version 1 (R1) by a single figure (Figure 8-R1), which gathers cell sorting and cytokine data from females and males. This new presentation provides evidence for statistically significant differences in the regulation of several types of immune cells and cytokines in DHT-treated females, but not in DHT-treated males. In addition, we provide as Supplementary Figure 9-R1 the direct comparison between female and male immune cell type proportions under the Vehicle- or DHT conditions. **Second**, data characterizing microglia/macrophages and astrocytes in the spinal cord from EAE females and males were already gathered in a single figure in the initial version of the manuscript (Figure 9). This figure is still called Figure 9-R1. However, we have now added Supplementary Figure 10-R1), which provides proper statistics for the direct comparison of microglia/macrophages and astrocytes in females versus males under either vehicle- or DHT-treatment. **Third**, regarding RNA-Seq data, we provide now single Supplementary Tables that display the data obtained in females versus males in particular for all deregulated genes (Supplementary Table 3-R1), ontology analysis (Supplementary Table 4-R1), genes specifically deregulated in microglia and astrocytes (Supplementary Tables 5-7-R1).

Major comments: Overall, the study shows experiments comparing control versus experimental groups with 4 to 5 animals per group. In some of the experiments, the values are showing high intragroup variance. Since the statements might be rather impactful, validation cohorts are necessary throughout the manuscript. Even better would be to provide evidence in other mouse strains for those experiments that do not rely on strain specific settings, e.g. knockout animals or the EAE model. Nevertheless, in these cases validation experiments using the same mouse strain showing the same findings are required.

REPLY: As requested by the Reviewer, we have performed validation experiments for all the experimental protocols using either LPC or EAE demyelination in order to increase our animal cohorts. Consequently, we have edited the corresponding data (in Results and Figures) throughout the whole manuscript. In addition, we have used another mouse strain (129X1/SvJ) and provide now evidence that the pharmacological blockade of AR by flutamide impedes myelin regeneration in 129X1/SvJ strain as it does in C57BL/6 mice used in our work. Data associated with 129X1/SvJ strain are now available in Supplementary Figure 3-R1.

The human data are not adequate. While it is certainly an unfortunate situation, if a reagent is not available anymore during the performance of a project, as the data are presented at the moment they cannot be really interpreted, since the number of patients studied is too small to make general statements (as they are currently in the manuscript). The link to the human is

critical, to avoid a murine model artifact concerning the role of AR in MS. Also, the impact of the findings in the murine system without a link to human disease is an important asset of the manuscript, but requires to be significantly improved. With the lack of the antibody, an alternative approach could be spatial transcriptomics with subcellular resolution. This actually would also allow to measure the other molecules presented in the murine model.

REPLY : The Reviewer is right. Since the submission of our manuscript, we have submitted another manuscript entitled 'Single nuclei RNA-Seq stratifies multiple sclerosis patients into three distinct white matter glia responses' by Macnair and collaborators presently available as a Biorxiv (<https://www.biorxiv.org/content/10.1101/2022.04.06.487263v1.article-info>). An interactive web browser to analyse cell-type specific expression levels of genes and transcriptomic changes in MS versus control tissue is available at https://malhotralab.shinyapps.io/MS_broad/ (for broad cell types) and at https://malhotralab.shinyapps.io/MS_fine/ (for fine cell types). AR sex difference can be observed and confirm our preliminary data using AR antibody (Supplementary Figure 1-R1) as shown now in the plot provided as Figure 2d-R1.

In addition, we have now performed new RNAscope experiments by using an AR RNAscope probe on human tissues. The data have now been included in the new Figure 2-R1. Results are described (lines 97-105). Corresponding Methods have been added (lines 549-575).

Along these lines the bulk transcriptomics is somewhat confusing. The use of gene terms derived from a study that is still in preparation (line 417) is not appropriate without exactly knowing what is really done by this approach.

REPLY : We agree with the Reviewer that several aspects of our bulk transcriptomic analysis require to be clarified. This is namely the case for the scoring procedure aimed at assessing the impact of DHT on the process of oligodendrogenesis. Therefore, we now provide four .xlsx tables and histograms with the detailed analyses based on our curation strategy of scoring the deregulated genes implicated in different aspects (processes) of oligodendrogenesis. These data are for the exclusive use of the Reviewers. They are called 'Additional data for the Reviewers_Tables 1-4). Additional data for the Reviewers_Tables 1 and 2 (Tab 1) correspond to all genes (n=391) selected according to the approach now described in Methods (lines 660-667) and for which the fold changes between DHT-treated data versus Vehicle-treated data are shown for females and males, respectively. Additional data for the Reviewers_Tables 3 and 4 (Tab 1) correspond to the deregulated genes (211 in females and 95 in males) for which the value of the False Discovery Rate (FDR) correction (that is considered to be an indicator of the strength of a study) is < 0.05. The full curation dataset with the corresponding references for each gene will be published in another study in preparation by the laboratory of Dr Carlos Parras and made accessible through a website resource for the scientific community. Tab 2 of each .xlsx file presents graphs derived from the above data. The four graphs visualize in females and males: 1) The total score characterizing each oligodendroglial process; 2) The number of genes involved in each process; 3) The score of genes promoting and inhibiting each process; 4) The number of genes promoting and inhibiting each process.

All these analyses and the corresponding graphs have been now performed after normalization of the female and male data according to the request of the Reviewer (see below). Consequently, we have updated the 'oligodendrogenesis' histograms (Figure 10d,e-R1), added the new Supplementary Figure 12-R1; see below for details) and edited the Results (lines 323-334).

Further, the text is not very clear about the similarities and differences of male versus female animals in the experimental setting.

REPLY : In order to better describe the similarities and differences of male versus female animals in the experimental setting, as mentioned above, we provide now :

- **RNA-Seq data** by systematically comparing females and males in a single table with statistics analyses of the differences between DHT and Vehicle (control) females, DHT and Vehicle males, DHT females and the normalized female/male Vehicle, DHT males and the normalized female/male Vehicle, DHT females and DHT males, Vehicle females and Vehicle males. Thus, 'Supplementary Table 3-R1_DEGs now replaces 'Tables 1 and 2_DEGs_DHT_cpm_EdgeR_FDR005' from the initial version of the manuscript. 'Supplementary Table 4-R1 now replaces 'Tables 3 and 4_GO_DHT_EdgeR_FDR005' from the initial version of the manuscript. Supplementary Tables 5-R1 and Table 6-R1 replace Table 5_microglia_genes and Table 6_astroglia_genes, respectively. The new 'Supplementary Table 7-R1' shows genes deregulated either in both females and males or only in females and only in males.
- **Data for oligodendroglial processes** in the Supplementary Figure 12-R1 displaying the scores and number of genes promoting or inhibiting these processes in females and in males as well as the list of genes deregulated in the myelination process according to the sex of the animals.
- **Cell sorting and cytokine data** in a single figure (Figure 8-R1) allowing a direct comparison between female and male DHT effects as well as in the Supplementary Figure 9-R1 comparing the proportions of immune cells and the levels of cytokines under Vehicle or DHT conditions in females versus males.
- **A direct comparison of microgliosis and astrogliosis** in females and males from the EAE model at 14 dpi, including statistics between Vehicle-treated females and males on one side and DHT-treated females and males on the other side as presented in Supplementary Figure 10.

Also as the data are presented, it has not been clear, whether the male and female data were analyzed as one dataset or independent datasets. If treated as different datasets, comparisons as they have been described in the paper are not valid, since confounding factors are not adequately addressed.

REPLY: The experiments of DHT administration were done first for females and then for males, given the amount of efforts and resources needed for these EAE experiments. In order to take into account the remark of the Reviewer, we have now compared male and female data as one dataset. For this purpose, both datasets have been integrated using a normalization by recently identified mouse housekeeping genes (Li et al., 2017) present in the two datasets, as a previously demonstrated strategy to reduce unwanted variation from RNA-Seq data (RUVSeq) (Risso et al., 2014). We provide now - only for the Reviewers - graphs visualizing the normalization of female and male data as well as the comparison of the differentially expressed genes in different group-comparisons, which show comparable numbers of DEGs in particular between females treated or not with DHT, and between males treated or not with DHT (Additional data for the Reviewers-Figure 1a,b).

Consequently, we have now :

- described the step of normalization in the revised version of the manuscript (Methods, lines 647-649)
- updated all panels in Figure 10-R1 and Supplementary Figure 12-R1.
- updated Supplementary Tables 3 and 4 reporting Gene ontology (GO) analysis of differentially expressed genes as well as Supplementary Tables 5 and 6 depicting microglia- and astroglia-related DEGs. These tables are now provided as ‘Supplementary Table 4_GO_DHT_females-males_EdgeR_FDR005_R1’, ‘Supplementary Table 5_microglia_genes_R1’ and ‘Supplementary Table 6_astroglia_genes_R1’.
- accordingly edited the Results (lines 315-343).

After this normalization, we still found that both DHT-treated females and males were clearly separate from their respective controls by principal component and clustering analyses indicating a clear effect of DTH treatment. The top processes involving up-regulated genes were still enriched in terms promoting neuronal activity and function while those enriched in down-regulated genes showed sexual dimorphism, in females related to the immune system / inflammation and in males related to lipid metabolic processes. Thus, this normalization step confirms that gene deregulation is indeed caused by the treatment, but not by other confounding factors not adequately addressed.

In addition, since very often bulk differences are more likely due to changes in cell numbers between conditions rather than true transcriptional changes, some of the results might be blurred by such cell type ratio differences. This needs to be further addressed with respective tools available to the expert in the field.

REPLY : We agree with the Reviewer to tell that bulk differences might be due to changes in cell numbers between conditions rather than true transcriptional changes. However, we provide now several arguments that do not support this hypothesis.

Regarding microglia/macrophages-related genes for which the differences were major, our cell sorting experiments indicate that the percentages of the whole population of phagocytes including CD45+ CD11b+ CD44- microglia and CD45+ CD11b+ CD44+ macrophages remained unmodified in the spinal cord from DHT-treated females and males compared to their respective control (Fig. 8f-R1). This observation supports the idea that the down-regulation of genes observed in DHT-treated females is likely related to true transcriptional changes.

In addition, accordingly to the Reviewer comment, we have used a scRNA-Seq dataset from mouse EAE model (Falcao et al., 2018; GSE113973) to show in the EAE microglial cells (microE cluster, see Additional data for the Reviewer_Figure 2), the specific expression of genes reported as markers of the different activated microglia profiles presently found to be exclusively down-regulated in DHT-treated females but not in DHT-treated males compared to their respective control. In this purpose, we have listed the p-values of DEGs from each prototypical class of microglia and identified 33 genes out of 101 that were down-regulated only in DHT-treated females compared to their control without being deregulated in Vehicle-treated females versus Vehicle-treated males (Supplementary Table 7-R1 ; pink-highlighted genes). Then, we established the dotplots of those 33 genes. 13 out of 33 (including *Capg*, *Ccl6*, *Cd33*, *cd52*, *Clec7a*, *Ctss*, *Cybb*, *Fcgr2b*, *Il1b*, *Itgax*, *Lgals3*, *Tlr4*, *Tnf*) displayed a high average expression mostly in microglial cells (Supplementary Figure 14) validating again that their

exclusive downregulation in DHT-treated females was likely not related to any decrease in cell number. The manuscript has been accordingly edited (lines 348-365).

Regarding prototypical classes of astroglia (Supplementary Table 7-R1). 83 genes out of a total number of 165 DEGs related to astroglia were down (50) or up (33) -regulated only in DHT-treated females (pink-highlighted) compared to female controls whereas only 20 were down (16) or up (4)-regulated only in DHT-treated males (blue-highlighted) compared to male controls. None of those 83 and 20 DEGs displayed differential expression between female and male controls suggesting that the deregulation was caused by the treatment, but not by the sex of the animal. Additionally, 62 genes were deregulated in both DHT-treated females and DHT-treated males with (29) or without (33) a significant difference between female and male controls. Additionally, the high majority of genes (21 out of 26) known to identify activated astrocytes, pro- (A1) and anti- (A2) inflammatory astroglial phenotypes (including *Ggta1*, *Psmb8*, *Serping1*, *Srgn*, *Cd109*, *Cd14*, *Clcf1*, *Emp1*, *S100a10*, *Sphk1*, *Tm4sf1*, *Actn1*, *Bgn*, *C1ql1*, *C4b*, *Ifitm3*, *Igfbp7*, *Ntrk2*, *S100a10*, *S100a11*, *S1pr3*) was exclusively down- (19) or up- (2) regulated in DHT-treated females compared to female controls. Although there are not astroglia in the EAE dataset from Falcao et al (Falcao et al., 2018) (GSE113973) allowing us to establish the dotplots of those genes as done above for microglia, this observation suggests the ability of DHT to control astroglia in females in a specifically different way compared to males (Supplementary Table 7-R1). The manuscript has been accordingly edited in the Results (lines 367-373).

Further, a validation experiment is missing, which could be best by e.g. single cell RNA-Seq, since this would better show, whether the transcriptome differences per cell type are indeed different between male and females.

Actually, our bulk RNA-Seq experiment validates and extends the phenotypic characterization made by different approaches throughout the manuscript including immunofluorescence or cytokine determination. Indeed, as a whole, bulk RNA-Seq data revealed the substantially higher ability of DHT to down-regulate activated profiles of microglia and astrocytes in demyelinated females than in demyelinated males. In agreement with this finding, 1) AR expression is detected mostly in microglia and to a lower extent in astrocytes in the demyelinated lesions from females, but not in the demyelinated lesions from males in which AR expression is quite below the detection threshold (Fig. 1-R1); 2) Among the genes related to microglia and shown to be downregulated in RNA-Seq analysis of DHT-treated females but not DHT-treated males (Supplementary Tables 5-R1 and 7-R1), the downregulation of *Il1b* in females is validated by the decrease of IL-1 β levels in the spinal cord from DHT-treated females, but not DHT-treated males (Fig. 8f-R1). Additionally, we have also validated the female-specific downregulation of *Tnf* (exclusively detected in microglia in the EAE model as shown in Supplementary Figure 14-R1) and *Csf2* genes by using quantitative RT-PCR amplification of DHT-treated female mRNA (Supplementary Figure 13-R1). 3) In the same line, the downregulation of the well-known marker of reactive astrocytes STAT3 is only observed in RNA-Seq analysis from DHT-treated females, which is validated by the visualization of GFAP+STAT3+ immunofluorescence decrease in spinal cord slices from DHT-treated females (Figure 3j-k-R1). Still validating this RNA-Seq data, we previously published that in the presence of the aromatase inhibitor fadrozole, testosterone was unable to decrease GFAP+ STAT3+ labelling in LPC demyelinated lesions from males (Laouarem et al, 2021). For all these reasons, we consider that the request of a third source of experimental evidence to reinforce the results obtained, such as the particularly expensive and difficult experiments like new scRNA-Seq datasets of EAE models would not be fully justified.

Lastly, the data need to be made available prior to publication (at least so that the reviewer sees this) and the respective links need to be added to the manuscript accordingly.

REPLY: The data have been made available to the Reviewers on March 18th, 2022 as indicated by the following message from NCBI and transferred to The Editorial Board of the Journal the same day :

Sujet : Reviewer link created for BioProject PRJNA816168
Date : Fri, 18 Mar 2022 02:47:39 -0400 (EDT)
De : sra@ncbi.nlm.nih.gov
Pour : elisabeth.traiffort@inserm.fr
Dear Elisabeth Traiffort,

Your BioProject's metadata is available at :
<https://dataview.ncbi.nlm.nih.gov/object/PRJNA816168?reviewer=m32uflgbbafp5c4ern5ukvb7g4> in read-only format. You may forward this email to your publisher to share with your reviewer(s) or send them the URL above. It will remain active and reflect all metadata associated with your BioProject until your BioProject is released to the public.

The link above has been now added to the manuscript (lines 689-691).

Further important comments:

Line 135: This is indeed a problem, since the numbers of patients are definitely too low to draw any conclusions. The authors need to make any attempt to better quantify this in a larger group of patients, see suggestions above.

REPLY : The Reviewer is right. Since the submission of our manuscript, we have submitted another manuscript entitled 'Single nuclei RNA-Seq stratifies multiple sclerosis patients into three distinct white matter glia responses' by Macnair and collaborators presently available as a Biorxiv (<https://www.biorxiv.org/content/10.1101/2022.04.06.487263v1.article-info>). An interactive web browser to analyse cell-type specific expression levels of genes and transcriptomic changes in MS versus control tissue is available at https://malhotralab.shinyapps.io/MS_broad/ (for broad cell types) and at https://malhotralab.shinyapps.io/MS_fine/ (for fine cell types). AR sex difference can be observed and confirm our preliminary data using AR antibody (Supplementary Figure S1-R1) as shown now in the plot provided as Figure 2d-R1.

In addition, we have now performed new RNAscope experiments by using an AR RNAscope probe on human tissues. The data have now been include in an additional figure in the revised version (Figure 2-R1). Results are now described (lines 97-105). Corresponding Methods have been added (lines 549-575).

Line 150: Findings reported in Fig 2 seem to be performed in a single experiment with 4 mice per group only. A validation experiment, best even in a different murine strain would be required to increase the validity and generalizability of the data, see major comment above in general.

REPLY: The experiment presented in Figure 2 has been repeated with a new cohort of animals and allowed us to pool animals data leading to n=8 per condition. Figure 2 has thus been edited and is now called Figure 3-R1. As indicated above, we have also performed validation

experiments for all the experimental protocols using either LPC or EAE demyelination in order to increase our cohorts until 6-8 animals / condition. Consequently, we have edited the corresponding histograms. In addition, we have used another mouse strain (129X1/SvJ) and provide now evidence that the pharmacological blockade of AR by flutamide impedes myelin regeneration in both C57BL/6 and 129X1/SvJ strains. These data are now available in Supplementary Figure 3-R1 and are mentioned at line 171-172.

Line 180: “The differential effects induced by testosterone and DHT on PLP expression and microglia response upon demyelination suggested that exogenous testosterone may induce its effects via both AR and/or ER after its aromatase-mediated conversion to estradiol (E2)”. To test this hypothesis, the authors performed an experiment on ovariectomized females injecting DHT, E2 or DHT+E2. However, this does not directly test the hypothesis on the aromatase activity. For this, the appropriate test will be to inhibit aromatase activity in the CNS.

REPLY: As recommended by the Reviewer, we have performed a new experiment by using the aromatase inhibitor fadrozole, a molecule that we previously used in a recent publication (Laouarem et al, *Glia*, 2021). These new data are included in Figure 4-R1 (panels j-n). They confirm the results derived from the experiments using DHT and E2. Indeed, inhibiting the conversion of testosterone to estradiol by fadrozole maintains the increase of MBP staining compared to the vehicle even though the increase is found significantly lower than the one induced by testosterone alone corroborating that DHT and E2 may display additive effects on MBP expression in the lesion. Moreover, in the presence of fadrozole, testosterone still decreases Iba1+ microglia staining, but also increases Arg-1+ staining in a consistent manner with our previous results (Fig. 4h, i-R1) showing that only DHT (not E2) induces Arg-1 expression in the LPC lesion. The results are included in the manuscript (lines 157-163).

Line 268: Why should there be a linear relationship. What is the basis for this assumption? Even visual inspection of the data points in panel E might be better explained by other models.

REPLY: The Reviewer is right. Our representation of the g-ratio according to axon diameter was not appropriate. We have withdrawn the sentence regarding the linear relationship (line 204), changed the panel dedicated to the analysis of the g-ratio (Figure 7e-R1). The graphs now clearly visualize that the g-ratio values are much lower under testosterone or DHT treatments than in the vehicle condition. We also provide the histogram representing the mean g-ratio value in each group of animals.

Fig 7 + 8 : The authors compare the DHT effect on EAE progression in females vs males. At the given time point the values for the control treatment for some parameters (e.g. Th17, Tbet+, TNF in the lymph node) are at higher values in females compared to males. Can the authors please clarify if this difference is significant? What is the value of comparing similar treatments of DHT on two different stages of inflammation in the model?

REPLY: Our choice to characterize the effects of DHT at both 14 dpi (Figure 8-R1 and Figure 9-R1) and 30 dpi (Figure 7-R1) has been guided by the idea to detect both precocious effects of the hormones at the time when the neurological scores become significantly different between the vehicle group and the hormone-treated group in order to avoid potential compensatory mechanisms occurring with time and, later effects when the scores have reached their respective plateau. We have now mentioned more clearly this point in our text (lines 228-230). The Reviewer is right regarding the higher value of some parameters in vehicle-treated females compared to males. Now, we provide the new Supplementary Figure 9-R1 that, indeed, shows significantly higher levels of Th1 and Th17 cells in the lymph nodes from vehicle-treated

females compared to males. The difference disappears in the presence of DHT. All sex-dependent differences in immune cells are now described and commented (lines 259-271).

Fig 7 and Fig 8 : in the text the authors compare the results presented in the two figures, though separating the results to two figures does not allow proper comparison for the reader.

REPLY: As recommended by the Reviewer, we have replaced Fig. 7 and 8 by a single figure in which both female and male data are gathered. As mentioned above, Supplementary Figure 9 also allows proper comparison for the reader.

Line 412: The knowledge driven scoring strategy for the pathways is not very clear from the methods, and therefore limits the ability to assess the quality of work and moreover will not allow reproducibility of the results. The authors should provide the list with their scoring and properly define the way the scoring was attributed to each gene. Moreover, different numbers of genes overlap with the DE genes in females and males, does this have an impact on the results of the scoring done?

REPLY: We agree with the Reviewer that several aspects of our bulk transcriptomic analysis require to be clarified. This is namely true for the scoring procedure aimed at assessing the impact of DHT on the process of oligodendrogenesis. Therefore, we now provide four .xlsx tables and histograms with the detailed analyses based on our curation strategy of scoring the deregulated genes implicated in different aspects (processes) of oligodendrogenesis for the exclusive use of the Reviewers (Additional data for the Reviewers_Tables 1-4). Tables 1 and 2 (Tab 1) correspond to all genes (n=391) selected according to the approach now described in Methods (lines 660-667) and for which the fold changes between DHT-treated versus Vehicle-treated data are shown for females and males, respectively. Tables 3 and 4 (Tab 1) correspond to the deregulated genes (211 in females and 95 in males) for which the value of the False Discovery Rate (FDR) correction (that is considered to be an indicator of the strength of a study) is < 0.05 . The full curation dataset with the corresponding references for each gene will be published in another study in preparation by the laboratory of Dr Carlos Parras and made accessible as a resource for the scientific community. Tab 2 of each .xlsx file presents graphs derived from the above data. The four graphs visualize in females and males: 1) The total score characterizing each oligodendroglial process; 2) The number of genes involved in each process; 3) The score of genes promoting and inhibiting each process; 4) The number of genes promoting and inhibiting each process.

All these analyses and the corresponding graphs have been performed after normalization of the female and male data according to the request of the Reviewer (see above). Consequently, we have updated the 'oligodendrogenesis' histograms (Figure 10-R1), added a new figure (Supplementary Figure 12-R1; see above for details) and edited the Results (lines 323-334).

Minor comments:

Markers mentioned are not always explained for their biological relevance. Would be good to be consistent with a brief explanation to provide a friendly reading for people who are not regularly working on CNS. Examples: MBP, line 195 + line line 214, only explained at line 254.

REPLY: We have now added the useful explanation for MBP (line 127), Iba1 (line 82), GFAP and Olig2 (line 84-85) that were indeed lacking.

Line 20, Figure 1B: For this reviewer the image was not clear. Maybe it's better to present it in an overlay. Moreover, it seems panel E should fit right after panel B.

REPLY : We have now modified Figure 1b by using an overlay and we have placed panel E right after panel B.

Paragraph 1 of results, missing a summarising sentence.

REPLY: A sentence has been added (lines 92-94).

Figure S1: for people with no experience in MS, the detection of lesions in the images are not clear. Can help to have arrows indicating the lesions in the image.

REPLY: We have now added further information to the legend of Supplementary Figure 1-R1 to help clarify what we are drawing attention to. The pictures are taken so that they encompass the lesions, so arrows are not helpful. However, this should be clearer.

Figure. 7G-H: flow-cytometry, what is the percentage of (%)? of immune cells populations of leukocytes?

REPLY: Spinal cord, in contrast with spleens and lymph nodes, contain mainly non-immune cells. We therefore used the CD45 leukocyte marker to discriminate between CD45+ leukocyte and CD45- non-immune cells and determined the percentage of CD45+ immune cells for each condition (Fig. 8f-R1). In contrast, in spleens and lymph nodes (Figure 8d, e-R1), excepted the capsule, all cells are leukocytes. Thus, we considered that CD45 markers was not necessary for this experiment since 100% of cells in suspension expressed CD45. This point has been written more clearly in the Methods (line 608) and in Figure 8 legend (line 1097).

Line 446-473: the discussion on specific genes will benefit adding plots to supplementary data or as part of the main figure.

REPLY: According to the Reviewer's comment, we have now provided a new 'xlsx' file (Supplementary Table 7-R1) reporting the DEGs related to the different microglia (Tab 1) and astroglia (Tab 2) profiles showing those specifically deregulated upon DHT treatment only in females or only in males or in both females and males. In addition, we have added histograms including genes characteristic of different classes of microglia (homeostatic, DAM, WAM) in Figure 10-R1 (panels j, k, l).

Reviewer #2 (Remarks to the Author):

This study addresses the role of androgens in remyelination and neuroinflammation in females, primarily using two model systems: LPC-induced demyelination of the corpus callosum and C57BL/6 EAE. They report that androgen receptor (AR) is unexpectedly expressed at greater levels in demyelinated lesions of female vs male LPC and MS lesions and further that androgens impact myelin repair (in LPC) and neuroinflammation (in EAE) in female mice. In LPC female mice, AR impacts macrophage/microglia phenotype, although macrophage/microglia-specific effects are not required for augmentation of oligodendrocyte differentiation. In EAE, clinical effect of androgens is similar between female and male mice, although aspects of the central and peripheral immune responses and bulk transcriptomics differ between males and females. Overall they conclude that androgens are required for remyelination in females and have gender-specific effects in both LPC and EAE models.

The strongest conclusion supported by the data is that androgens play a role in myelin repair in female mice, as evidenced by Fig 4 using flutamide to block endogenous AR. This is a consequential finding that may have clinical relevance. The authors also demonstrate that androgen effects at the cellular level differ between male and female mice in the 2 models, which is not surprising but nonetheless of interest with regard to the growing recognition of the importance of sex differences in MS. The overall amount of work included in the manuscript is impressive.

The biggest weakness is that the findings are largely descriptive rather than mechanistic, as even the cellular targets that are important for AR effects in females remain uncertain. The key cellular target identified in the LPC model (macrophages and microglia) do not mediate the effects of androgens on remyelination, leaving their role and the key cellular targets unidentified.

REPLY: We agree with the Reviewer that our investigation of the key cellular target identified in the LPC model in females was really disappointing. However, as previously recommended by Reviewer 1 and because some data were showing intragroup variance (in particular for the determination of the ability of OPC to differentiate into CC1⁺ oligodendrocytes in the presence of DHT in the conditional mutant), we performed validation cohorts for all EAE and LPC experiments including the one using the mouse strain allowing the conditional removal of AR from microglia. Now, the data indicate that the removal of AR from microglia in females clearly prevents DHT to induce OPC differentiation and astrogliosis decrease whereas the first cohort led to show only lower but non-significant effects of DHT in the mutant compared to the wild-type animals. This new data identifies microglia as a key cellular target of AR-mediated effects of androgens in females. In addition, we have now clearly listed genes known to characterize various classes of microglia and astrocytes which here appear to be selectively deregulated under DHT treatment only in females and not in males towards an anti-inflammatory effect. These mechanistic data have been now included in Fig. 6-R1 and Supplementary Table 7-R1 and have been described in the text (lines 181-185; 348-373).

A minor concern is that there is no unifying theme between the LPC and EAE experiments, such that the manuscript feels like two separate stories.

REPLY: We understand the comment of the Reviewer and have included more clearly in the text the requirement for considering remyelination in the context of peripheral immune cell infiltration as it occurs in MS (lines 188-189).

Specific comments:

1. Given that the most consequential finding is the role of androgens in myelin repair in females, this data should be the most convincing in order to best justify the conclusion. A few concerns arise in this regard:

- All treatments began on the same day as LPC injection, such that the effects of treatment could reflect protection of OPC/oligos from LPC-induced injury rather than a specific effect on remyelination. Ideally treatment would begin on day 2 or 3, but it would suffice to do a control experiment in which mice treated on day 0 are sacrificed on day 2 to ensure similar extent of demyelination in treated vs vehicle mice.

REPLY: Actually, LPC injection has been done 15 hrs after EAE injection in order to use a 'therapeutic administration of the hormones' as done in the EAE model where hormone

administration starts at the onset of neurological disabilities. However, the Reviewer is fully right, we did not correctly mention this point neither in the text nor in the schemes describing the LPC protocols. We have now edited the text (line 121) and the different schemes visualizing the experimental protocols in Figures 3-R1 to 6-R1.

- It seems odd that g-ratios were performed in EAE (in which remyelination is minimal) but not LPC. Quantification of remyelination by EM g-ratio should be performed in the LPC model

REPLY: The reviewer is right. We have now performed a new experiment in which animals have been analyzed at 14 dpl in order to visualize axons and myelin sheaths at the ultrastructural level. The data have been added in Figure 3-R1 (panels l-o) and described in lines 134-139. They indicate a highly significant decrease of the g-ratio values in the DHT-treated mice.

- LPC mice are only analyzed at day 7 post-lesion, which is an early time point when oligodendrocyte differentiation has just begun. It's difficult to feel confident in conclusions drawn solely from such an early time point, and a subsequent time point would yield greater insight into the effect of androgens on microglia responses and myelin repair, at least in the flutamide model. As reported (doi: 10.1038/s41593-019-0418-z), microglia phenotype changes over time following LPC demyelination in corpus callosum.

REPLY: As requested by the Reviewer, we performed a new LPC experiment including flutamide-treated mice analyzed for MBP and Iba1/Arg-1 staining at 10 dpl when remyelination is ongoing. The data are included in Figure 5-R1 (panels j-m). Results show that at 10 dpl, flutamide consistently prevents the increase of MBP and Arg-1 expression in the demyelinated area as now indicated in the text (lines 170-172).

2. Given the authors' focus on macrophages/microglia, a disappointing aspect of the story is that macrophage/microglia-specific knockout of AR has no impact on the observed effects on OPC and oligodendrocyte differentiation, even though expression of AR on other CNS cells appears minimal in female mice. Yet this finding is glossed over in the manuscript. Could the impact of androgens on myelin repair be mediated by peripheral, infiltrating cells?

REPLY: As mentioned in our first reply above, we agree with the Reviewer to say that this aspect of the paper was quite disappointing. As for our other experiments, we increased our animal cohorts in order to more strongly validate the data. The experiment presented in Figure 5 comprised only n=3 animals because of some difficulties in the production of animals. Since the first submission of our manuscript, we succeeded in boosting the animal production and thus performed a new experiment. The pooled data are now presented in Figure 6-R1. Although DHT induced only a lower effect on the percentage of Olig2+ CC1+ cells in the mutant compared to the wild-type animals in the initial version of the manuscript, its effect is now found to be clearly prevented on OPC differentiation supporting the hypothesis that in females the expression of AR in microglia is not only involved in the ability of this cell type to express the anti-inflammatory marker Arg-1, but also in the capacity of OPCs to differentiate into CC1+ oligodendrocytes. The effect of DHT on astrogliosis is also clearly impaired in the mutant. Therefore, we have edited the corresponding text (lines 181-185).

3. The human data is difficult to interpret, largely because the extent of samples from which it is derived is not explained clearly. Supplemental Table 7 provides descriptions of the donors, but it's not clear if all these donors were included in the IHC experiments and how many of each type of lesion were included (i.e., the sample size from which numbers were derived). This

should be clarified. Further, does “% of CD68+AR+” cells mean the % of CD68+ cells that are double positive for AR+? This is unclear and should be clarified. Finally, these experiments were not completed because an antibody was discontinued. From the methods, it appears 2 other antibodies were tested unsuccessfully. Given the translational importance of this data, are no other methods possible to complete the study? Perhaps in situ hybridization as was used for the LPC mice?

REPLY: Actually, the number of each type of lesions was indicated in the last column of Supplementary Tables 7 in the first version of the manuscript. However, we have now performed RNAscope analysis of AR in another series of tissues derived from MS patients (4 females and 4 males) and non-MS patients (2 females and 2 males). The data are presented in the new Figure 2-R1. The full description of the patients and corresponding samples (including the type of lesions that were examined are shown in Supplementary Tables 1 and 2. RNAscope data corroborate the data obtained by immunostaining as now described (lines 97-105) in the revised manuscript. In addition, since the submission of our manuscript, we have submitted another manuscript entitled ‘Single nuclei RNA-Seq stratifies multiple sclerosis patients into three distinct white matter glia responses’ by Macnair and collaborators presently available as a Biorxiv (<https://www.biorxiv.org/content/10.1101/2022.04.06.487263v1.article-info>). An interactive web browser to analyse cell-type specific expression levels of genes and transcriptomic changes in MS versus control tissue is available at https://malhotralab.shinyapps.io/MS_broad/ (for broad cell types) and at https://malhotralab.shinyapps.io/MS_fine/ (for fine cell types). AR sex difference can be observed and confirm our preliminary data using AR antibody as shown now in the plot provided as Figure 2d-R1.

4. For all experiments, why are the female mice ovariectomized? The expression of AR was determined in non-ovariectomized LPC mice (Fig 1), but all subsequent interventional experiments are performed in ovariectomized mice. Might not the expression of AR and other physiologic effects of androgens be altered in female mice after ovariectomy?

REPLY: We have chosen to assess the effects of the sexual hormones in gonadectomized females in order to exclude the confounding effects of endogenous gonadal steroid hormones as previously done for our work regarding male animals. We have nevertheless considered the comment of the Reviewer and therefore performed RNAscope analysis for evaluating AR transcription in the LPC lesion from ovariectomized female mice. As shown in Supplementary Figure 15, AR transcription is still detected in Iba1+ cells in the lesion from ovariectomized animals. The text has been modified accordingly (lines 456-454).

Minor point:

1. The number of sections examined per animal for microscopy studies is specified in the methods, but this should be stated in figure legends to make evaluation of rigor easier for the reader.

REPLY: We have now added the number of slices in the different figure legends.

Reviewer #3 (Remarks to the Author):

This is an exhaustive study demonstrating a plethora of data which, in summary, suggest that androgens play in the context of inflammatory, demyelinating disorders, such as Multiple Sclerosis, a major role in females that is critically different from their role in males. To address this interesting hypothesis, the authors use a well characterized model for remyelination (id est

the LPC model) and a model of auto-immune driven inflammatory demyelination (id est the EAE model). As there are fundamental biological differences between each gender, it is pivotal to address divergent effects of drugs and potential treatment strategies. This, the topic of the presented study is of high relevance. As a major limitation, the presented study addresses two distinct but fundamentally different aspects of the MS pathology, remyelination and auto-immune driven inflammation. It would have been more convincing to focus on one aspects and try to understand the cellular mechanisms in more detail. Nevertheless, while the study is worth to be published there are several major and minor aspects that need the full attention of the authors. In particular, the following aspects should be addressed:

1. In the material and methods section it is stated that “drugs were administered at the onset of clinical symptoms until Day 30 after immunization.” Was this done per individual animal or was drug treatment started at the same days for the entire cohort? Please specify.

REPLY: The onset of clinical symptoms occurred after the same delay in all animals. Thus, drug treatment was started on the same day for each entire cohort. This is now specified in the Methods (lines 489-490).

2. Some minor typos should be corrected such as “The RNA-seq libraries were prepared using either the NEBNext Ultra II Directional RNA Library Prep 811 Kit (NEB) and sequenced with the Novaseq” (either should be deleted).

REPLY: The text has been edited (line 640).

3. The authors state that “In case of absence of distribution normality, non-parametric tests (Mann-Whitney two-tailed, Kruskal-Wallis with Dunn’s post tests for comparison) were used.” Please state how normal data distribution was evaluated.

REPLY: Two tests have been used including D’Agostino & Pearson normality test and Shapiro-Wilk normality test now indicated in lines 681.

4. The authors state that “At 7 days post-lesion (dpl), when the process of spontaneous remyelination is ongoing and corresponds to the end of OPC recruitment and the beginning of their differentiation...” Please either provide appropriate citations for this statement or demonstrated it with the used samples.

REPLY: Prof. Robin Franklin and collaborators have previously described that ‘5, 10, and 14 days post-lesion (dpl), corresponded to the timing of peak OPC recruitment, initiation of OPC differentiation, and myelin sheath formation, respectively’ (Fancy et al, 2009). However, in our own experiments (Laouarem et al, *Glia*, 2021, Figure 7), we found that at 7 dpl, we could still detect OPC proliferation and already visualize OPC differentiation, hence the sentence used above. Both references have now been added (lines 477).

5. Figure 1B demonstrates a LPC-induced lesions with ongoing remyelination, as stated by the authors. Is this true for the entire lesions or for the lesion rim? Was there any difference of AR-expression throughout the lesion? Beyond, it is stated that “we observed a strong AR upregulation in the lesion from females while AR transcripts could be detected at a much lower level in the lesion from males”. The demonstrated images are not convincing and quantification should be performed. Is the AR-expression induction due to LPC-induced demyelination, or due to the mechanical, needle-induced injury. Vehicle-treated mice would be required to answer this important question.

REPLY: We agree with the Reviewer regarding the fact that images in Figure 1B were not convincing enough. Therefore, we have edited the figure and now show separate channels and overlays in order to better visualize nuclei concentration reflecting the LPC lesion and the high

AR transcript signals in the lesion from females but not males. In contrast, the cortex express AR in both females and males. The requested quantification was already provided in the first version, but the histogram was located far from panel 1B. Therefore, we have also modified the panel organization (Figure 1-R1 panels b-d).

The Reviewer is right when he/she speaks of the existence of a rim corresponding to the progressive recruitment of new oligodendrocytes from the outside of the lesion. AR expression was nevertheless found homogenously distributed throughout the lesion in agreement with its main localization in microglia and not in OPCs/oligodendrocytes.

Finally, we took into consideration the remark of the Reviewer suggesting that AR expression induction might be due to the mechanical, needle-induced injury. In order to investigate this hypothesis, we performed a new experiment including vehicle-treated mice. RNAscope was performed in slices from intact females, which have received either LPC or the vehicle. As shown in Supplementary Figure 15, animals receiving the vehicle display inflammation (as shown by Iba1 expression) but no induction of AR expression. This point is now indicated in line 454.

6. A key step in androgen action is AR nuclear translocation. Can the authors provide evidence that the AR is indeed expressed in the nuclear compartment? The high-power insert in Fig1C rather suggest a perinuclear expression pattern, especially in IBA1+ cells.

REPLY: The Reviewer is right, the images in Figure 1C showed a predominant perinuclear expression pattern. Now, we provide images at a higher magnification leading to detect besides the perinuclear localization, an immunofluorescent signal in the nucleus (Figure 1f-R1). Because of this unexpected observation, we have also used an antibody directed to DHT ligand (Figure 1g-R1), which leads to a clear nuclear labeling in a wide majority of Iba1+ cells. We can also notice that besides the nuclear DHT+ signals (white arrows in Figure 1g-R1), a perinuclear signal can also be observed (yellow arrowheads in Figure 1g-R1). Our data do not allow to exclude that our polyclonal antibody mostly detect the ligand-unbound AR, which is likely present in the cytoplasm. However, in other cell types out of the CNS, the existence of classical and non-classical pathways of androgen action have been proposed to co-exist. This is true for the Sertoli cells where ligand-bound AR monomers can either migrate to the inner side of the cell membrane and interact with Src, thus activating the non-classical/non-genomic pathway of androgen action or alternatively translocate to the nucleus and form homodimers that can interact with androgen response elements or with other transcription factors, thus activating the classical genomic pathway (Edelsztein and Rey, 2019). The visualization of both extra and intra-nuclear staining by using the DHT-antibody might support the latter hypothesis. However, this remains to be investigated. A few sentences of the discussion indicate these hypothesis (lines 385-389).

7. The authors should clearly state which control experiments were performed to demonstrate the specificity of their stains.

REPLY: Additional data for the Reviewers-Figure 3 provides images obtained with the AR antibody in the ipsilateral and contralateral sides of the LPC lesion showing the exclusive labeling of cortical neurons in the contralateral side compared to the labeling of both the cortical neurons (at a higher level) and the callosal lesion in the ipsilateral side. Additional data for the Reviewers-Figure 3 also shows that the secondary antibody does not lead to any non-specific labeling.

8. The studies using MS tissues appear to be preliminary, and I am not sure whether they add much to the paper with this limited number of investigated cases. In case no reliable antibodies are available the authors could dissect different lesions of cyrosections and perform mRNA expression analyses.

REPLY: We have now performed RNAscope analysis of AR in another series of tissues derived from MS patients (4 females and 4 males) and non-MS patients (2 females and 2 males). The data are presented in the new Figure 2-R1. The full description of the patients and corresponding samples (including the type of lesions that were examined are shown in Supplementary Tables 1 and 2. RNAscope data corroborate the data obtained by immunostaining as now described (lines 97-105) in the revised manuscript. In addition, since the submission of our manuscript, we have submitted another manuscript entitled ‘Single nuclei RNA-Seq stratifies multiple sclerosis patients into three distinct white matter glia responses’ by Macnair and collaborators presently available as a Biorxiv (<https://www.biorxiv.org/content/10.1101/2022.04.06.487263v1.article-info>). An interactive web browser to analyse cell-type specific expression levels of genes and transcriptomic changes in MS versus control tissue is available at https://malhotralab.shinyapps.io/MS_broad/ (for broad cell types) and at https://malhotralab.shinyapps.io/MS_fine/ (for fine cell types). AR sex difference can be observed and they confirm our preliminary data using AR antibody as shown now in the plot provided as Figure 2d-R1.

9. Again, in figure 2 it remains unclear whether the observed effects of testosterone and DHT on astrocytes and microglia are linked to the LPC-induced demyelination or the mechanical injury induced during the LPC-application. Sham-operated groups would be required.

REPLY: In order to investigate the hypothesis, we performed GFAP and Iba1 staining on slices derived from intact females stereotaxically injected with PBS. As shown in Additional data for the Reviewers-Figure 4, the mechanical injury does not promote any substantial astrogliosis and/or microgliosis comparable to the one induced by LPC as shown in the present manuscript.

10. In the text it is stated that aromatase to be upregulated in the lesion from female mice. The corresponding figure demonstrates indeed aromatase expression but whether this is due to the LPC-induced demyelination remains unclear. Again, sham-operated mice would be supportive.

REPLY: In order to investigate the hypothesis, we performed aromatase staining on slices derived from intact females stereotaxically injected with PBS. As shown in Supplementary Figure 2, the mechanical injury does not promote a substantial increase in aromatase expression comparable to the one induced by LPC. This observation is indicated in line 146.

11. The authors state that “flutamide-treated animals displayed a significant decrease in the percentage of OPCs that are able to differentiate into CC1+ oligodendrocytes compared to the vehicle condition”. This sentence appears misleading. As the percentage of OLIG2/CC1 double positive cells in relation to the entire OLIG2 cell population is lower in flutamide-treated mice, this would mean less cells mature under flutamide treatment. Please rephrase.

REPLY: The Reviewer is right, we have rephrased (lines 167-168).

12. In some cases, myelination is estimated by anti-PLP, in other by anti-MBP stains. The authors should either comment on this discrepancy or perform both stains for all the subexperiments.

REPLY: We agree with the Reviewer and thus have performed MBP staining for the panel f in Figure 3-R1 instead of PLP. The result obtained with PLP is similar to the one obtained with MBP.

13. The authors state that microglia are the main AR-expressing cells in the CNS. Did the authors consider that a significant proportion of these cells are IBA1+ recruited monocytes in the LPC model.

REPLY: Indeed, we cannot exclude that a restricted number of macrophages could have infiltrated the LPC-induced lesion as previously shown for T cells (Ghasemlou et al, 2007) and as proposed by (El Wali et al, 2020). Consequently, we have edited the text (line 84).

14. Figure 6K should be Claudin5+ area instead of Claudin+ area.

REPLY: The figure 7k-R1 has been edited.

15. The presented results using the EAE model are somewhat irritating. First, the authors should try to focus on the most relevant findings and move the less relevant ones into the supplements. Second, their FACS analysis clearly demonstrate that the observed protective effects of DHT are at least in part due to immunosuppressive functions. For example, proportions of CD4+ T cells as well as the proinflammatory Th1 and Th17 cells are lower in the secondary lymphoid organs in DHT-treated compared to Vehicle-treated mice. However, the main focus of this manuscript so far was induction of remyelination. It is not clear to me how these data help to strengthen the so far observed pro-myelinating effects. I rather would suggest to verify the pro-myelinating effects in another model of remyelination (such as the cuprizone model) or to start treatment during the chronic phase of the EAE disease when the lesions are fully established.

REPLY: We understand the comment of the Reviewer. However, the discrepancy observed at the level of the inflammatory cells (microglia, macrophages, astrocytes) between females and males let us consider that those cells have a major importance in the ability of androgens to promote remyelination and thus should not be set aside. The higher infiltration of macrophages in females compared to males may also be a critical point in the remyelination process. We have now included more clearly in the text the requirement for considering remyelination in the context of peripheral immune cell infiltration as it occurs in MS (lines 188-189). Regarding the use of another model allowing to support once more the ability of androgens to induce remyelination, we preferred the LPC model since the cuprizone model had been already used to show the ability of androgens (even though it was not DHT but testosterone) to promote MBP staining increase in females (Hussain et al, 2013).

16. I am not sure if the NGS data add much to the manuscript. Since bulk RNA sequencing was performed, it is hard to assign the observed expressional changes to a specific cell type.

REPLY: Actually, our bulk RNA-Seq experiment validates and extends the phenotypic characterization made by different approaches throughout the manuscript including immunofluorescence or cytokine determination. Indeed, as a whole, bulk RNA-Seq data revealed the substantially higher ability of DHT to downregulate activated profiles of microglia and astrocytes in demyelinated females than in demyelinated males. In agreement with this finding, 1) AR expression is detected mostly in microglia and to a lower extent in astrocytes in the demyelinated lesions from females, but not in the demyelinated lesions from males in which AR expression is quite below the detection threshold (Fig. 1-R1); 2) Among the genes related to microglia activation and shown to be downregulated in our RNA-Seq analysis of DHT-treated females but not DHT-treated males (Supplementary Tables 5-R1 and 7-R1), the downregulation of *Illb* in females is validated by the decrease of IL-1 β levels in the spinal cord

from DHT-treated females, but not DHT-treated males (Fig. 8f-R1). Additionally, we have also validated the downregulation of *Tnf* (exclusively detected in microglia in the EAE model as shown in Supplementary Figure 14-R1) and *Csf2* genes by using quantitative RT-PCR amplification of mRNA from DHT-treated females Supplementary Fig. 14-R1). 3) In the same line, the downregulation of the well-known marker of reactive astrocytes *STAT3* is only observed in RNA-Seq analysis from DHT-treated females, which is validated by the visualization of *GFAP+STAT3+* immunofluorescence decrease in spinal cord slices from DHT-treated females (Fig. 3j, k-R1). Still validating this RNA-Seq data, we previously published that in the presence of the aromatase inhibitor fadrozole, testosterone was unable to decrease *GFAP+ STAT3+* labelling in LPC demyelinated lesions from males (Laouarem et al, 2021). For all these reasons, we consider that our RNA-Seq data are consistent with our slice immunolabeling and flow cytometry experiments. All these complementary data make our statements more convincing.

Additional references :

- Edelstein NY, Rey RA (2019) Importance of the Androgen Receptor Signaling in Gene Transactivation and Transrepression for Pubertal Maturation of the Testis. *Cells* 8.
- Falcao AM, van Bruggen D, Marques S, Meijer M, Jakel S, Agirre E, Samudyata, Floriddia EM, Vanichkina DP, Ffrench-Constant C, Williams A, Guerreiro-Cacais AO, Castelo-Branco G (2018) Disease-specific oligodendrocyte lineage cells arise in multiple sclerosis. *Nature medicine* 24:1837-1844.
- Li B, Qing T, Zhu J, Wen Z, Yu Y, Fukumura R, Zheng Y, Gondo Y, Shi L (2017) A Comprehensive Mouse Transcriptomic BodyMap across 17 Tissues by RNA-seq. *Scientific reports* 7:4200.
- Risso D, Ngai J, Speed TP, Dudoit S (2014) Normalization of RNA-seq data using factor analysis of control genes or samples. *Nat Biotechnol* 32:896-902.

REVIEWER COMMENTS_R2

Reviewer #1 (Remarks to the Author):

1.Human cohort: Despite the increase in data size and the new measurements provided, the cohorts, especially control, are still very small to address the within control/disease group sex comparison.

REPLY: Indeed, the number of human tissues was still low. Therefore, we performed new RNAscope experiments including n=5 male controls and n=5 female controls, n=5 male MS and n=6 female MS. Figure 2 and the corresponding legend as well as Supplementary Table 1 have been accordingly updated.

2.RNaseq analysis:

Despite the additional information of the scoring method, it is still unclear how the scoring was performed. The authors explain in the rebuttal the scoring is based on a work that is in preparation in another lab, and provide partial information for the review of this manuscript. It is not possible to assess the quality of the RNaseq analysis work in this manner, without knowing the papers that were used for the curation, and the full details of the scoring scheme. It is therefore an absolute requirement that the authors only provide results based on already published and established methods, such as GO enrichment, or provide additional analysis that can be properly reviewed and accepted. At the moment the scoring method given cannot be reproduced or verified. It is also disturbing that the RNA-seq data being presented in this manuscript are supposed to be published in a different paper, for which it remains unclear,

whether this will happen in a timely fashion. Following FAIR principles the data have to be published with this manuscript or the authors wait until the other paper is published and then reference it. Choose either way, but as suggested, it cannot be accepted.

REPLY: In agreement with the Reviewer's comment, we have decided to release now the OligoScore (<https://oligoscore-staging.icm-institute.org/>) resource, an open resource to the community, to provide access to it in this paper, and have described the approach in Methods section (lines 668-677). Furthermore, we now provide as supplementary data, four Excel tables and histograms with the detailed analyses based on OligoScore curation strategy of scoring the deregulated genes for their implication in the different processes of oligodendrogenesis (Supplementary Tables 4-7). Tables 4 and 5 (Tab 1) contain all curated genes (n=391, at the moment when the analysis was performed) and for which the fold changes between DHT-treated versus Vehicle-treated data are shown for females and males. Tables 6 and 7 (Tab 1) contain only the deregulated genes (FDR correction is < 0.05) being curated (211 in females and 95 in males). These Supplementary Tables are now cited in the results line 329.

- The authors used different gene sets that represent the main subpopulations of microglia. In order to assess the gene set behavior in the different groups, the authors are requested to perform the results of an enrichment analysis for these gene sets in the different groups and indicate clearly in the text the changes that were observed. At the moment the methodology is not optimal and the description in the text is unclear.

REPLY : Following the suggestion of the reviewer, we have performed gene set enrichment analysis (GSEA), using Cluster Profiler in R, with all the microglial genes combined (105 genes) ordered accordingly to the logFC in females (DTHFvsCTF) or males (DTHMvsCTM) comparisons. This GSEA analysis shows an enrichment of many (219) gene sets in DTH-treated females (compared to their controls), most of them (208) suppressed (genes being downregulated), and many related to immune and inflammatory processes, including 'lymphocyte mediated immunity', 'response to stress', 'defense response', 'immune system process', 'immune response', and 'cytokine production'. On the contrary, in males only two GSEA processes were activated ('translation' and 'peptide biosynthetic process') and none were suppressed.

We have added the following paragraph in the results (lines 374-383) and the data in Supplementary Fig. 16 and Supplementary Table 12:

'By combining all microglial gene sets (105 genes), we performed gene set enrichment analysis (GSEA) with these genes ordered by their changes in expression either in female or male comparisons (DTH-treated vs. non treated). In line with previous results, this GSEA analysis showed large enrichment of many gene sets in DTH-treated females but almost none in males, with many of the suppressed gene sets (genes being downregulated) related to immune and inflammatory processes, including 'lymphocyte mediated immunity', 'response to stress', 'defense response', 'immune system process', 'immune response', and 'cytokine production' (Supplementary figure 16; Supplementary Table 12).'

We have also listed the sets of genes down-regulated in microglia in order to make the text clearer (lines 352-358 and 361-364).

Finally, we have described the approach (lines 678-685) in the Methods as follows:

‘Gene set enrichment analysis (GSEA). We used *gseGO* function of *Cluster profiler* R package to find gene sets enriched in the gene list of 105 microglial genes (Supplementary Table 12) ranged by the differential expression (logarithmic fold change, logFC) in DTH-treated vs non- treated females and males, respectively. We found 219 gene set enriched in females but only 2 in males. Dotplot and gseaplot were used for visualization of enriched gene sets. All gene sets enriched are provided in Supplementary Table 12. R script has been deposited in <https://github.com/ParrasLab/Androgen-signaling-and-remyelination-Nat-Commun-paper>.’

- Analysis of **scRNA-seq data**: not clear which tool was used for the analysis. Please provide further information.

REPLY : We have now provided further information in the Methods (lines 686-694) as follows :

‘scRNA-seq analysis. EAERaw.RData object was obtained from Gonçalo Castelo-Branco's lab and processed in R (4.0) using the following packages: *Seurat* (3.0) for data processing and *ggplot2* for graphical plots. Seurat objects were first generated using *CreateSeuratObject* function (min.cells = 5, min.features = 100). Normalized with *sctransform* function. Cell neighbors and clusters were found using *FindNeighbors* (dims = 1:30) and *FindClusters* (resolution = 0. 8) functions. *RunPCA*, and *RunUMAP* functions with default parameters. Clusters were annotated based on cell-subtype markers as detailed in the R script, which has been deposited in <https://github.com/ParrasLab/Androgen-signaling-and-remyelination-Nat-Commun-paper>.

- The current results presented from **the scRNA-seq data** do not resolve the cell number bias that might be present. Therefore, the authors should provide better evidence, such as deconvolution analysis using the scRNA-seq from Falcao et al.

REPLY: In agreement with the suggestion of the reviewer, we have used CIBERSORTx (<https://doi.org/10.1038/s41587-019-0114-2>) and two scRNA-seq datasets: Falcao & Castelo-Branco (GSE113973, not having neurons or astrocytes in it) and the Meijer & Castelo-Branco (GSE166179, not having neurons but containing an astrocyte cluster and only one microglial cluster downloaded from <https://cells.ucsc.edu/>). We obtained similar results, indicating similar number of microglial cells/clusters in males (DTH-treated or not), while in DTH-treated females the deconvolution varied cell proportions in microglial and immune OL/OPC clusters, likely due to the downregulation of the immune related genes. In the present version of the paper, we present the results obtained by deconvolution with Falcao & Castelo-Branco's dataset and refer to it as follows:

In the results section (lines 383-393):

‘To try to exclude bias putatively related to changes in cell numbers between conditions, we used CIBERSORTx, a machine learning method to determine cell type abundance and expression from bulk tissues (<https://doi.org/10.1038/s41587-019-0114-2>), together with a single cell RNA-Seq dataset from mouse EAE model (Falcao et al, 2018 Nat Med. 2019 Apr 26; 24(12): 1837–1844. doi: [10.1038/s41591-018-0236-y](https://doi.org/10.1038/s41591-018-0236-y)) (GSE113973). This deconvolution of our bulk-RNA-Seq datasets suggested that while microglial clusters did not change in proportions upon DTH-treatment in males, DTH-treated females presented some changes in microglial clusters and EAE immune-OL/OPC clusters model (Falcao et al, 2018 Nat Med. 2019 Apr 26; 24(12): 1837–1844. doi: [10.1038/s41591-018-0236-y](https://doi.org/10.1038/s41591-018-0236-y)) (Supplementary Table

13), likely due to the abovementioned dysregulation of microglial/inflammatory genes. Indeed, 21 genes out of 31 downregulated genes only in DTH-treated females are expressed in the EAE microglial cells from this scRNA-Seq dataset (Supplementary Fig. 15)'.

And in the Supplementary Methods :

Bulk RNA-seq deconvolution. We used CIBERSORTx tool (<https://doi.org/10.1038/s41587-019-0114-2>) on the docker module Cibersortx/fractions, with 100 permutations as input parameter, in order to deconvolute our bulk RNA-Seq datasets obtained from EAE spinal cord samples. The signature of scRNA-Seq matrix was generated according to the book methods described by Steen et al (https://doi.org/10.1007/978-1-0716-0301-7_7) with the GSE113973 public scRNA-Seq dataset from mouse EAE model. The deconvolution analysis was performed on two mixture files corresponding to the RNA-Seq count matrices generated as described before, containing females and males' comparisons, 'DHTFvsCTF' and 'DHTMvsCTM', respectively. The results obtained are estimated as the proportions of each cell types in each RNA-Seq sample inferred from the prior knowledge of the scRNA-Seq sample. R script has been deposited in <https://github.com/ParrasLab/Androgen-signaling-and-remyelination-Nat-Commun-paper>.

- Why are the RNA-seq analyses for male and female (rebuttal, page 4 first paragraph) only for the reviewers? They should be included in the paper.

REPLY: We now included the Additional data for the Reviewers - Figure 1a,b as 'Supplementary Figure 11'. This figure has been mentioned in the text (line 316).

Minor:

3. Lines 265-266: vehicle-treated females displayed as much microglia as macrophages whereas males displayed predominant microglia compared to macrophages – since this is not directly shown in the same plot, readers will benefit by specifying the values in the text to.

REPLY : In agreement with the Reviewer request, we have now specified the values in the text (lines 266-268).

4. Line 105: please add: “compared to males” – please provide the statistical test performed and values.

REPLY : According to the request of the Editor, we have withdrawn Figure 2d (described in line 105) and moved this panel as Supplementary Figure 1 panel I in support of our own data concerning the higher expression of AR in female microglia compared to males. This graph was built by analyzing the freely available and searchable data using the shiny app provided in the paper as an open resource to the community (https://malhotralab.shinyapps.io/MS_broad/). Since statistics were not tested as there are not enough cells to do it robustly, the sentence regarding Supplementary Figure 1 panel I was modified as follows : “Moreover, AR mRNA expression in microglia from MS and control donors from a publicly available single-nuclei RNA sequencing database appeared higher in MS female samples compared to males (Supplementary Fig. 1 panel I)' (lines 112-115).

5. Supplementary table 3- please provide an index for abbreviation of groups.

REPLY: Done.

6. Line 300: out of place +

REPLY : Done.

7. Figure S12: indicate in legend the y and x axis more clearly. Without the figure provided only to reviewers this plot is not very clear.

REPLY: As mentioned above (point 1 of Comment 2), we have now included in the Supplementary data of the revised version R2 the figures that were previously provided only to reviewers in the revised version R1. This should clarify Supplementary Figure 12-R1 (which has become Supplementary Figure 13-R2). In the Supplementary Figure 13-R2 (like in Figure 10) all the bar plots have the title of the x-axis just above the axis. In the 'y-axis', if one want to consider this an axis, it is written the oligodendrogenesis processes labeled in each bar (the same in all graphics). The legend of Supplementary Figure 13-R2 has been accordingly edited.

Reviewer #2 :

We thank the Reviewer for his/her comments regarding the R1 revised version.

Reviewer #3 (Remarks to the Author):

1. The authors claim that “Because of this unexpected observation, we have also used an antibody directed to DHT ligand (Figure 1g-R1), which leads to a clear nuclear labeling in a wide majority of Iba1+ cells. We can also notice that besides the nuclear DHT+ signals (white arrows in Figure 1g-R1), a perinuclear signal can also be observed (yellow arrowheads in Figure 1g-R1).” I would recommend to use, for this claim, a higher magnification, essentially the same provided in figure 1g. Beyond, nuclear AR localization in figure 1g is still not convincing for me. Unbiased quantification of the proposed co-localization would be an elegant option.

REPLY: As recommended by the reviewer, we provided a higher magnification for the colocalization of DHT and Iba1. These new images allow the visualization of both nuclear (white arrows) and perinuclear (white arrowheads) DHT immunostaining in Iba1-expressing cells. We have also indicated DHT-expressing cells, which do not co-express Iba1 (yellow arrowheads) and cells expressing neither DHT nor Iba1 (yellow arrows). In addition, as also recommended by the reviewer, we quantified the percentage of Iba1-expressing cells displaying a nuclear versus a perinuclear DHT labeling (Fig. 1hR2), which indicates a higher proportion of nuclear DHT staining. Fig. 1h has been added in the results (line 87) and the figure legend has been edited accordingly.

2. The authors state “Additional data for the Reviewers-Figure 3 also shows that the secondary antibody does not lead to any non-specific labeling.” Please indicate where in figure 3 this is shown.

REPLY: In the right panel of the Additional Figure 3 for the reviewers, the secondary antibody (AbII) shown in the ipsilateral side does not lead to any fluorescent signal neither in the lesion (delineated by the dashed line) nor in the cerebral cortex (Cx) whereas the AR antibody similarly used in the ipsilateral side (left panel) lead to clear signals detected both in the lesion and above the lesion in the cerebral cortex.

3. Early in the p2p response the authors state that “As shown in Supplementary Figure 15, animals receiving the vehicle display inflammation (as shown by Iba1 expression) but no induction of AR expression. This point is now indicated in line 454.” Later on, it is stated that “As shown in Additional data for the Reviewers-Figure 4, the mechanical injury does not promote any substantial astrogliosis and/or microgliosis comparable to the one induced by LPC as shown in the present manuscript:” These contradictors statements are somewhat confusing. Please clarify.

REPLY: The Reviewer is right. Our text was not very clear. What we mean is that the stereotactic injection of the vehicle - instead of LPC - leads to a limited Iba1+ inflammatory process (due to the mechanical injury) as shown in supplementary Figure 15-R1 (which has become Supplementary Figure 17-R2) right panel as well as in the Additional Figure 4 for the reviewers, right panel. In both figures, Iba1+ immunofluorescent signal observed in the vehicle-treated animals is much lower than the Iba1+ signal observed after LPC injection (left and middle panels in Supplementary Figure 17-R2).

4. The authors state that “At 7dpl, flutamide significantly decreased the number of OPCs and Olig2+ cells as well as the percentage of Olig2+ cells differentiated into CC1+ oligodendrocytes (Fig. 5b-e)”. One cannot state that olig2+/cc1+ cells are derived from olig2+/cc1- cells. Thus, the last statement of the sentence is speculative. Beyond, if there are less OPC, less OLIG2+ cells and less more mature oligodendrocytes, would that mean that flutamide treatment induced OPC death? If so, is there any data supporting such a scenario?

REPLY: We edited our text in order to take into account the comment of the Reviewer and thus removed the statement that is speculative (lines 168-170). Regarding the hypothesis that flutamide might induce OPC death, there is no supporting data. 7 dpl is not the appropriate time point to investigate this hypothesis. However, since DHT was previously reported to increase the survival of new neurons in the dentate gyrus, an effect blocked by flutamide (Hamson et al, 2013, *Endocrinology* 154 DOI:10.1210/en.2013-1129), this hypothesis would merit to be investigated in the future via a more accurate analysis of DHT effect on the expression of survival markers in the presence or absence of flutamide at much earlier time points after LPC injection.

5. The authors correctly state that “testosterone or DHT-treated females displayed significantly lower scores throughout the whole experiment”. In their rebuttal letter, the authors argue that peripheral immune cells might play a role during myelin repair in their model. While this might well be true, this claim is not substantiated by data. Beyond, DHT and Testosterone-treated EAE mice show a milder EAE disease score days after the occurrence of first symptoms, which is maybe a bit too fast for remyelination-mediated effects.

REPLY: Actually, we wanted to say that even though EAE is not suitable for studying the remyelination process, it is important to consider demyelination / remyelination both in the context of immune-mediated and nonimmune-mediated animal models of CNS demyelination. In the former, phagocytes and T cells are known to be included in a vicious circle where the pro-inflammatory phagocytes promote the activation of T cells while conversely T cells induce myeloid cells to become pro-inflammatory giving rise to a microenvironment detrimental for myelin regeneration (Codarri et al, *Nat Immunol*, 2011; <https://doi.org/10.1038/ni.2027>). In addition, it was proposed that some subsets of macrophages actively participate in the destructive demyelination process (Croxford et al, *Trends Immunol*, 2015; doi:

10.1016/j.it.2015.08.004). This is the reason why we introduced a short sentence in our R1 revision indicating that ‘remyelination cannot be considered independently of the peripheral immune process characterizing MS’ (lines 188-189). We agree with the reviewer to say that the milder EAE disease scores induced by DHT and testosterone at 8 days after the onset of the neurological symptoms is likely the result of the decrease in deleterious T cells and cytokines (as shown by our FACS analyses and cytokine dosages). However, these milder scores may also be related to remyelination since our GO analyses identified DHT-induced upregulation of genes involved in (re)myelination.

REVIEWERS' COMMENTS_R3

Reviewer #1 (Remarks to the Author):

The manuscript presents very important findings concerning sex differences in context of androgen signaling and demyelination within the CNS. The revisions have made this manuscript much stronger. Along these lines, this revision further improved the manuscript, and now only very few points need to be addressed.

1.Line 74: DHT mentioned here the first time. Abbreviation needs to be introduced here

The text has been edited.

2.The manuscript would benefit greatly from a schematic figure (graphical abstract) summarizing the many findings that differ between females and males.

A graphical abstract is now provided.

Reviewer #3 (Remarks to the Author):

The authors have addressed my concerns, I can, thus, recommend publication of this nice work.